# Solving stochastic weak Minty variational inequalities without increasing batch size

**Thomas Pethick**[*]  **Olivier Fercoq**[†]  **Puya Latafat**[‡]  **Panagiotis Patrinos**[‡]  **Volkan Cevher**[*]

## Abstract

This paper introduces a family of stochastic extragradient-type algorithms for a class of nonconvex-nonconcave problems characterized by the weak Minty variational inequality (MVI). Unlike existing results on extragradient methods in the monotone setting, employing diminishing stepsizes is no longer possible in the weak MVI setting. This has led to approaches such as increasing batch sizes per iteration which can however be prohibitively expensive. In contrast, our proposed methods involves two stepsizes and only requires one additional oracle evaluation per iteration. We show that it is possible to keep one fixed stepsize while it is only the second stepsize that is taken to be diminishing, making it interesting even in the monotone setting. Almost sure convergence is established and we provide a unified analysis for this family of schemes which contains a nonlinear generalization of the celebrated primal dual hybrid gradient algorithm.

## 1 Introduction

Stochastic first-order methods have been at the core of the current success in deep learning applications. These methods are mostly well-understood for minimization problems at this point. This is even the case in the nonconvex setting where there exists matching upper and lower bounds on the complexity for finding an approximately stable point (Arjevani et al., 2019).

The picture becomes less clear when moving beyond minimization into nonconvex-nonconcave minimax problems—or more generally nonmonotone variational inequalities. Even in the deterministic case, finding a stationary point is in general intractable (Daskalakis et al., 2021; Hirsch & Vavasis, 1987). This is in stark contrast with minimization where only global optimality is NP-hard.

An interesting nonmonotone class for which we *do* have efficient algorithms is characterized by the so called *weak Minty variational inequality* (MVI) (Diakonikolas et al., 2021). This problem class captures nontrivial structures such as attracting limit cycles and is governed by a parameter $\rho$ whose negativity increases the degree of nonmonotonicity. It turns out that the stepsize $\gamma$ for the exploration step in extragradient-type schemes lower bounds the problem class through $\rho > -\gamma/2$ (Pethick et al., 2022). In other words, it seems that we need to take $\gamma$ large to guarantee convergence for a large class.

This reliance on a large stepsize is at the core of why the community has struggled to provide a stochastic variants for weak MVIs. The only known results effectively increase the batch size *at every iteration* (Diakonikolas et al., 2021, Thm. 4.5)—a strategy that would be prohibitively expensive in most machine learning applications. Pethick et al. (2022) proposed (SEG+) which attempts to tackle the noise by only diminishing the second stepsize. This suffices in the special case of unconstrained quadratic games but can fail even in the monotone case as illustrated in Figure 1. This naturally raises the following research question:

*Can stochastic weak Minty variational inequalities be solved without increasing the batch size?*

We resolve this open problem in the affirmative when the stochastic oracles are Lipschitz in mean, with a modification of stochastic extragradient called *bias-corrected stochastic extragradient* (BC-SEG+). The scheme only requires one additional first order oracle call, while crucially maintaining the fixed stepsize. Specifically, we make the following contributions:

---

[*]Laboratory for Information and Inference Systems (LIONS), EPFL (thomas.pethick@epfl.ch)
[†]Laboratoire Traitement et Communication d'Information, Télécom Paris, Institut Polytechnique de Paris
[‡]Department of Electrical Engineering (ESAT-STADIUS), KU Leuven

*(i)* We show that it is possible to converge for weak MVI *without* increasing the batch size, by introducing a bias-correction term. The scheme introduces *no additional hyperparameters* and recovers the maximal range $\rho \in (-\gamma/2, \infty)$ of explicit deterministic schemes. The rate we establish is interesting already in the star-monotone case where only *asymptotic* convergence of the norm of the operator was known when refraining from increasing the batch size (Hsieh et al., 2020, Thm. 1). Our result additionally carries over to another class of problem treated in Appendix G, which we call *negative* weak MVIs.

*(ii)* We generalize the result to a whole family of schemes that can treat constrained and regularized settings. First and foremost the class includes a generalization of the forward-backward-forward (FBF) algorithm of Tseng (2000) to stochastic weak MVIs. The class also contains a stochastic nonlinear extension of the celebrated primal dual hybrid gradient (PDHG) algorithm (Chambolle & Pock, 2011). Both methods are obtained as instantiations of the same template scheme, thus providing a unified analysis and revealing an interesting requirement on the update under weak MVI when only stochastic feedback is available.

*(iii)* We prove almost sure convergence under the classical Robbins-Monro stepsize schedule of the second stepsize. This provides a guarantee on the last iterate, which is especially important in the nonmonotone case, where average guarantees cannot be converted into a single candidate solution. Almost sure convergence is challenging already in the monotone case where even stochastic extragradient may not converge (Hsieh et al., 2020, Fig. 1).

## 2 RELATED WORK

**Weak MVI**   Diakonikolas et al. (2021) was the first to observe that an extragradient-like scheme called extragradient+ (EG+) converges globally for weak MVIs with $\rho \in (-1/8L_F, \infty)$. This results was later tightened to $\rho \in (-1/2L_F, \infty)$ and extended to constrained and regularized settings in (Pethick et al., 2022). A single-call variant has been analysed in Böhm (2022). Weak MVI is a star variant of cohypomonotonicity, for which an inexact proximal point method was originally studied in Combettes & Pennanen (2004). Later, a tight characterization was carried out by Bauschke et al. (2021) for the exact case. It was shown that acceleration is achievable for an extragradient-type scheme even for cohypomonotone problems (Lee & Kim, 2021). Despite this array of positive results the stochastic case is largely untreated for weak MVIs. The only known result (Diakonikolas et al., 2021, Theorem 4.5) requires the batch size to be increasing. Similarly, the accelerated method in Lee & Kim (2021, Thm. 6.1) requires the variance of the stochastic oracle to decrease as $O(1/k)$.

**Stochastic & monotone**   When more structure is present the story is different since diminishing stepsizes becomes permissible. In the monotone case rates for the gap function was obtained for stochastic Mirror-Prox in Juditsky et al. (2011) under bounded domain assumption, which was later relaxed for the extragradient method under additional assumptions (Mishchenko et al., 2020). The norm of the operator was shown to asymptotically converge for unconstrained MVIs in Hsieh et al. (2020) with a double stepsize policy. There exists a multitude of extensions for monotone problems: Single-call stochastic methods are covered in detail by Hsieh et al. (2019), variance reduction was applied to Halpern-type iterations (Cai et al., 2022), cocoercivity was used in Beznosikov et al. (2022), and bilinear games studied in Li et al. (2022). Beyond monotonicity, a range of structures have been explored such as MVIs (Song et al., 2020), pseudomonotonicity (Kannan & Shanbhag, 2019; Boţ et al., 2021), two-sided Polyak-Łojasiewicz condition (Yang et al., 2020), expected cocoercivity (Loizou et al., 2021), sufficiently bilinear (Loizou et al., 2020), and strongly star-monotone (Gorbunov et al., 2022).

**Variance reduction**   The assumptions we make about the stochastic oracle in Section 3 are similar to what is found in the variance reduction literature (see for instance Alacaoglu & Malitsky (2021, Assumption 1) or Arjevani et al. (2019)). However, our use of the assumption are different in a crucial way. Whereas the variance reduction literature uses the stepsize $\gamma \propto 1/L_{\hat{F}}$ (see e.g. Alacaoglu & Malitsky (2021, Theorem 2.5)), we aim at using the much larger $\gamma \propto 1/L_F$. For instance, in the special case of a finite sum problem of size $N$, the mean square smoothness constant $L_{\hat{F}}$ from Assumption III can be $\sqrt{N}$ times larger than $L_F$ (see Appendix I for details). This would lead to a prohibitively strict requirement on the degree of allowed nonmonotonicity through the relationship $\rho > -\gamma/2$.

**Bias-correction** The idea of adding a correction term has also been exploited in minimization, specifically in the context of compositional optimization Chen et al. (2021). Due to their distinct problem setting it suffices to simply extend stochastic gradient descent (SGD), albeit under additional assumptions such as (Chen et al., 2021, Assumption 3). In our setting, however, SGD is not possible even when restricting ourselves to monotone problems.

## 3 PROBLEM FORMULATION AND PRELIMINARIES

We are interested in finding $z \in \mathbb{R}^n$ such that the following inclusion holds,

$$0 \in Tz := Az + Fz. \tag{3.1}$$

A wide range of machine learning applications can be cast as an inclusion. Most noticeable, a structured minimax problem can be reduced to (3.1) as shown in Section 8.1. We will rely on common notation and concepts from monotone operators (see Appendix B for precise definitions).

**Assumption I.** *In problem* (3.1),

*(i) The operator $F : \mathbb{R}^n \to \mathbb{R}^n$ is $L_F$-Lipschitz with $L_F \in [0, \infty)$, i.e.,*

$$\|Fz - Fz'\| \le L_F \|z - z'\| \quad \forall z, z' \in \mathbb{R}^n. \tag{3.2}$$

*(ii) The operator $A : \mathbb{R}^n \rightrightarrows \mathbb{R}^n$ is a maximally monotone operator.*

*(iii) Weak Minty variational inequality (MVI) holds, i.e., there exists a nonempty set $\mathcal{S}^\star \subseteq \mathbf{zer}\, T$ such that for all $z^\star \in \mathcal{S}^\star$ and some $\rho \in (-\frac{1}{2L_F}, \infty)$*

$$\langle v, z - z^\star \rangle \ge \rho \|v\|^2, \quad \text{for all } (z, v) \in \mathbf{gph}\, T. \tag{3.3}$$

**Remark 1.** In the unconstrained and smooth case ($A \equiv 0$), Assumption I*(iii)* reduces to $\langle Fz, z - z^\star \rangle \ge \rho \|Fz\|^2$ for all $z \in \mathbb{R}^n$. When $\rho = 0$ this condition reduces to the MVI (i.e. star-monotonicity), while negative $\rho$ makes the problem increasingly nonmonotone. Interestingly, the inequality is not symmetric and one may instead consider that the assumption holds for $-F$. Through this observation, Appendix G extends the reach of the extragradient-type algorithms developed for weak MVIs.

**Stochastic oracle** We assume that we cannot compute $Fz$ easily, but instead we have access to the stochastic oracle $\hat{F}(z, \xi)$, which we assume is unbiased with bounded variance. We additionally assume that $z \mapsto \hat{F}(z, \xi)$ is $L_{\hat{F}}$ Lipschitz continuous in mean and that it can be simultaneously queried under the same randomness.

**Assumption II.** *For the operator $\hat{F}(\cdot, \xi) : \mathbb{R}^n \to \mathbb{R}^n$ the following holds.*

*(i) Two-point oracle: The stochastic oracle can be queried for any two points $z, z' \in \mathbb{R}^n$,*

$$\hat{F}(z, \xi), \hat{F}(z', \xi) \quad \text{where} \quad \xi \sim \mathcal{P}. \tag{3.4}$$

*(ii) Unbiased: $\mathbb{E}_\xi\big[\hat{F}(z, \xi)\big] = Fz \quad \forall z \in \mathbb{R}^n$.*

*(iii) Bounded variance: $\mathbb{E}_\xi\big[\|\hat{F}(z, \xi) - \hat{F}(z)\|^2\big] \le \sigma_F^2 \quad \forall z \in \mathbb{R}^n$.*

**Assumption III.** *The operator $\hat{F}(\cdot, \xi) : \mathbb{R}^n \to \mathbb{R}^n$ is Lipschitz continuous in mean with $L_{\hat{F}} \in [0, \infty)$:*

$$\mathbb{E}_\xi\big[\|\hat{F}(z, \xi) - \hat{F}(z', \xi)\|^2\big] \le L_{\hat{F}}^2 \|z - z'\|^2 \quad \text{for all } z, z' \in \mathbb{R}^n. \tag{3.5}$$

**Remark 2.** Assumptions II*(i)* and III are also common in the variance reduction literature (Fang et al., 2018; Nguyen et al., 2019; Alacaoglu & Malitsky, 2021), but in contrast with variance reduction we will not necessarily need knowledge of $L_{\hat{F}}$ to specify the algorithm, in which case the problem constant will only affect the complexity. Crucially, this decoupling of the stepsize from $L_{\hat{F}}$ will allow the proposed scheme to converge for a larger range of $\rho$ in Assumption I*(iii)*. Finally, note that Assumption II*(i)* commonly holds in machine learning applications, where usually the stochasticity is induced by the sampled mini-batch.

## 4 METHOD

To arrive at a stochastic scheme for weak MVI we first need to understand the crucial ingredients in the deterministic setting. For simplicity we will initially consider the unconstrained and smooth

---

**Algorithm 1** (BC-SEG+) Stochastic algorithm for problem (3.1) when $A \equiv 0$

---

REQUIRE $\quad z^{-1} = \bar{z}^{-1} = z^0 \in \mathbb{R}^n \; \alpha_k \in (0,1), \; \gamma \in (\lfloor -2\rho \rfloor_+, 1/L_F)$
REPEAT for $k = 0, 1, \ldots$ until convergence
**1**.1: $\quad$ Sample $\xi_k \sim \mathcal{P}$
**1**.2: $\quad \bar{z}^k = z^k - \gamma \hat{F}(z^k, \xi_k) + (1 - \alpha_k)(\bar{z}^{k-1} - z^{k-1} + \gamma \hat{F}(z^{k-1}, \xi_k))$
**1**.3: $\quad$ Sample $\bar{\xi}_k \sim \mathcal{P}$
**1**.4: $\quad z^{k+1} = z^k - \alpha_k \gamma \hat{F}(\bar{z}^k, \bar{\xi}_k)$
RETURN $z^{k+1}$

---

setting, i.e. $A \equiv 0$ in (3.1). The first component is taking the second stepsize $\alpha$ smaller as done in extragradient+ (EG+),

$$
\begin{aligned}
\bar{z}^k &= z^k - \gamma F z^k \\
z^{k+1} &= z^k - \alpha \gamma F \bar{z}^k
\end{aligned}
\tag{EG+}
$$

where $\alpha \in (0, 1)$. Convergence in weak MVI was first shown in Diakonikolas et al. (2021) and later tightened by Pethick et al. (2022), who characterized that smaller $\alpha$ allows for a larger range of the problem constant $\rho$. Taking $\alpha$ small is unproblematic for a stochastic scheme where usually the stepsize is taken diminishing regardless.

However, Pethick et al. (2022) also showed that the extrapolation stepsize $\gamma$ plays a critical role for convergence under weak MVI. Specifically, they proved that a larger stepsize $\gamma$ leads to a looser bound on the problem class through $\rho > -\gamma/2$. While a lower bound has not been established we provide an example in Figure 3 of Appendix H where small stepsize prevents convergence. Unfortunately, picking $\gamma$ large (e.g. as $\gamma = 1/L_F$) causes significant complications in the stochastic case where both stepsizes are usually taken to be diminishing as in the following scheme,

$$
\begin{aligned}
\bar{z}^k &= z^k - \beta_k \gamma \hat{F}(z^k, \xi_k) \quad \text{with} \quad \xi_k \sim \mathcal{P} \\
z^{k+1} &= z^k - \alpha_k \gamma \hat{F}(\bar{z}^k, \bar{\xi}_k) \quad \text{with} \quad \bar{\xi}_k \sim \mathcal{P}
\end{aligned}
\tag{SEG}
$$

where $\alpha_k = \beta_k \propto 1/k$. Even with a two-timescale variant (when $\beta_k > \alpha_k$) it has only been possible to show convergence for MVI (i.e. when $\rho = 0$) (Hsieh et al., 2020). Instead of decreasing both stepsizes, Pethick et al. (2022) proposes a scheme that keeps the first stepsize constant,

$$
\begin{aligned}
\bar{z}^k &= z^k - \gamma \hat{F}(z^k, \xi_k) \quad \text{with} \quad \xi_k \sim \mathcal{P} \\
z^{k+1} &= z^k - \alpha_k \gamma \hat{F}(\bar{z}^k, \bar{\xi}_k) \quad \text{with} \quad \bar{\xi}_k \sim \mathcal{P}
\end{aligned}
\tag{SEG+}
$$

However, (SEG+) does not necessarily converge even in the monotone case as we illustrate in Figure 1. The non-convergence stems from the bias term introduced by the randomness of $\bar{z}^k$ in $\hat{F}(\bar{z}^k, \bar{\xi}_k)$. Intuitively, the role of $\bar{z}^k$ is to approximate the deterministic exploration step $\tilde{\bar{z}}^k := z^k - \gamma F z^k$. While $\bar{z}^k$ is an unbiased estimate of $\tilde{\bar{z}}^k$ this does not imply that $\hat{F}(\bar{z}^k, \bar{\xi}_k)$ is an unbiased estimate of $F(\tilde{\bar{z}}^k)$. Unbiasedness only holds in special cases, such as when $F$ is linear and $A \equiv 0$ for which we show convergence of (SEG+) in Section 5 under weak MVI. In the monotone case it suffice to take the exploration stepsize $\gamma$ diminishing (Hsieh et al., 2020, Thm. 1), but this runs counter to the fixed stepsize requirement of weak MVI.

Instead we propose *bias-corrected stochastic extragradient+* (BC-SEG+) in Algorithm 1. BC-SEG+ adds a bias correction term of the previous operator evaluation using the current randomness $\xi_k$. This crucially allows us to keep the first stepsize fixed. We further generalize this scheme to constrained and regularized setting with Algorithm 2 by introducing the use of the resolvent, $(\mathrm{id} + \gamma A)^{-1}$.

## 5 ANALYSIS OF SEG+

In the special case where $F$ is affine and $A \equiv 0$ we can show convergence of (SEG+) under weak MVI up to arbitrarily precision even with a large stepsize $\gamma$.

**Theorem 5.1.** *Suppose that Assumptions I and II hold. Assume $Fz := Bz + v$ and choose $\alpha_k \in (0, 1)$ and $\gamma \in (0, 1/L_F)$ such that $\rho \geq \gamma(\alpha_k - 1)/2$. Consider the sequence $(z^k)_{k \in \mathbb{N}}$ generated by (SEG+). Then for all $z^\star \in \mathcal{S}^\star$,*

$$
\sum_{k=0}^{K} \frac{\alpha_k}{\sum_{j=0}^{K} \alpha_j} \mathbb{E} \|Fz^k\|^2 \leq \frac{\|z^0 - z^\star\|^2 + \gamma^2(\gamma^2 L_F^2 + 1)\sigma_F^2 \sum_{j=0}^{K} \alpha_j^2}{\gamma^2(1 - \gamma^2 L_F^2)\sum_{j=0}^{K} \alpha_j}.
\tag{5.1}
$$

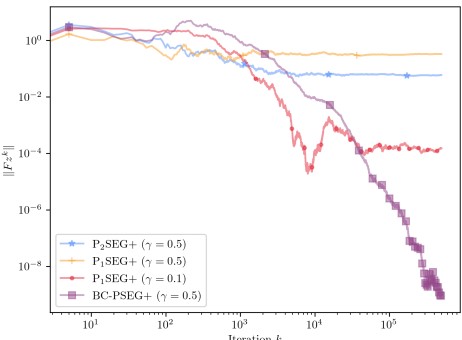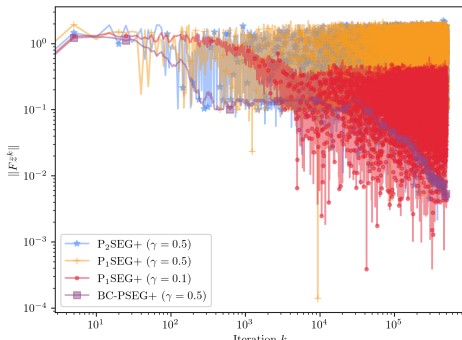

**Figure 1:** *Monotone constrained case illustrating the issue for projected variants of* (SEG+) *(see Appendix H.2 for algorithmic details). The objective is bilinear* $\phi(x,y) = (x - 0.9) \cdot (y - 0.9)$ *under box constraints* $\|(x,y)\|_\infty \leq 1$. *The unique stationary point* $(x^\star, y^\star) = (0.9, 0.9)$ *lies in the interior, so even* $\|Fz\|$ *can be driven to zero. Despite the simplicity of the problem both projected variants of* (SEG+) *only converges to a* $\gamma$-*dependent neighborhood. For weak MVI with* $\rho < 0$ *this neighborhood cannot be made arbitrarily small since* $\gamma$ *cannot be taken arbitrarily small (see Figure 3 of Appendix H).*

The underlying reason for this positive results is that $\hat{F}(\bar{z}^k, \bar{\xi}_k)$ is unbiased when $F$ is linear. This no longer holds when either linearity of $F$ is dropped or when the resolvent is introduced for $A \not\equiv 0$, in which case the scheme only converges to a $\gamma$-dependent neighborhood as illustrated in Figure 1. This is problematic in weak MVI where $\gamma$ cannot be taken arbitrarily small (see Figure 3 of Appendix H).

## 6  ANALYSIS FOR UNCONSTRAINED AND SMOOTH CASE

For simplicity we first consider the case where $A \equiv 0$. To mitigate the bias introduced in $F(\bar{z}^k, \bar{\xi}_k)$ for (SEG+), we propose Algorithm 1 which modifies the exploration step. The algorithm can be seen as a particular instance of the more general scheme treated in Section 7.

**Theorem 6.1.** *Suppose that Assumptions I to III hold. Suppose in addition that* $\gamma \in (\lfloor -2\rho \rfloor_+, 1/L_F)$ *and* $(\alpha_k)_{k \in \mathbb{N}} \subset (0,1)$ *is a diminishing sequence such that*

$$2\gamma L_{\hat{F}} \sqrt{\alpha_0} + \left(1 + \left(\frac{1+\gamma^2 L_F^2}{1-\gamma^2 L_F^2}\gamma^2 L_F^2\right)\gamma^2 L_{\hat{F}}^2\right)\alpha_0 \leq 1 + \frac{2\rho}{\gamma}. \tag{6.1}$$

*Then, the following estimate holds for all* $z^\star \in \mathcal{S}^\star$

$$\mathbb{E}[\|F(z^{k_\star})\|^2] \leq \frac{(1 + \eta\gamma^2 L_F^2)\|z^0 - z^\star\|^2 + C\sigma_F^2\gamma^2 \sum_{j=0}^{K} \alpha_j^2}{\mu \sum_{j=0}^{K} \alpha_j} \tag{6.2}$$

*where* $C = 1 + 2\eta((\gamma^2 L_{\hat{F}}^2 + 1) + 2\alpha_0)$, $\eta = \frac{1}{2}\frac{1+\gamma^2 L_F^2}{1-\gamma^2 L_F^2}\gamma^2 L_F^2 + \frac{1}{\gamma L_{\hat{F}} \sqrt{\alpha_0}}$, $\mu = \gamma^2(1 - \gamma^2 L_F^2)/2$ *and* $k_\star$ *is chosen from* $\{0, 1, \ldots, K\}$ *according to probability* $\mathcal{P}[k_\star = k] = \frac{\alpha_k}{\sum_{j=0}^{K} \alpha_j}$.

**Remark 6.2.** *As* $\alpha_0 \to 0$, *the requirement* (6.1) *reduces to* $\rho > -\gamma/2$ *as in the deterministic setting of Pethick et al. (2022). Letting* $\alpha_k = \alpha_0/\sqrt{k+r}$ *the rate becomes* $O(1/\sqrt{k})$, *thus matching the rate for the gap function of stochastic extragradient in the monotone case (see e.g. Juditsky et al. (2011)).* ☐

The above result provides a rate for a *random iterate* as pioneered by Ghadimi & Lan (2013). Showing *last iterate* results even asymptotically is more challenging. Already in the monotone case, vanilla (SEG) (where $\beta_k = \alpha_k$) only has convergence guarantees for the average iterate (Juditsky et al., 2011). In fact, the scheme can cycle even in simple examples (Hsieh et al., 2020, Fig. 1).

Under the classical (but more restrictive) Robbins-Monro stepsize policy, it is possible to show almost sure convergence for the iterates generates by Algorithm 1. The following theorem demonstrates the result in the particular case of $\alpha_k = 1/k+r$. The more general statement is deferred to Appendix D.

**Theorem 6.3** (almost sure convergence). *Suppose that Assumptions I to III hold. Suppose* $\gamma \in (\lfloor -2\rho \rfloor_+, 1/L_F)$, $\alpha_k = \frac{1}{k+r}$ *for any positive natural number* $r$ *and*

$$(\gamma L_{\hat{F}} + 1)\alpha_k + 2\left(\frac{1+\gamma^2 L_F^2}{1-\gamma^2 L_F^2}\gamma^4 L_F^2 L_{\hat{F}}^2 \alpha_{k+1} + \gamma L_{\hat{F}}\right)(\alpha_{k+1} + 1)\alpha_{k+1} \leq 1 + \frac{2\rho}{\gamma}. \tag{6.3}$$

---

**Algorithm 2** (BC-PSEG+) Stochastic algorithm for problem (3.1)

---

REQUIRE $z^{-1} = z^0 \in \mathbb{R}^n$, $h^{-1} \in \mathbb{R}^n$, $\alpha_k \in (0,1)$, $\gamma \in (\lfloor -2\rho \rfloor_+, 1/L_F)$
REPEAT for $k = 0, 1, \dots$ until convergence
2.1: Sample $\xi_k \sim \mathcal{P}$
2.2: $h^k = (z^k - \gamma \hat{F}(z^k, \xi_k)) + (1 - \alpha_k)\left(h^{k-1} - (z^{k-1} - \gamma \hat{F}(z^{k-1}, \xi_k))\right)$
2.3: $\bar{z}^k = (\mathrm{id} + \gamma A)^{-1} h^k$
2.4: Sample $\bar{\xi}_k \sim \mathcal{P}$
2.5: $z^{k+1} = z^k - \alpha_k(h^k - \bar{z}^k + \gamma \hat{F}(\bar{z}^k, \bar{\xi}_k))$
RETURN $z^{k+1}$

---

*Then, the sequence $(z^k)_{k \in \mathbb{N}}$ generated by Algorithm 1 converges almost surely to some $z^\star \in$ zer $T$.*

**Remark 6.4.** As $\alpha_k \to 0$ the condition on $\rho$ reduces to $\rho > -\gamma/2$ like in the deterministic case. $\qquad \square$

To make the results more accessible, both theorems have made particular choices of the free parameters from the proof, that ensures convergence for a given $\rho$ and $\gamma$. However, since the parameters capture inherent tradeoffs, the choice above might not always provide the tightest rate. Thus, the more general statements of the theorems have been preserved in the appendix.

## 7 ANALYSIS FOR CONSTRAINED CASE

The result for the unconstrained smooth case can be extended when the resolvent is available. Algorithm 2 provides a direct generalization of the unconstrained Algorithm 1. The construction relies on approximating the deterministic algorithm proposed in Pethick et al. (2022), which iteratively projects onto a half-space which is guaranteed to contain the solutions. By defining $Hz = z - \gamma Fz$, the scheme can concisely be written as,

$$
\begin{aligned}
\bar{z}^k &= (I + \gamma A)^{-1}(Hz^k) \\
z^{k+1} &= z^k - \alpha_k(Hz^k - H\bar{z}^k),
\end{aligned}
\tag{CEG+}
$$

for a particular adaptive choice of $\alpha_k \in (0,1)$. With a fair amount of hindsight we choose to replace $Hz^k$ with the bias-corrected estimate $h^k$ (as defined in Step 2.2 in Algorithm 2), such that the estimate is also reused in the second update.

**Theorem 7.1.** *Suppose that Assumptions I to III hold. Moreover, suppose that $\alpha_k \in (0,1)$, $\gamma \in (\lfloor -2\rho \rfloor_+, 1/L_F)$ and the following holds,*

$$
\mu := \frac{1 - \sqrt{\alpha_0}}{1 + \sqrt{\alpha_0}} - \alpha_0(1 + 2\gamma^2 L_{\hat{F}}^2 \eta) + \frac{2\rho}{\gamma} > 0
\tag{7.1}
$$

*where $\eta \geq \frac{1}{\sqrt{\alpha_0}(1 - \gamma^2 L_{\hat{F}}^2)} + \frac{1 - \sqrt{\alpha_0}}{\sqrt{\alpha_0}}$. Consider the sequence $(z^k)_{k \in \mathbb{N}}$ generated by Algorithm 2. Then, the following estimate holds for all $z^\star \in \mathcal{S}^\star$*

$$
\mathbb{E}[\mathbf{dist}(0, T\bar{z}^{k_\star})^2] \leq \frac{\mathbb{E}[\|z^0 - z^\star\|^2] + \eta \mathbb{E}[\|h^{-1} - Hz^{-1}\|^2] + C\gamma^2 \sigma_F^2 \sum_{j=0}^K \alpha_j^2}{\gamma^2 \mu \sum_{j=0}^K \alpha_j}
$$

*where $C = 1 + 2\eta(1 + \gamma^2 L_{\hat{F}}^2) + 2\alpha_0 \eta$ and $k_\star$ is chosen from $\{0, 1, \dots, K\}$ according to probability $\mathcal{P}[k_\star = k] = \frac{\alpha_k}{\sum_{j=0}^K \alpha_j}$.*

**Remark 3.** The condition on $\rho$ in (7.1) reduces to $\rho > -\gamma/2$ when $\alpha_0 \to 0$ as in the deterministic case. As oppose to Theorem 6.3 which tracks $\|Fz^k\|^2$, the convergence measure of Theorem 7.1 reduces to $\mathbf{dist}(0, T\bar{z}^k)^2 = \|F\bar{z}^k\|^2$ when $A \equiv 0$. Since Algorithm 1 and Algorithm 2 coincide when $A \equiv 0$, Theorem 7.1 also applies to Algorithm 1 in the unconstrained case. Consequently, we obtain rates for both $\|F\bar{z}^k\|^2$ and $\|Fz^k\|^2$ in the unconstrained smooth case.

## 8 ASYMMETRIC & NONLINEAR PRECONDITIONING

In this section we show that the family of stochastic algorithms which converges under weak MVI can be expanded beyond Algorithm 2. This is achieved by extending (CEG+) through introducing

---

**Algorithm 3** Nonlinearly preconditioned primal dual extragradient (NP-PDEG) for solving (8.5)

---

REQUIRE $z^{-1} = z^0 = (x^0, y^0)$ with $x^0, x^{-1}, \hat{x}^{-1}, \bar{x}^{-1} \in \mathbb{R}^n$, $y^0, y^{-1} \in \mathbb{R}^r$, $\theta \in [0, \infty)$, $\Gamma_1 > 0$, $\Gamma_2 > 0$
REPEAT for $k = 0, 1, \dots$ until convergence
3.1: $\quad \xi_k \sim \mathcal{P}$
3.2: $\quad \hat{x}^k = x^k - \Gamma_1 \nabla_x \hat{\varphi}(z^k, \xi_k) + (1 - \alpha_k)(\hat{x}^{k-1} - x^{k-1} + \Gamma_1 \nabla_x \hat{\varphi}(x^{k-1}, y^{k-1}, \xi_k))$
3.3: $\quad \bar{x}^k = \mathbf{prox}_f^{\Gamma_1^{-1}}(\hat{x}^k)$
3.4: $\quad \xi_k' \sim \mathcal{P}$
3.5: $\quad \hat{y}^k = y^k + \Gamma_2(\theta \nabla_y \hat{\varphi}(\bar{x}^k, y^k, \xi_k') + (1 - \theta) \nabla_y \hat{\varphi}(z^k, \xi_k))$
3.6: $\quad\quad\quad + (1 - \alpha_k)\big(\hat{y}^{k-1} - y^{k-1} - \Gamma_2(\theta \nabla_y \hat{\varphi}(\bar{x}^{k-1}, y^{k-1}, \xi_k') + (1 - \theta) \nabla_y \hat{\varphi}(z^{k-1}, \xi_k))\big)$
3.7: $\quad \bar{y}^k = \mathbf{prox}_g^{\Gamma_2^{-1}}(\hat{y}^k)$
3.8: $\quad \bar{\xi}_k \sim \mathcal{P}$
3.9: $\quad x^{k+1} = x^k + \alpha_k\big(\bar{x}^k - \hat{x}^k - \Gamma_1 \nabla_x \hat{\varphi}(\bar{z}^k, \bar{\xi}_k)\big)$
3.10: $\quad y^{k+1} = y^k + \alpha_k\big(\bar{y}^k - \hat{y}^k + \Gamma_2 \nabla_y \hat{\varphi}(\bar{z}^k, \bar{\xi}_k)\big)$
RETURN $z^{k+1} = (x^{k+1}, y^{k+1})$

---

a nonlinear and asymmetrical preconditioning. Asymmetrical preconditioning has been used in the literature to unify a large range of algorithm in the monotone setting Latafat & Patrinos (2017). A subtle but crucial difference, however, is that the preconditioning considered here depends nonlinearly on the current iterate. As it will be shown in Section 8.1 this nontrivial feature is the key for showing convergence for primal-dual algorithms in the nonmonotone setting.

Consider the following generalization of (CEG+) by introducing a potentially *asymmetric nonlinear* preconditioning $P_{z^k}$ that depends on the current iterate $z^k$.

$$\text{find } \bar{z}^k \text{ such that} \quad H_{z^k}(z^k) \in P_{z^k}(\bar{z}^k) + A(\bar{z}^k), \tag{8.1a}$$

$$\text{update} \quad z^{k+1} = z^k + \alpha \Gamma\big(H_{z^k}(\bar{z}^k) - H_{z^k}(z^k)\big). \tag{8.1b}$$

where $H_u(v) := P_u(v) - F(v)$ and $\Gamma$ is some positive definite matrix. The iteration independent and diagonal choice $P_{z^k} = \gamma^{-1}I$ and $\Gamma = \gamma I$ correspond to the basic (CEG+). More generally we consider

$$P_u(z) := \Gamma^{-1}z + Q_u(z) \tag{8.2}$$

where $Q_u(z)$ captures the nonlinear and asymmetric part, which ultimately enables alternating updates and relaxing the Lipschitz conditions (see Remark 8.1*(ii)*). Notice that the iterates above does not always yield well-defined updates and one must inevitably impose additional structures on the preconditioner (we provide sufficient condition in Appendix F.1). Consistently with (8.2), in the stochastic case we define

$$\hat{P}_u(z, \xi) := \Gamma^{-1}z + \hat{Q}_u(z, \xi). \tag{8.3}$$

The proposed stochastic scheme, which introduces a carefully chosen bias-correction term, is summarized as

$$\text{compute} \quad h^k = \hat{P}_{z^k}(z^k, \xi_k) - \hat{F}(z^k, \xi_k) + (1 - \alpha_k)\big(h^{k-1} - \hat{P}_{z^{k-1}}(z^{k-1}, \xi_k) + \hat{F}(z^{k-1}, \xi_k) \tag{8.4a}$$

$$- \hat{Q}_{z^{k-1}}(\bar{z}^{k-1}, \xi_{k-1}') + \hat{Q}_{z^{k-1}}(\bar{z}^{k-1}, \xi_k')\big) \quad \text{with} \quad \xi_k, \xi_k' \sim \mathcal{P}$$

$$\text{find } \bar{z}^k \text{ such that} \quad h^k \in \hat{P}_{z^k}(\bar{z}^k, \xi_k') + A\bar{z}^k \tag{8.4b}$$

$$\text{update} \quad z^{k+1} = z^k + \alpha_k \Gamma\big(\hat{P}_{z^k}(\bar{z}^k, \bar{\xi}_k) - \hat{F}(\bar{z}^k, \bar{\xi}_k) - h^k\big) \quad \text{with} \quad \bar{\xi}_k \sim \mathcal{P} \tag{8.4c}$$

**Remark 4.** The two additional terms in (8.4a) are due to the interesting interplay between weak MVI and stochastic feedback, which forces a change of variables (see Appendix F.4).

To make a concrete choice of $\hat{Q}_u(z, \xi)$ we will consider a minimax problem as a motivating example (see Appendix F.1 for a more general setup).

## 8.1 NONLINEARLY PRECONDITIONED PRIMAL DUAL HYBRID GRADIENT

We consider the problem of

$$\underset{x \in \mathbb{R}^n}{\text{minimize}} \underset{y \in \mathbb{R}^r}{\text{maximize}} \quad f(x) + \varphi(x, y) - g(y). \tag{8.5}$$

where $\varphi(x, y) := \mathbb{E}_\xi[\hat{\varphi}(x, y, \xi)]$. The first order optimality conditions may be written as the inclusion

$$0 \in Tz := Az + Fz, \quad \text{where} \quad A = (\partial f, \partial g), \quad F(z) = (\nabla_x \varphi(z), -\nabla_y \varphi(z)), \tag{8.6}$$

while the algorithm only has access to the stochastic estimates $\hat{F}(z, \xi) := (\nabla_x \hat{\varphi}(z, \xi), -\nabla_y \hat{\varphi}(z, \xi))$.

**Assumption IV.** *For problem* (8.5)*, let the following hold with a stepsize matrix* $\Gamma = \mathbf{blkdiag}(\Gamma_1, \Gamma_2)$ *where* $\Gamma_1 \in \mathbb{R}^n$ *and* $\Gamma_2 \in \mathbb{R}^r$ *are symmetric positive definite matrices:*

*(i)* $f, g$ *are proper lsc convex*

*(ii)* $\varphi : \mathbb{R}^{n+r} \to \mathbb{R}$ *is continuously differentiable and for some symmetric positive definite matrices* $D_{xx}, D_{xy}, D_{yx}, D_{yy}$, *the following holds for all* $z = (x, y), z' = (x', y') \in \mathbb{R}^{n+r}$

$$\|\nabla_x \varphi(z') - \nabla_x \varphi(z)\|_{\Gamma_1}^2 \leq L_{xx}^2 \|x' - x\|_{D_{xx}}^2 + L_{xy}^2 \|y' - y\|_{D_{xy}}^2,$$

$$\|\nabla_y \varphi(z') - \theta \nabla_y \varphi(x', y) - (1 - \theta) \nabla_y \varphi(z)\|_{\Gamma_2}^2 \leq L_{yx}^2 \|x' - x\|_{D_{yx}}^2 + L_{yy}^2 \|y' - y\|_{D_{yy}}^2.$$

*(iii) Stepsize condition:* $L_{xx}^2 D_{xx} + L_{yx}^2 D_{yx} \prec \Gamma_1^{-1}$ *and* $L_{xy}^2 D_{xy} + L_{yy}^2 D_{yy} \prec \Gamma_2^{-1}$.

*(iv) Bounded variance:* $\mathbb{E}_\xi\left[\|\hat{F}(z, \xi) - \hat{F}(z', \xi)\|_\Gamma^2\right] \leq \sigma_F^2 \quad \forall z, z' \in \mathbb{R}^n$.

*(v)* $\hat{\varphi}(\cdot, \xi) : \mathbb{R}^{n+r} \to \mathbb{R}$ *is continuously differentiable and for some symmetric positive definite matrices* $D_{\widehat{xz}}, D_{\widehat{yz}}, D_{\widehat{yx}}, D_{\widehat{yy}}$, *the following holds for all* $z = (x, y), z' = (x', y') \in \mathbb{R}^{n+r}$ *and* $v, v' \in \mathbb{R}^n$

$$\text{for } \theta \in [0, \infty): \quad \mathbb{E}_\xi\left[\|\nabla_x \hat{\varphi}(z', \xi) - \nabla_x \hat{\varphi}(z, \xi)\|_{\Gamma_1}^2\right] \leq L_{\widehat{xz}}^2 \|z' - z\|_{D_{\widehat{xz}}}^2$$

$$\text{if } \theta \neq 1: \quad \mathbb{E}_\xi\left[\|\nabla_y \hat{\varphi}(z, \xi) - \nabla_y \hat{\varphi}(z', \xi)\|_{\Gamma_2}^2\right] \leq L_{\widehat{yz}}^2 \|z' - z\|_{D_{\widehat{yz}}}^2$$

$$\text{if } \theta \neq 0: \quad \mathbb{E}_\xi\left[\|\nabla_y \hat{\varphi}(v', y', \xi) - \nabla_y \hat{\varphi}(v, y, \xi)\|_{\Gamma_2}^2\right] \leq L_{\widehat{yx}}^2 \|v' - v\|_{D_{\widehat{yx}}}^2 + L_{\widehat{yy}}^2 \|y' - y\|_{D_{\widehat{yy}}}^2.$$

**Remark 8.1.** In Algorithm 3 the choice of $\theta \in [0, \infty)$ leads to different algorithmic oracles and underlying assumptions in terms of Lipschitz continuity in Assumptions IV*(ii)* and IV*(v)*.

*(i)* If $\theta = 0$ then the first two steps may be computed in parallel and we recover Algorithm 2. Moreover, to ensure Assumption IV*(ii)* in this case it suffices to assume for $L_x, L_y \in [0, \infty)$,

$$\|\nabla_x \varphi(z') - \nabla_x \varphi(z)\| \leq L_x \|z' - z\|, \quad \|\nabla_y \varphi(z') - \nabla_y \varphi(z)\| \leq L_y \|z' - z\|.$$

*(ii)* Taking $\theta = 1$ leads to Gauss-Seidel updates and a nonlinear primal dual extragradient algorithm with sufficient Lipschitz continuity assumptions for some $L_x, L_y \in [0, \infty)$,

$$\|\nabla_x \varphi(z') - \nabla_x \varphi(z)\| \leq L_x \|z' - z\|, \quad \|\nabla_y \varphi(z') - \nabla_y \varphi(x', y)\| \leq L_y \|y' - y\|. \qquad \square$$

Algorithm 3 is an application of (8.4) applied for solving (8.6). In order to cast the algorithm as an instance of the template algorithm (8.4), we choose the positive definite stepsize matrix as $\Gamma = \mathbf{blkdiag}(\Gamma_1, \Gamma_2)$ with $\Gamma_1 > 0$, $\Gamma_2 > 0$, and the nonlinear part of the preconditioner as

$$\hat{Q}_u(\bar{z}, \xi) := \left(0, -\theta \nabla_y \hat{\varphi}(\bar{x}, y, \xi)\right), \quad \text{and} \quad Q_u(\bar{z}) := \left(0, -\theta \nabla_y \varphi(\bar{x}, y)\right) \tag{8.7}$$

where $u = (x, y)$ and $\bar{z} = (\bar{x}, \bar{y})$. Recall $H_u(z) := P_u(z) - F(z)$ and define $S_u(z; \bar{z}) := H_u(z) - Q_u(\bar{z})$. The convergence in Theorem 8.2 depends on the distance between the initial estimate $\Gamma^{-1} \hat{z}^{-1}$ with $\hat{z}^{-1} = (\hat{x}^{-1}, \hat{y}^{-1})$ and the deterministic $S_{z^{-1}}(z^{-1}; \bar{z}^{-1})$. See Appendix B for additional notation.

**Theorem 8.2.** *Suppose that Assumption I(iii) to II(ii) and IV hold. Moreover, suppose that* $\alpha_k \in (0, 1)$, $\theta \in [0, \infty)$ *and the following holds,*

$$\mu := \frac{1 - \sqrt{\alpha_0}}{1 + \sqrt{\alpha_0}} + \frac{2\rho}{\bar{\gamma}} - \alpha_0 - 2\alpha_0(\hat{c}_1 + 2\hat{c}_2(1 + \hat{c}_3))\eta > 0 \quad \text{and} \quad 1 - 4\hat{c}_2\alpha_0 > 0 \tag{8.8}$$

*where* $\bar{\gamma}$ *denotes the smallest eigenvalue of* $\Gamma$, $\eta \geq (1 + 4\hat{c}_2\alpha_0^2)(\frac{1}{\sqrt{\alpha_0}(1 - L_M)^2} + \frac{1 - \sqrt{\alpha_0}}{\sqrt{\alpha_0}})/(1 - 4\hat{c}_2\alpha_0)$ *and*

$$\hat{c}_1 := L_{\widehat{xz}}^2 \|\Gamma D_{\widehat{xz}}\| + 2(1 - \theta)^2 L_{\widehat{yz}}^2 \|\Gamma D_{\widehat{yz}}\| + 2\theta^2 L_{\widehat{yy}}^2 \|\Gamma_2 D_{\widehat{yy}}\|, \quad \hat{c}_2 := 2\theta^2 L_{\widehat{yx}}^2 \|\Gamma_1 D_{\widehat{yx}}\|, \quad \hat{c}_3 := L_{\widehat{xz}}^2 \|\Gamma D_{\widehat{xz}}\|,$$

$$L_M^2 := \max\{L_{xx}^2 \|D_{xx}\Gamma_1\| + L_{yx}^2 \|D_{yx}\Gamma_1\|, \|L_{xy}^2\|D_{xy}\Gamma_2\| + L_{yy}^2 \|D_{yy}\Gamma_2\|\}.$$

*Consider the sequence* $(z^k)_{k \in \mathbb{N}}$ *generated by Algorithm 3. Then, the following holds for all* $z^\star \in \mathcal{S}^\star$

$$\mathbb{E}[\mathbf{dist}_\Gamma(0, T\bar{z}^{k_\star})^2] \leq \frac{\mathbb{E}[\|z^0 - z^\star\|_{\Gamma^{-1}}^2] + \eta \mathbb{E}[\|\Gamma^{-1}\hat{z}^{-1} - S_{z^{-1}}(z^{-1}; \bar{z}^{-1})\|_\Gamma^2] + C\sigma_F^2 \sum_{j=0}^K \alpha_j^2}{\mu \sum_{j=0}^K \alpha_j}$$

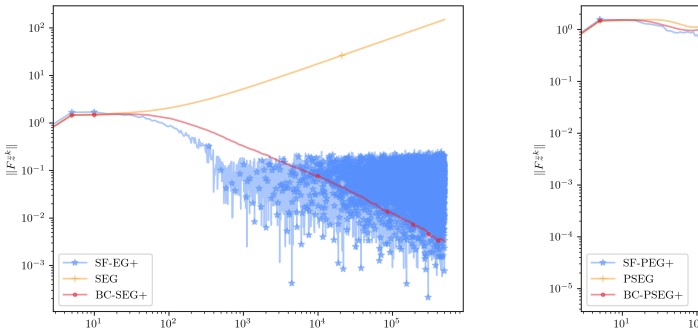

**Figure 2:** *Comparison of methods in the unconstrained setting of Example 2 (left) and the constrained setting of Example 3 (right). Notice that only BC-SEG+ and BC-PSEG+ converges properly while (SEG) diverges, (PSEG) cycles and both (SF-EG+) and (SF-PEG+) only converge to a neighborhood. BC-(P)SEG+ is guaranteed to converge with probability 1 as established through Theorem 6.3 and ??.*

where $C := 2(\eta + \alpha_0(\frac{1}{\sqrt{\alpha_0}(1-L_M)^2} + \frac{1-\sqrt{\alpha_0}}{\sqrt{\alpha_0}}))(1 + 2\hat{c}_2) + 1 + 2(\hat{c}_1 + 2\hat{c}_2(\Theta + \hat{c}_3))\eta$ with $\Theta = (1-\theta)^2 + 2\theta^2$ and $k_\star$ is chosen from $\{0, 1, \dots, K\}$ according to probability $\mathcal{P}[k_\star = k] = \frac{\alpha_k}{\sum_{j=0}^K \alpha_j}$.

**Remark 5.** When $\alpha_0 \to 0$ the conditions in (8.2) reduces to $1 + \frac{2\rho}{\bar{\gamma}} > 0$ as in the deterministic case.

For $\theta = 0$ Algorithm 3 reduces to Algorithm 2. With this choice Theorem 8.2 simplifies, since the constant $\hat{c}_2 = 0$, and we recover the convergence result of Theorem 7.1.

## 9 EXPERIMENTS

We compare BC-SEG+ and BC-PSEG+ against (EG+) using *stochastic feedback* (which we refer to as (SF-EG+)) and (SEG) in both an unconstrained setting and a constrained setting introduced in Pethick et al. (2022). See Appendix H.2 for the precise formulation of the projected variants which we denote (SF-PEG+) and (PSEG) respectively. In the unconstrained example we control all problem constant and set $\rho = -1/10L_F$, while the constrained example is a specific minimax problem where $\rho > -1/2L_F$ holds within the constrained set for a Lipschitz constant $L_F$ restricted to the same constrained set. To simulate a stochastic setting in both examples, we consider additive Gaussian noise, i.e. $\hat{F}(z, \xi) = Fz + \xi$ where $\xi \sim \mathcal{N}(0, \sigma^2 I)$. In the experiments we choose $\sigma = 0.1$ and $\alpha_k \propto 1/k$, which ensures almost sure convergence of BC-(P)SEG+. For a more aggressive stepsize choice $\alpha_k \propto 1/\sqrt{k}$ see Figure 4. Further details can be found in Appendix H.

The results are shown in Figure 2. The sequence generated by (SEG) and (PSEG) diverges for the unconstrained problem and cycles in the constrained problem respectively. In comparison (SF-EG+) and (SF-PEG+) gets within a neighborhood of the solutions but fails to converge due to the non-diminishing stepsize, while BC-SEG+ and BC-PSEG+ converges in the examples.

## 10 CONCLUSION

This paper shows that nonconvex-nonconcave problems characterize by the weak Minty variational inequality can be solved efficiently even when only *stochastic* gradients are available. The approach crucially avoids increasing batch sizes by instead introducing a bias-correction term. We show that convergence is possible for the same range of problem constant $\rho \in (-\gamma/2, \infty)$ as in the deterministic case. Rates are established for a *random iterate*, which matches those of stochastic extragradient in the monotone case, and the result is complemented with almost sure convergence, thus providing asymptotic convergence for the *last iterate*. We show that the idea extends to a family of extragradient-type methods which includes a nonlinear extension of the celebrated primal dual hybrid gradient (PDHG) algorithm. For future work it is interesting to see if the rate can be improved by considering accelerated methods such as Halpern iterations.

## 11 ACKNOWLEDGMENTS AND DISCLOSURE OF FUNDING

This project has received funding from the European Research Council (ERC) under the European Union's Horizon 2020 research and innovation programme (grant agreement n° 725594 - time-data). This work was supported by the Swiss National Science Foundation (SNSF) under grant number 200021_205011. The work of the third and fourth author was supported by the Research Foundation Flanders (FWO) postdoctoral grant 12Y7622N and research projects G081222N, G033822N, G0A0920N; Research Council KU Leuven C1 project No. C14/18/068; European Union's Horizon 2020 research and innovation programme under the Marie Skłodowska-Curie grant agreement No. 953348. The work of Olivier Fercoq was supported by the Agence National de la Recherche grant ANR-20-CE40-0027, Optimal Primal-Dual Algorithms (APDO).

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

# Appendix

## TABLE OF CONTENTS

**Table 1:** *Overview of the results. The second row is obtained as special cases of the first row.*

| | Unconstrained & smooth ($A \equiv 0$) | | Constrained ($A \not\equiv 0$) | |
|---|---|---|---|---|
| | Random iterate | Last iterate | BC-PSEG+ | NP-PDHG |
| Appendix | Theorem D.2 | Theorem D.3 | Theorem E.2 | Theorem F.5 |
| | ⇓ | ⇓ | ⇓ | ⇓ |
| Main paper | Theorem 6.1 | Theorem 6.3 | Theorem 7.1 | Theorem 8.2 |

## A  PRELUDE

For the unconstrained and smooth setting Appendix C treats convergences of (SEG+) for the restricted case where $F$ is linear. Appendix D shows both random iterate results and almost sure convergence of Algorithm 1. Theorems 6.1 and 6.3 in the main body are implied by the more general results in this section, which preserves certain free parameters and more general stepsize requirements. Appendices E and F moves beyond the unconstrained and smooth case by showing convergence for instances of the template scheme (8.1). The analysis of Algorithm 3 in Appendix F applies to Algorithm 2, but for completeness we establish convergence for general $F$ separately in Appendix E. The relationship between the theorems are presented in Table 1.

## B  PRELIMINARIES

Given a psd matrix $V$ we define the inner product as $\langle \cdot, \cdot \rangle_V := \langle \cdot, V \cdot \rangle$ and the corresponding norm $\| \cdot \| := \sqrt{\langle \cdot, \cdot \rangle_V}$. The distance from $u \in \mathbb{R}^n$ to a set $\mathcal{U} \subseteq \mathbb{R}^n$ with respect to a positive definite matrix $V$ is defined as $\mathbf{dist}_V(u, \mathcal{U}) := \min_{u' \in \mathcal{U}} \|u - u'\|_V$, which we simply denote $\mathbf{dist}(u, \mathcal{U})$ when $V = I$. The norm $\|X\|$ refers to spectral norm when $X$ is a matrix.

We summarize essential definitions from operator theory, but otherwise refer to Bauschke & Combettes (2017); Rockafellar (1970) for further details.

An operator $A : \mathbb{R}^n \rightrightarrows \mathbb{R}^d$ maps each point $x \in \mathbb{R}^n$ to a subset $Ax \subseteq \mathbb{R}^d$, where the notation $A(x)$ and $Ax$ will be used interchangably. We denote the domain of $A$ by $\mathbf{dom}\, A := \{x \in \mathbb{R}^n \mid Ax \neq \emptyset\}$, its graph by $\mathbf{gph}\, A := \{(x, y) \in \mathbb{R}^n \times \mathbb{R}^d \mid y \in Ax\}$. The inverse of $A$ is defined through its graph, $\mathbf{gph}\, A^{-1} := \{(y, x) \mid (x, y) \in \mathbf{gph}\, A\}$ and the set of its zeros by $\mathbf{zer}\, A := \{x \in \mathbb{R}^n \mid 0 \in Ax\}$.

**Definition B.1** ((co)monotonicity Bauschke et al. (2021))**.** *An operator $A : \mathbb{R}^n \rightrightarrows \mathbb{R}^n$ is said to be $\rho$-monotone for some $\rho \in \mathbb{R}$, if for all $(x, y), (x', y') \in \mathbf{gph}\, A$*

$$\langle y - y', x - x' \rangle \geq \rho \|x - x'\|^2,$$

*and it is said to be $\rho$-comonotone if for all $(x, y), (x', y') \in \mathbf{gph}\, A$*

$$\langle y - y', x - x' \rangle \geq \rho \|y - y'\|^2.$$

*The operator $A$ is said to be maximally (co)monotone if there exists no other (co)monotone operator $B$ for which $\mathbf{gph}\, A \subset \mathbf{gph}\, B$ properly.*

If $A$ is 0-monotone we simply say it is monotone. When $\rho < 0$, $\rho$-comonotonicity is also referred to as $|\rho|$-cohypomonotonicity.

**Definition B.2** (Lipschitz continuity and cocoercivity)**.** *Let $\mathcal{D} \subseteq \mathbb{R}^n$ be a nonempty subset of $\mathbb{R}^n$. A single-valued operator $A : \mathcal{D} \to \mathbb{R}^n$ is said to be $L$-Lipschitz continuous if for any $x, x' \in \mathcal{D}$*

$$\|Ax - Ax'\| \leq L\|x - x'\|,$$

*and $\beta$-cocoercive if*

$$\langle x - x', Ax - Ax' \rangle \geq \beta \|Ax - Ax'\|^2.$$

*Moreover, $A$ is said to be nonexpansive if it is 1-Lipschitz continuous, and firmly nonexpansive if it is 1-cocoercive.*

A $\beta$-cocoercive operator is also $\beta^{-1}$-Lipschitz continuity by direct implication of Cauchy-Schwarz. The resolvent operator $J_A = (\mathrm{id} + A)^{-1}$ is firmly nonexpansive (with $\mathbf{dom}\, J_A = \mathbb{R}^n$) if and only if $A$ is (maximally) monotone.

We will make heavy use of the Fenchel-Young inequality. For all $a, b \in \mathbb{R}^n$ and $e > 0$ we have,

$$2\langle a, b\rangle \le e\|a\|^2 + \tfrac{1}{e}\|b\|^2 \tag{B.1}$$

$$\|a + b\|^2 \le (1 + e)\|a\|^2 + (1 + \tfrac{1}{e})\|b\|^2 \tag{B.2}$$

$$-\|a - b\|^2 \le -\tfrac{1}{1+e}\|a\|^2 + \tfrac{1}{e}\|b\|^2 \tag{B.3}$$

## C  PROOF FOR SEG+

***Proof of Theorem 5.1.*** Following (Hsieh et al., 2020) closely, define the reference state $\bar{\bar{z}}^k := z^k - \gamma F z^k$ to be the exploration step using the *deterministic* operator and denote the second stepsize as $\eta_k := \alpha_k \gamma$. We will let $\zeta$ denote the additive noise term, i.e. $\hat{F}(z, \xi) := F(z) + \zeta$. Expanding the distance to solution,

$$
\begin{aligned}
\|z^{k+1} - z^\star\|^2 &= \|z^k - \eta_k \hat{F}(\bar{z}^k, \bar{\xi}_k) - z^\star\|^2 \\
&= \|z^k - z^\star\|^2 - 2\eta_k\langle \hat{F}(\bar{z}^k, \bar{\xi}_k), z^k - z^\star\rangle + \eta_k^2\|\hat{F}(\bar{z}^k, \bar{\xi}_k)\|^2 \\
&= \|z^k - z^\star\|^2 - 2\eta_k\langle \hat{F}(\bar{z}^k, \bar{\xi}_k), \bar{z}^k - z^\star\rangle - 2\gamma\eta_k\langle \hat{F}(\bar{z}^k, \bar{\xi}_k), F(z^k)\rangle + \eta_k^2\|\hat{F}(\bar{z}^k, \bar{\xi}_k)\|^2.
\end{aligned} \tag{C.1}
$$

Recall that the operator is assumed to be linear $Fz = Bz + v$ in which case we have,

$$
\begin{aligned}
\hat{F}(\bar{z}^k, \bar{\xi}_k) &= B\bar{z}^k + v + \bar{\zeta}_k \\
&= B(z^k - \gamma\hat{F}(z^k, \xi_k)) + v + \bar{\zeta}_k \\
&= B(z^k - \gamma Bz^k - \gamma v - \gamma\zeta_k) + v + \bar{\zeta}_k \\
&= B(z^k - \gamma(Bz^k + v)) + v - \gamma B\zeta_k + \bar{\zeta}_k \\
&= F(\bar{\bar{z}}^k) - \gamma B\zeta_k + \bar{\zeta}_k.
\end{aligned} \tag{C.2}
$$

The two latter terms are zero in expectation due to the unbiasedness from Assumption II*(ii)*, which lets us write the terms on the RHS of (C.1) as,

$$-\mathbb{E}_k\langle \hat{F}(\bar{z}^k, \bar{\xi}_k), \bar{z}^k - z^\star\rangle = -\langle F(\bar{\bar{z}}^k), \bar{\bar{z}}^k - z^\star\rangle \tag{C.3}$$

$$-\mathbb{E}_k\langle \hat{F}(\bar{z}^k, \bar{\xi}_k), F(z^k)\rangle = -\langle F(\bar{\bar{z}}^k), F(z^k)\rangle \tag{C.4}$$

$$\mathbb{E}_k\|\hat{F}(\bar{z}^k, \bar{\xi}_k)\|^2 = \|F(\bar{\bar{z}}^k)\|^2 + \mathbb{E}_k\|\gamma B\zeta_k\|^2 + \mathbb{E}_k\|\bar{\zeta}_k\|^2. \tag{C.5}$$

We can bound (C.3) directly through the weak MVI in Assumption I*(iii)* which might still be positive,

$$-\langle F(\bar{\bar{z}}^k), \bar{\bar{z}}^k - z^\star\rangle \le -\rho\|F(\bar{\bar{z}}^k)\|^2. \tag{C.6}$$

For the latter two terms of (C.5) we have

$$\mathbb{E}_k\|\gamma B\zeta_k\|^2 + \mathbb{E}_k\|\bar{\zeta}_k\|^2 = \gamma^2\mathbb{E}_k\|F(\zeta_k) - F(0)\|^2 + \mathbb{E}_k\|\bar{\zeta}_k\|^2 \le (\gamma^2 L_F^2 + 1)\sigma_F^2, \tag{C.7}$$

where the last inequality follows from Lipschitz in Assumption I*(i)* and bounded variance in Assumption II*(iii)*.

Combining everything into (C.1) we are left with

$$\mathbb{E}_k\|z^{k+1} - z^\star\|^2 \le \|z^k - z^\star\|^2 + \eta_k^2(\gamma^2 L_F^2 + 1)\sigma_F^2 - 2\gamma\eta_k\langle F(\bar{\bar{z}}^k), F(z^k)\rangle + (\eta_k^2 - 2\eta_k\rho)\|F(\bar{\bar{z}}^k)\|^2 \tag{C.8}$$

By assuming the stepsize condition, $\rho \ge (\eta_k - \gamma)/2$, we have $\eta_k^2 - 2\eta_k\rho \le \gamma\eta_k$. This allows us to complete the square,

$$
\begin{aligned}
-2\gamma\eta_k\langle F(\bar{\bar{z}}^k), F(z^k)\rangle + (\eta_k^2 - 2\eta_k\rho)\|F(\bar{\bar{z}}^k)\|^2 &\le -2\gamma\eta_k\langle F(\bar{\bar{z}}^k), F(z^k)\rangle + \gamma\eta_k\|F(\bar{\bar{z}}^k)\|^2 \\
&= \gamma\eta_k(\|F(z^k) - F(\bar{\bar{z}}^k)\|^2 - \|F(z^k)\|^2) \\
&\le \gamma\eta_k(\gamma^2 L_F^2 - 1)\|F(z^k)\|^2,
\end{aligned} \tag{C.9}
$$

where the last inequality follows from Lipschitzness of $F$ and the definition of the update rule. Plugging into (C.8) we are left with

$$\mathbb{E}_k\|z^{k+1} - z^\star\|^2 \le \|z^k - z^\star\|^2 + \eta_k^2(\gamma^2 L_F^2 + 1)\sigma_F^2 - \gamma\eta_k(1 - \gamma^2 L_F^2)\|F(z^k)\|^2. \tag{C.10}$$

The result is obtained by total expectation and summing. □

# D  PROOF FOR SMOOTH UNCONSTRAINED CASE

**Lemma D.1.** *Consider the recurrent relation $B_{k+1} = \xi_k B_k + d_k$ such that $\xi_k > 0$ for all $k \geq 0$. Then*

$$B_{k+1} = (\Pi_{p=0}^k \xi_p)\left( B_0 + \sum_{\ell=0}^k \frac{d_\ell}{\Pi_{p=0}^\ell \xi_p} \right).$$

**Assumption V.** *$\gamma \in (\lfloor -2\rho \rfloor_+, 1/L_F)$ and for positive real valued $b$,*

$$\mu := \gamma^2 (1 - \gamma^2 L_F^2 (1 + b^{-1})) > 0. \tag{D.1}$$

**Theorem D.2.** *Suppose that Assumptions I to III hold. Suppose in addition that Assumption V holds and that $(\alpha_k)_{k \in \mathbb{N}} \subset (0, 1)$ is a diminishing sequence such that*

$$2\gamma L_{\hat{F}} \sqrt{\alpha_0} + \left( 1 + ((b+1)\gamma^2 L_F^2)\gamma^2 L_{\hat{F}}^2 \right)\alpha_0 \leq 1 + \frac{2\rho}{\gamma}. \tag{D.2}$$

*Consider the sequence $(z^k)_{k \in \mathbb{N}}$ generated by Algorithm 1. Then, the following estimate holds*

$$\sum_{k=0}^K \frac{\alpha_k}{\sum_{j=0}^K \alpha_j} \mathbb{E}[\|F(z^k)\|^2] \leq \frac{\|z^0 - z^\star\|^2 + \eta\gamma^2 \|F(z^0)\|^2 + C\sigma_F^2 \gamma^2 \sum_{j=0}^K \alpha_j^2}{\mu \sum_{j=0}^K \alpha_j}, \tag{D.3}$$

*where $C = 1 + 2\eta((\gamma^2 L_{\hat{F}}^2 + 1) + 2\alpha_0)$ and $\eta = \frac{1}{2}(b+1)\gamma^2 L_F^2 + \frac{1}{\gamma L_{\hat{F}} \sqrt{\alpha_0}}$.*

*Proof of Theorem D.2.* The proof relies on establishing a (stochastic) descent property on the following potential function

$$\mathcal{U}_{k+1} := \|z^{k+1} - z^\star\|^2 + A_{k+1}\|u^k\|^2 + B_{k+1}\|z^{k+1} - z^k\|^2.$$

where $u^k := \bar{z}^k - z^k + \gamma F(z^k)$ measures the difference of the bias-corrected step from the deterministic exploration step, and $(A_k)_{k \in \mathbb{N}}, (B_k)_{k \in \mathbb{N}}$ are positive scalar parameters to be identified. We proceed to consider each term individually.

Let us begin by quantifying how well $\bar{z}^k$ estimates $z^k - \gamma F(z^k)$.

$$u^k = \bar{z}^k - z^k + \gamma F(z^k) = \gamma F(z^k) - \gamma \hat{F}(z^k, \xi_k) + (1 - \alpha_k)(\bar{z}^{k-1} - z^{k-1} + \gamma \hat{F}(z^{k-1}, \xi_k)).$$

Therefore,

$$\|u^k\|^2 = \|\gamma F(z^k) - \gamma \hat{F}(z^k, \xi_k) + (1 - \alpha_k)(\gamma \hat{F}(z^{k-1}, \xi_k) - \gamma F(z^{k-1}))\|^2 + (1 - \alpha_k)^2\|u^{k-1}\|^2$$
$$+ 2(1 - \alpha_k)\langle \bar{z}^{k-1} - z^{k-1} + \gamma F(z^{k-1}), \gamma F(z^k) - \gamma \hat{F}(z^k, \xi_k) + (1 - \alpha_k)(\gamma \hat{F}(z^{k-1}, \xi_k) - \gamma F(z^{k-1}))\rangle.$$

Conditioned on $\mathcal{F}_k$, in the inner product the left term is known and the right term has an expectation that equals zero. Therefore, we obtain

$$\mathbb{E}[\|u^k\|^2 | \mathcal{F}_k] = \mathbb{E}[\|(1 - \alpha_k)(\gamma F(z^k) - \gamma \hat{F}(z^k, \xi_k) + \gamma \hat{F}(z^{k-1}, \xi_k) - \gamma F(z^{k-1})) + \alpha_k(\gamma F(z^k) - \gamma \hat{F}(z^k, \xi_k))\|^2 | \mathcal{F}_k]$$
$$+ (1 - \alpha_k)^2\|u^{k-1}\|^2$$
$$\leq (1 - \alpha_k)^2\|u^{k-1}\|^2 + 2(1 - \alpha_k)^2\gamma^2 \mathbb{E}[\|\hat{F}(z^k, \xi_k) - \hat{F}(z^{k-1}, \xi_k)\|^2 | \mathcal{F}_k]$$
$$+ 2\alpha_k^2\gamma^2 \mathbb{E}[\|F(z^k) - \hat{F}(z^k, \xi_k)\|^2 | \mathcal{F}_k]$$
$$\leq (1 - \alpha_k)^2\|u^{k-1}\|^2 + 2(1 - \alpha_k)^2\gamma^2 L_{\hat{F}}^2 \|z^k - z^{k-1}\|^2 + 2\alpha_k^2\gamma^2 \sigma_F^2 \tag{D.4}$$

where in the first inequality we used Young inequality and the fact that the second moment is larger than the variance, and Assumptions II(iii) and III were used in the second inequality.

By step 1.4, the equality

$$\|z^{k+1} - z^\star\|^2 = \|z^k - z^\star\|^2 - 2\alpha_k \gamma \langle \hat{F}(\bar{z}^k, \bar{\xi}_k), z^k - z^\star \rangle + \alpha_k^2 \gamma^2 \|\hat{F}(\bar{z}^k, \bar{\xi}_k)\|^2, \tag{D.5}$$

holds. The inner product in (D.5) can be upper bounded using Young inequalities with positive parameters $\varepsilon_k$, $k \geq 0$, and $b$ as follows.

$$
\begin{aligned}
\mathbb{E}[\langle -\gamma \hat{F}(\bar{z}^k, \bar{\xi}_k), z^k - z^\star \rangle \mid \bar{\mathcal{F}}_k] = {}& -\gamma \langle F(\bar{z}^k), z^k - \bar{z}^k \rangle - \gamma \langle F(\bar{z}^k), \bar{z}^k - z^\star \rangle \\
= {}& -\gamma^2 \langle F(\bar{z}^k), F(z^k) \rangle + \gamma \langle F(\bar{z}^k), \bar{z}^k - z^k + \gamma F(z^k) \rangle - \gamma \langle F(\bar{z}^k), \bar{z}^k - z^\star \rangle \\
\leq {}& \gamma^2 \Big( \frac{1}{2} \| F(\bar{z}^k) - F(z^k) \|^2 - \frac{1}{2} \| F(\bar{z}^k) \|^2 - \frac{1}{2} \| F(z^k) \|^2 \Big) + \frac{\gamma^2 \varepsilon_k}{2} \| F(\bar{z}^k) \|^2 \\
& + \frac{1}{2\varepsilon_k} \| \bar{z}^k - z^k + \gamma F(z^k) \|^2 - \gamma \rho \| F(\bar{z}^k) \|^2 \\
\leq {}& \gamma^2 L_F^2 \frac{1+b}{2} \| u^k \|^2 + \frac{1 + b^{-1}}{2} \gamma^4 L_F^2 \| F(z^k) \|^2 - \frac{\gamma^2}{2} \| F(\bar{z}^k) \|^2 \\
& - \frac{\gamma^2}{2} \| F(z^k) \|^2 + \frac{\gamma^2 \varepsilon_k}{2} \| F(\bar{z}^k) \|^2 + \frac{1}{2\varepsilon_k} \| u^k \|^2 - \gamma \rho \| F(\bar{z}^k) \|^2 \\
= {}& (\gamma^2 L_F^2 \frac{1+b}{2} + \frac{1}{2\varepsilon_k}) \| u^k \|^2 + \frac{\gamma^2 (\gamma^2 L_F^2 (1 + b^{-1}) - 1)}{2} \| F(z^k) \|^2 \\
& + (\frac{\gamma^2 (\varepsilon_k - 1)}{2} - \gamma \rho) \| F(\bar{z}^k) \|^2.
\end{aligned}
\tag{D.6}
$$

Conditioning (D.6) with $\mathbb{E}[\cdot \mid \mathcal{F}_k] = \mathbb{E}[\mathbb{E}[\cdot \mid \bar{\mathcal{F}}_k] \mid \mathcal{F}_k]$, since $\mathcal{F}_k \subset \bar{\mathcal{F}}_k$, yields

$$
\begin{aligned}
2\mathbb{E}[\langle -\gamma \hat{F}(\bar{z}^k, \bar{\xi}_k), z^k - z^\star \rangle \mid \mathcal{F}_k] \leq {}& (\gamma^2 L_F^2 (1 + b) + \frac{1}{\varepsilon_k}) \mathbb{E}[\| u^k \|^2 \mid \mathcal{F}_k] - \mu \| F(z^k) \|^2 \\
& + (\gamma^2 (\varepsilon_k - 1) - 2\gamma \rho) \mathbb{E}[\| F(\bar{z}^k) \|^2 \mid \mathcal{F}_k],
\end{aligned}
\tag{D.7}
$$

where $\mu$ was defined in (D.1).

The condition expectation of the third term in (D.5) is bounded through Assumption II*(iii)* by

$$
\mathbb{E}[\| \hat{F}(\bar{z}^k, \bar{\xi}_k) \|^2 \mid \mathcal{F}_k] = \mathbb{E}[\mathbb{E}[\| \hat{F}(\bar{z}^k, \bar{\xi}_k) \|^2 \mid \bar{\mathcal{F}}_k] \mid \mathcal{F}_k] \leq \| F(\bar{z}^k) \|^2 + \sigma_F^2,
$$

which in turn implies

$$
\mathbb{E}[\| z^{k+1} - z^k \|^2 \mid \mathcal{F}_k] = \alpha_k^2 \gamma^2 \mathbb{E}[\| \hat{F}(\bar{z}^k, \bar{\xi}_k) \|^2 \mid \mathcal{F}_k] \leq \alpha_k^2 \gamma^2 \mathbb{E}[\| F\bar{z}^k \|^2 \mid \mathcal{F}_k] + \alpha_k^2 \gamma^2 \sigma_F^2
\tag{D.8}
$$

Combining (D.7), (D.8), and (D.5) yields

$$
\begin{aligned}
& \mathbb{E}[\| z^{k+1} - z^\star \|^2 + A_{k+1} \| u^k \|^2 + B_{k+1} \| z^{k+1} - z^k \|^2 \mid \mathcal{F}_k] \\
& \leq \| z^k - z^\star \|^2 + \Big( A_{k+1} + \alpha_k (\gamma^2 L_F^2 (1 + b) + \frac{1}{\varepsilon_k}) \Big) \mathbb{E}[\| u^k \|^2 \mid \mathcal{F}_k] - \alpha_k \mu \| F(z^k) \|^2 \\
& \quad + \Big( \alpha_k (\gamma^2 (\varepsilon_k - 1) - 2\gamma \rho) + \alpha_k^2 \gamma^2 \Big) \mathbb{E}[\| F(\bar{z}^k) \|^2 \mid \mathcal{F}_k] + \alpha_k^2 \gamma^2 \sigma_F^2 \\
& \quad + B_{k+1} \alpha_k^2 \gamma^2 \mathbb{E}[\| F\bar{z}^k \|^2 \mid \mathcal{F}_k] + B_{k+1} \alpha_k^2 \gamma^2 \sigma_F^2.
\end{aligned}
\tag{D.9}
$$

Further using (D.4) and denoting

$$
\begin{aligned}
X_1^k &:= \alpha_k \Big( \gamma^2 L_F^2 (1 + b) + \tfrac{1}{\varepsilon_k} \Big) + A_{k+1}, \\
X_2^k &:= \alpha_k \Big( \gamma^2 (\varepsilon_k - 1) - 2\rho\gamma + \alpha_k \gamma^2 \Big)
\end{aligned}
$$

leads to

$$
\begin{aligned}
\mathbb{E}[\mathcal{U}_{k+1} \mid \mathcal{F}_k] - \mathcal{U}_k \leq {}& -\alpha_k \mu \| F(z^k) \|^2 + \Big( X_1^k (1 - \alpha_k)^2 - A_k \Big) \| u^{k-1} \|^2 \\
& + \Big( 2X_1^k (1 - \alpha_k)^2 \gamma^2 L_{\hat{F}}^2 - B_k \Big) \| z^k - z^{k-1} \|^2 + \Big( X_2^k + B_{k+1} \alpha_k^2 \gamma^2 \Big) \mathbb{E}[\| F(\bar{z}^k) \|^2 \mid \mathcal{F}_k] \\
& + \Big( B_{k+1} \alpha_k^2 + \alpha_k^2 + 2X_1^k \alpha_k^2 \Big) \gamma^2 \sigma_F^2.
\end{aligned}
\tag{D.10}
$$

Having established (D.10), set $A_k = A$, $B_k = 2A\gamma^2 L_{\hat{F}}^2$, and $\varepsilon_k = \varepsilon$ to obtain by the law of total expectation that

$$
\begin{aligned}
\mathbb{E}[\mathcal{U}_{k+1}] - \mathbb{E}[\mathcal{U}_k] \leq {}& -\alpha_k \mu \mathbb{E}[\| F(z^k) \|^2] + \Big( X_1^k (1 - \alpha_k)^2 - A \Big) \mathbb{E}[\| u^{k-1} \|^2] \\
& + 2\gamma^2 L_{\hat{F}}^2 \Big( X_1^k (1 - \alpha_k)^2 - A \Big) \mathbb{E}[\| z^k - z^{k-1} \|^2] + \Big( X_2^k + 2A\gamma^4 L_{\hat{F}}^2 \alpha_k^2 \Big) \mathbb{E}[\| F(\bar{z}^k) \|^2] \\
& + \Big( 2A\gamma^2 L_{\hat{F}}^2 + 1 + 2X_1^k \Big) \alpha_k^2 \gamma^2 \sigma_F^2.
\end{aligned}
\tag{D.11}
$$

To get a recursion we require

$$X_1^k(1 - \alpha_k)^2 - A \leq 0 \quad \text{and} \quad X_2^k + 2A\gamma^4 L_{\hat{F}}^2 \alpha_k^2 \leq 0. \tag{D.12}$$

By developing the first requirement of (D.12) we have,

$$0 \geq X_1^k(1 - \alpha_k)^2 - A = \alpha_k(1 - \alpha_k)^2\left(\gamma^2 L_F^2(1 + b) + \tfrac{1}{\varepsilon}\right) + \alpha_k(\alpha_k - 2)A. \tag{D.13}$$

Equivalently, $A$ needs to satisfy

$$A \geq \frac{(1 - \alpha_k)^2}{2 - \alpha_k}\left(\gamma^2 L_F^2(1 + b) + \tfrac{1}{\varepsilon}\right). \tag{D.14}$$

for any $\alpha_k \in (0, 1)$. Since $\frac{(1-\alpha_k)^2}{2-\alpha_k} \leq \tfrac{1}{2}$ given $\alpha_k \in (0, 1)$ it suffice to pick

$$A = \tfrac{1}{2}\left((b + 1)\gamma^2 L_F^2 + \tfrac{1}{\varepsilon}\right). \tag{D.15}$$

For the second requirement of (D.12) note that we can equivalently require that the following quantity is negative

$$\frac{1}{\alpha_k \gamma^2}\left(X_2^k + 2A\gamma^4 L_{\hat{F}}^2 \alpha_k^2\right) = \varepsilon - 1 - \tfrac{2\rho}{\gamma} + \alpha_k + 2A\gamma^2 L_{\hat{F}}^2 \alpha_k$$

$$\leq \varepsilon - 1 - \tfrac{2\rho}{\gamma} + \left(1 + \left((b + 1)\gamma^2 L_F^2 + \tfrac{1}{\varepsilon}\right)\gamma^2 L_{\hat{F}}^2\right)\alpha_0$$

where we have used that $\alpha_k \leq \alpha_0$ and the choice of $A$ from (D.15). Setting the Young parameter $\varepsilon = \gamma L_{\hat{F}} \sqrt{\alpha_0}$ we obtain that $X_2^k + 2A\gamma^4 L_{\hat{F}}^2 \alpha_k^2 \leq 0$ owing to (D.2).

On the other hand, the last term in (D.11) may be upper bounded by

$$2A\gamma^2 L_{\hat{F}}^2 + 1 + 2X_1^k = 1 + \left((b + 1)\gamma^2 L_F^2 + \tfrac{1}{\gamma L_{\hat{F}} \sqrt{\alpha_0}}\right)\left((\gamma^2 L_{\hat{F}}^2 + 1) + 2\alpha_k\right)$$

$$\leq 1 + \left((b + 1)\gamma^2 L_F^2 + \tfrac{1}{\gamma L_{\hat{F}} \sqrt{\alpha_0}}\right)\left((\gamma^2 L_{\hat{F}}^2 + 1) + 2\alpha_0\right) = C.$$

Thus, it follows from (D.11) that

$$\mathbb{E}[\mathcal{U}_{k+1}] - \mathbb{E}[\mathcal{U}_k] \leq -\alpha_k \mu \mathbb{E}\left[\|F(z^k)\|^2\right] + C\alpha_k^2 \gamma^2 \sigma_F^2.$$

Telescoping the above inequality completes the proof. $\qquad \square$

***Proof of Theorem 6.1.*** The theorem is obtained as a particular instantiation of Theorem D.2.

The condition in (D.1) can be rewritten as $b > \frac{\gamma^2 L_F^2}{1 - \gamma^2 L_F^2}$. A reasonable choice is $b = \frac{2\gamma^2 L_F^2}{1 - \gamma^2 L_F^2}$. Substituting back into $\mu$ we obtain

$$\mu = \gamma^2(1 - \gamma^2 L_F^2(1 + \tfrac{1 - \gamma^2 L_F^2}{2\gamma^2 L_F^2})) = \tfrac{\gamma^2(1 - \gamma^2 L_F^2)}{2} > 0. \tag{D.16}$$

Similarly, the choice of $b$ is substituted into $\eta$ and (D.2) of Theorem D.2.

The rate in (D.2) is further simplified by applying Lipschitz continuity of $F$ from Assumption I*(i)* to $\|Fz^0\|^2 = \|Fz^0 - Fz^\star\|^2$. The proof is complete by observing that the guarantee on the weighted sum can be converted into an expectation over a sampled iterate in the style of Ghadimi & Lan (2013). $\qquad \square$

**Assumption VI** (almost sure convergence). *Let $d \in [0, 1]$, $b > 0$. Suppose that the following holds*

*(i) the diminishing sequence $(\alpha_k)_{k \in \mathbb{N}} \subset (0, 1)$ satisfies the classical conditions*

$$\textstyle\sum_{k=0}^\infty \alpha_k = \infty, \qquad \bar{\alpha} := \sum_{k=0}^\infty \alpha_k^2 < \infty;$$

*(ii) letting $c_k := (1 + b)\gamma^2 L_F^2 + \tfrac{1}{\gamma L_{\hat{F}}}\alpha_k^{-d}$ for all $k \geq 0$*

$$\eta_k := \textstyle\sum_{\ell=k}^\infty \left(c_l \alpha_l \Pi_{p=0}^\ell (1 - \alpha_p)^2\right) < \infty, \qquad \nu := \sum_{k=0}^\infty \eta_{k+1}\alpha_k^2\left(\Pi_{p=0}^k \tfrac{1}{(1-\alpha_p)^2}\right) < \infty, \tag{D.17}$$

*and*

$$\gamma L_{\hat{F}}\alpha_k^d + \alpha_k + 2\gamma^2 L_{\hat{F}}^2 \alpha_k \eta_{k+1}\Pi_{p=0}^k \tfrac{1}{(1-\alpha_p)^2} \leq 1 + \tfrac{2\rho}{\gamma}. \tag{D.18}$$

Although at first look the above assumptions may appear involved, as shown in Theorem D.3 classical stepsize choice of $\frac{\alpha_0}{k+1}$ is sufficient to satisfy (D.17), and to ensure almost sure convergence provided that instead (D.20) holds. Note that with this choice as $k$ goes to infinity, $\alpha_k \searrow 0$ and the deterministic range $\gamma + 2\rho > 0$ is obtained.

**Theorem D.3** (almost sure convergence). *Suppose that Assumptions I to III hold. Additionally, suppose the stepsize conditions in Assumptions V and VI. Then, the sequence $(z^k)_{k \in \mathbb{N}}$ generated by Algorithm 1 converges almost surely to some $z^\star \in \mathbf{zer}\, T$. Moreover, the following estimate holds*

$$\sum_{k=0}^{K} \frac{\alpha_k}{\sum_{j=0}^{K} \alpha_j} \mathbb{E}[\|F(z^k)\|^2] \le \frac{\|z^0 - z^\star\|^2 + \eta_0 \gamma^2 \|F(z^0)\|^2 + \bar{C}}{\mu \sum_{j=0}^{K} \alpha_j}, \tag{D.19}$$

*where $\bar{C} = 2\gamma^2 \sigma_F^2 ((\gamma^2 L_{\hat{F}}^2 + 1)\nu + \bar{\alpha}(\frac{1}{2} + (b+1)\gamma^2 L_F^2 + \frac{1}{\gamma L_{\hat{F}}}))$ is finite.*

*In particular, if $\alpha_k = \frac{1}{k+r}$ for any positive natural number $r$, and $d = 1$, then Assumption VI(ii) can be replaced by*

$$(\gamma L_{\hat{F}} + 1)\alpha_k + 2\big((1+b)\gamma^4 L_F^2 L_{\hat{F}}^2 \alpha_{k+1} + \gamma L_{\hat{F}}\big)(\alpha_{k+1} + 1)\alpha_{k+1} \le 1 + \frac{2\rho}{\gamma}. \tag{D.20}$$

***Proof of Theorem D.3** (almost sure convergence).* Having established (D.10), let $B_k = 2A_k \gamma^2 L_{\hat{F}}^2$ such that

$$\big(2X_1^k (1-\alpha_k)^2 \gamma^2 L_{\hat{F}}^2 - B_k\big)\|z^k - z^{k-1}\|^2 = 2\gamma^2 L_{\hat{F}}^2 \big(X_1^k (1-\alpha_k)^2 - A_k\big)\|z^k - z^{k-1}\|^2. \tag{D.21}$$

In what follows we show that it is sufficient to ensure

$$X_1^k (1-\alpha_k)^2 \le A_k, \quad X_2^k + 2A_{k+1}\gamma^4 L_{\hat{F}}^2 \alpha_k^2 \le 0, \tag{D.22}$$

resulting in the inequality

$$\mathbb{E}[\mathcal{U}_{k+1} \mid \mathcal{F}_k] - \mathcal{U}_k \le -\alpha_k \mu \|F(z^k)\|^2 + \big(2A_{k+1}\gamma^2 L_{\hat{F}}^2 + 1 + 2X_1^k\big)\alpha_k^2 \gamma^2 \sigma_F^2. \tag{D.23}$$

A reasonable choice for the Young parameter $\varepsilon_k$ is to choose

$$\varepsilon_k = \gamma L_{\hat{F}} \alpha_k^d \quad \text{for some} \ d \in [0,1]. \tag{D.24}$$

The rational for this choice will become more clear in what follows.

The first inequality in (D.22) is linear and we can solve it to equality by Lemma D.1. Let

$$A_0 := \sum_{\ell=0}^{\infty} \big(c_l \alpha_l \Pi_{p=0}^{\ell}(1-\alpha_p)^2\big) \overset{\text{(D.17)}}{=} \eta_0 < \infty, \quad \text{and} \quad \nu = \sum_{k=0}^{\infty} A_{k+1}\alpha_k^2 \overset{\text{(D.17)}}{<} \infty. \tag{D.25}$$

Furthermore, let $c_k$ and $\eta_k$ be as in Assumption VI(ii). Then, Lemma D.1 yields

$$A_{k+1} = \left(\Pi_{p=0}^{k} \frac{1}{(1-\alpha_p)^2}\right)\left(A_0 - \sum_{\ell=0}^{k}\big(c_l \alpha_l \Pi_{p=0}^{\ell}(1-\alpha_p)^2\big)\right) = \eta_{k+1}\Pi_{p=0}^{k}\frac{1}{(1-\alpha_p)^2} \tag{D.26}$$

which would ensure $A_k \ge 0$ for all $k$. Therefore, assumptions (D.17) and (D.18) (which is a restatement of the conditions in (D.22)) are sufficient for ensuring (D.23). Substituting $X_1^k$ and $A_{k+1}$ in (D.23) yields

$$\mathbb{E}[\mathcal{U}_{k+1} \mid \mathcal{F}_k] - \mathcal{U}_k \le -\alpha_k \mu \|F(z^k)\|^2 + \xi_k, \tag{D.27}$$

where $\xi_k = 2\big(A_{k+1}(\gamma^2 L_{\hat{F}}^2 + 1) + \frac{1}{2} + (b+1)\gamma^2 L_F^2 \alpha_k + \frac{1}{\gamma L_{\hat{F}}}\alpha_k^{1-d}\big)\alpha_k^2 \gamma^2 \sigma_F^2$. By Assumption VI we have that

$$\sum_{k=0}^{\infty} \xi_k = 2\gamma^2 \sigma_F^2 \left((\gamma^2 L_{\hat{F}}^2 + 1)\sum_{k=0}^{\infty} A_{k+1}\alpha_k^2 + \sum_{k=0}^{\infty} \frac{\alpha_k^2}{2} + (b+1)\gamma^2 L_F^2 \sum_{k=0}^{\infty} \alpha_k^3 + \frac{1}{\gamma L_{\hat{F}}}\sum_{k=0}^{\infty} \alpha_k^{3-d}\right)$$

$$\le 2\gamma^2 \sigma_F^2 \left((\gamma^2 L_{\hat{F}}^2 + 1)\sum_{k=0}^{\infty} A_{k+1}\alpha_k^2 + \big(\frac{1}{2} + (b+1)\gamma^2 L_F^2 + \frac{1}{\gamma L_{\hat{F}}}\big)\sum_{k=0}^{\infty} \alpha_k^2\right) < \infty$$

where we used the fact that $\alpha_k^3 \le \alpha_k^2$ and $d \le 1$ in the first inequality, while the second inequality uses (D.25), and Assumption VI(i). The claimed convergence result follows by the Robbins-Siegmund supermartingale theorem (Bertsekas, 2011, Prop. 2) and standard arguments as in (Bertsekas, 2011, Prop. 9).

The claimed rate follows by taking total expectation and summing the above inequality over $k$ and noting that initial iterates were set as $\bar{z}^{-1} = z^{-1} = z^0$.

To provide an instance of the sequence $(\alpha_k)_{k\in\mathbb{N}}$ that satisfy the assumptions, let $r$ denote a positive natural number and set

$$\alpha_k = \tfrac{1}{k+r}. \tag{D.28}$$

Then,

$$\Pi_{p=0}^\ell (1-\alpha_p)^2 = \Pi_{p=0}^\ell (\tfrac{p+r-1}{p+r})^2 = \tfrac{(r-1)^2}{(\ell+r)^2} = (r-1)^2\alpha_\ell^2,$$

and for any $K \geq 0$

$$\sum_{\ell=0}^K \left(c_\ell\alpha_\ell\Pi_{p=0}^\ell(1-\alpha_p)^2\right) = \sum_{\ell=0}^K \tfrac{(r-1)^2}{(\ell+r)^3}c_\ell.$$

Plugging the value of $c_\ell$ and $\varepsilon_k$ from Assumption VI*(ii)* and (D.24) we obtain that $A_0$ is finite valued since $\sum_{\ell=0}^\infty \tfrac{1}{(\ell+r)^3\varepsilon_\ell} = \sum_{\ell=0}^\infty \tfrac{1}{(\ell+r)^{3-d}} < \infty$ owing to the fact that $d \leq 1$.

Moreover,

$$A_{k+1} = \frac{(k+r)^2}{(r-1)^2}\left(A_0 - \sum_{\ell=0}^k \left(\tfrac{(r-1)^2}{(\ell+r)^3}c_\ell\right)\right) = (k+r)^2 \sum_{\ell=k+1}^\infty \tfrac{1}{(\ell+r)^3}c_\ell = \tfrac{1}{\alpha_k^2} \sum_{\ell=k+1}^\infty \alpha_\ell^3 c_\ell \tag{D.29}$$

On the other hand, for $e > 1$ we have the following bound

$$\sum_{\ell=k+1}^\infty \alpha_\ell^e \leq \tfrac{1}{(k+1+r)^e} + \int_{k+1}^\infty \tfrac{1}{(x+r)^e}dx = \tfrac{1}{(k+1+r)^e} + \tfrac{1}{(e-1)(k+1+r)^{e-1}}. \tag{D.30}$$

Therefore, it follows from (D.29) that

$$A_{k+1}\alpha_k = \tfrac{1}{\alpha_k} \sum_{\ell=k+1}^\infty \left(\alpha_\ell^3(1+b)\gamma^2 L_F^2 + \tfrac{1}{\gamma L_F}\alpha_\ell^{3-d}\right)$$

$$\stackrel{\text{(D.30)}}{\leq} \left((1+b)\gamma^2 L_F^2 \tfrac{1}{2(k+1+r)}\right)\left(\tfrac{2}{k+1+r} + 1\right)\tfrac{1}{k+1+r} + \left(\tfrac{1}{\gamma L_F} \tfrac{1}{(2-d)(k+1+r)^{1-d}}\right)\left(\tfrac{1}{k+1+r} + 1\right)\tfrac{1}{k+1+r}$$

$$= \left(\tfrac{1+b}{2}\gamma^2 L_F^2\alpha_{k+1}\right)(2\alpha_{k+1} + 1)\alpha_{k+1} + \left(\tfrac{1}{\gamma L_F(2-d)}\alpha_{k+1}^{1-d}\right)(\alpha_{k+1} + 1)\alpha_{k+1}$$

$$\leq \left((1+b)\gamma^2 L_F^2\alpha_{k+1} + \tfrac{1}{\gamma L_F(2-d)}\alpha_{k+1}^{1-d}\right)(\alpha_{k+1} + 1)\alpha_{k+1} \tag{D.31}$$

In turn, this inequality ensures that $\nu$ as defined in Assumption VI*(ii)* is finite. To see this note that

$$\nu = \sum_{k=0}^\infty A_{k+1}\alpha_k^2 \stackrel{\text{(D.31)}}{\leq} \sum_{k=0}^\infty \left((1+b)\gamma^2 L_F^2\alpha_{k+1} + \tfrac{1}{\gamma L_F(2-d)}\alpha_{k+1}^{1-d}\right)(\alpha_{k+1} + 1)\alpha_{k+1}\alpha_k \leq \delta \sum_{k=0}^\infty \alpha_k^2 < \infty,$$

where in the last two inequalities Assumption VI*(i)* was used.

It remains to confirm the second inequality in (D.22). With the choice of $\alpha_k$ and $\varepsilon_k$ as in (D.28) and (D.24) we have

$$\tfrac{1}{\alpha_k\gamma^2}(X_2 + 2A_{k+1}\gamma^4 L_{\hat F}^2\alpha_k^2)$$

$$= \gamma L_{\hat F}\alpha_k^d - 1 - \tfrac{2\rho}{\gamma} + \alpha_k + 2A_{k+1}\gamma^2 L_{\hat F}^2\alpha_k$$

$$\stackrel{\text{(D.31)}}{\leq} \gamma L_{\hat F}\alpha_k^d + \alpha_k + 2\gamma^2 L_{\hat F}^2\left((1+b)\gamma^2 L_F^2\alpha_{k+1} + \tfrac{1}{\gamma L_{\hat F}(2-d)}\alpha_{k+1}^{1-d}\right)(\alpha_{k+1} + 1)\alpha_{k+1} - 1 - \tfrac{2\rho}{\gamma}.$$

It follows that with $d = 1$ the assumption (D.20) is sufficient to ensure that the second condition in (D.22) holds. □

**Proof of Theorem 6.3** (*almost sure convergence*). The result is a restatement of the special case in Theorem D.3 where $\alpha_k = \tfrac{1}{k+r}$. We proceed similarly to the proof of Theorem 6.1.

The condition in (D.1) can be rewritten as $b > \tfrac{\gamma^2 L_F^2}{1-\gamma^2 L_F^2}$. A reasonable choice is $b = \tfrac{2\gamma^2 L_F^2}{1-\gamma^2 L_F^2}$. The choice of $b$ is substituted into (D.1), (D.20) and $\bar C$ of Theorem D.3. This completes the proof.

□

# E    PROOF FOR CONSTRAINED CASE

We will rely on two well-known and useful properties of the *deterministic* operator $H = \text{id} - \gamma F$ from (Pethick et al., 2022, Lm. A.3) that we restate here for convenience.

**Lemma E.1.** *Let $F : \mathbb{R}^n \to \mathbb{R}^n$ be a $L_F$-Lipschitz operator and $H = \text{id} - \gamma F$ with $\gamma \in (0, 1/L_F]$. Then,*

  *(i) The operator $H$ is $1/2$-cocoercive.*

  *(ii) The operator $H$ is $(1 - \gamma L_F)$-monotone, and in particular*
$$\|Hz' - Hz\| \geq (1 - \gamma L_F)\|z' - z\| \quad \forall z, z' \in \mathbb{R}^n. \tag{E.1}$$

*Proof.* The first claim follows from direct computation
$$\begin{aligned}
\langle Hz - Hz', z - z' \rangle &= \langle Hz - Hz', Hz - Hz' + \gamma Fz - \gamma Fz' \rangle \\
&= \tfrac{1}{2}\|Hz - Hz'\|^2 - \tfrac{\gamma^2}{2}\|Fz' - Fz\|^2 + \tfrac{1}{2}\|z' - z\|^2 \\
&\geq \tfrac{1}{2}\|Hz - Hz'\|^2,
\end{aligned} \tag{E.2}$$
where the last inequality is due to Lipschitz continuity and $\gamma \leq 1/L_F$. The strongly monotonicity of $H$ is a consequence of Cauchy-Schwarz and Lipschitz continuity of $F$,
$$\langle Hz' - Hz, z' - z \rangle = \|z' - z\|^2 - \gamma\langle Fz' - Fz, z' - z \rangle \geq (1 - \gamma L)\|z' - z\|^2.$$
The last claim follows from the Cauchy-Schwarz inequality. $\qquad\square$

**Theorem E.2.** *Suppose that Assumptions I to III hold. Moreover, suppose that $\alpha_k \in (0, 1)$, $\gamma \in (\lfloor -2\rho \rfloor_+, 1/L_F)$ and for positive parameters $\varepsilon$ and $b$ the following holds,*
$$\mu := \frac{1}{1+b}\left(1 - \frac{1}{\varepsilon(1-\gamma L_F)^2}\right) - \alpha_0(1 + 2\gamma^2 L_{\hat{F}}^2 A) + \frac{2\rho}{\gamma} > 0 \quad \text{and} \quad 1 - \frac{1}{\varepsilon(1-\gamma L_F)^2} \geq 0 \tag{E.3}$$
*where $A \geq \varepsilon + \frac{1}{b}\left(1 - \frac{1}{\varepsilon(1-\gamma L_F)^2}\right)$. Consider the sequence $(z^k)_{k\in\mathbb{N}}$ generated by Algorithm 2. Then, the following estimate holds for all $z^\star \in \mathcal{S}^\star$*
$$\sum_{k=0}^{K} \frac{\alpha_k}{\sum_{j=0}^{K}\alpha_j}\mathbb{E}[\|h^k - H\bar{z}^k\|^2] \leq \frac{\mathbb{E}[\|z^0 - z^\star\|^2] + A\mathbb{E}[\|h^{-1} - Hz^{-1}\|^2] + C\gamma^2\sigma_F^2\sum_{j=0}^{K}\alpha_j^2}{\mu\sum_{j=0}^{K}\alpha_j} \tag{E.4}$$
*where $C = 1 + 2A(1 + \gamma^2 L_{\hat{F}}^2) + 2\alpha_0 A$.*

*Proof of Theorem E.2.* We rely on the following potential function,
$$\mathcal{U}_{k+1} := \|z^{k+1} - z^\star\|^2 + A_{k+1}\|h^k - Hz^k\|^2 + B_{k+1}\|z^{k+1} - z^k\|^2,$$
where $(A_k)_{k\in\mathbb{N}}$ and $(B_k)_{k\in\mathbb{N}}$ are positive scalar parameters to be identified.

We will denote $\hat{H}_k := \bar{z}^k - \gamma\hat{F}(\bar{z}^k, \bar{\xi}_k)$, so that $z^{k+1} = z^k - \alpha_k(h^k - \hat{H}_k)$. Then, expanding one step,
$$\|z^{k+1} - z^\star\|^2 = \|z^k - z^\star\|^2 - 2\alpha_k\langle h^k - \hat{H}_k, z^k - z^\star \rangle + \alpha_k^2\|h^k - \hat{H}_k\|^2. \tag{E.5}$$
Recall that $Hz := z - \gamma Fz$ in the deterministic case. In the Algorithm 2, $h^k$ estimates $Hz^k$. Let us quantify how good this estimation is.
$$\begin{aligned}
h^k - Hz^k &= \gamma Fz^k - \gamma\hat{F}(z^k, \xi_k) + (1 - \alpha_{k-1})(h^{k-1} - z^{k-1} + \gamma\hat{F}(z^{k-1}, \xi_k)) \\
\|h^k - Hz^k\|^2 &= (1 - \alpha_{k-1})^2\|h^{k-1} - z^{k-1} + \gamma Fz^{k-1}\|^2 \\
&\quad + \|\gamma Fz^k - \gamma\hat{F}(z^k, \xi_k) + (1 - \alpha_{k-1})(\gamma\hat{F}(z^{k-1}, \xi_k) - \gamma Fz^{k-1})\|^2 \\
&\quad + 2(1 - \alpha_{k-1})\langle h^{k-1} - z^{k-1} + \gamma Fz^{k-1}, \\
&\qquad\qquad \gamma Fz^k - \gamma\hat{F}(z^k, \xi_k) + (1 - \alpha_{k-1})(\gamma\hat{F}(z^{k-1}, \xi_k) - \gamma Fz^{k-1})\rangle
\end{aligned}$$

In the scalar product, the left term is known when $z^k$ is known and the right term has an expectation equal to 0 by Assumption II*(ii)* when $z^k$ is known. Thus, taking conditional expectation and using

the fact that the second moment is larger than the variance, we can go on as

$$\mathbb{E}[\|h^k - Hz^k\|^2 \mid \mathcal{F}_k] \leq (1 - \alpha_k)^2 \|h^{k-1} - Hz^{k-1}\|^2$$
$$+ \mathbb{E}[2(1 - \alpha_k)^2 \gamma^2 \|\hat{F}(z^k, \xi_k) - \hat{F}(z^{k-1}, \xi_k)\|^2 \mid \mathcal{F}_k]$$
$$+ \mathbb{E}[2\alpha_k^2 \gamma^2 \|Fz^k - \hat{F}(z^k, \xi_k)\|^2 \mid \mathcal{F}_k]$$
$$\leq (1 - \alpha_k)^2 \|h^{k-1} - Hz^{k-1}\|^2 + 2(1 - \alpha_k)^2 L_{\hat{F}}^2 \gamma^2 \|z^k - z^{k-1}\|^2 + 2\alpha_k^2 \gamma^2 \sigma_F^2 \quad \text{(E.6)}$$

where we have used Assumption II*(iii)* and Assumption III.

We continue with the conditional expectation of the inner term in (E.5).

$$-\mathbb{E}[\langle h^k - \hat{H}_k, z^k - z^\star \rangle \mid \mathcal{F}_k] = -\langle h^k - H\bar{z}^k, z^k - z^\star \rangle$$
$$= -\langle h^k - H\bar{z}^k, z^k - \bar{z}^k \rangle - \langle h^k - H\bar{z}^k, \bar{z}^k - z^\star \rangle$$
$$= -\langle h^k - Hz^k, z^k - \bar{z}^k \rangle - \langle Hz^k - H\bar{z}^k, z^k - \bar{z}^k \rangle - \langle h^k - H\bar{z}^k, \bar{z}^k - z^\star \rangle$$
$$\leq -\langle h^k - Hz^k, z^k - \bar{z}^k \rangle - \tfrac{1}{2}\|Hz^k - H\bar{z}^k\|^2 - \langle h^k - H\bar{z}^k, \bar{z}^k - z^\star \rangle$$
$$\text{(E.7)}$$

where the last inequality uses ½-cocoercivity of $H$ from Lemma F.2*(i)* under Assumption I*(i)* and the choice $\gamma \leq 1/L_F$.

By definition of $\bar{z}^k$ in Step 2.3, we have $h^k \in \bar{z}^k + \gamma A(\bar{z}^k)$, so that $\frac{1}{\gamma}(h^k - H\bar{z}^k) \in F(\bar{z}^k) + A(\bar{z}^k)$. Hence, using the weak MVI from Assumption I*(iii)*,

$$\langle h^k - H\bar{z}^k, \bar{z}^k - z^\star \rangle \geq \tfrac{\rho}{\gamma}\|h^k - H\bar{z}^k\|^2 . \quad \text{(E.8)}$$

Using (E.8) in (E.7) leads to the following inequality, true for any $\varepsilon_k > 0$:

$$-\mathbb{E}[\langle h^k - \hat{H}_k, z^k - z^\star \rangle \mid \mathcal{F}_k] \leq \tfrac{\varepsilon_k}{2}\|h^k - Hz^k\|^2 + \tfrac{1}{2\varepsilon_k}\|\bar{z}^k - z^k\|^2 - \tfrac{1}{2}\|Hz^k - H\bar{z}^k\|^2 - \tfrac{\rho}{\gamma}\|h^k - H\bar{z}^k\|^2 .$$

To majorize the term $\|\bar{z}^k - z^k\|^2$, we use Lemma F.2*(ii)* to get

$$\|H\bar{z}^k - Hz^k\|^2 \geq (1 - \gamma L_F)^2 \|\bar{z}^k - z^k\|^2 .$$

Hence, as long as $\gamma L_F < 1$, then

$$-\mathbb{E}[\langle h^k - \hat{H}_k, z^k - z^\star \rangle \mid \mathcal{F}_k] \leq \tfrac{\varepsilon_k}{2}\|h^k - Hz^k\|^2 + \Big(\tfrac{1}{2\varepsilon_k(1-\gamma L_F)^2} - \tfrac{1}{2}\Big)\|Hz^k - H\bar{z}^k\|^2 - \tfrac{\rho}{\gamma}\|h^k - H\bar{z}^k\|^2 .$$
$$\text{(E.9)}$$

The third term in (E.5) is bounded by

$$\alpha_k^2 \mathbb{E}[\|h^k - \hat{H}_k\|^2 \mid \mathcal{F}_k] = \alpha_k^2\|h^k - H\bar{z}^k\|^2 + \alpha_k^2 \gamma^2 \mathbb{E}[\|F\bar{z}^k - \hat{F}(\bar{z}^k, \bar{\xi}_k)\|^2 \mid \mathcal{F}_k] \leq \alpha_k^2\|h^k - H\bar{z}^k\|^2 + \alpha_k^2 \gamma^2 \sigma_F^2$$
$$\text{(E.10)}$$

Combined with the update rule, (E.10) can also be used to bound the difference of iterates

$$\mathbb{E}[\|z^{k+1} - z^k\|^2 \mid \mathcal{F}_k] = \mathbb{E}[\alpha_k^2\|h^k - \hat{H}_k\|^2 \mid \mathcal{F}_k] \leq \alpha_k^2\|h^k - H\bar{z}^k\|^2 + \alpha_k^2 \gamma^2 \sigma_F^2 \quad \text{(E.11)}$$

Using (E.5), (E.9), (E.10) and (E.11) we have,

$$\mathbb{E}[\mathcal{U}_{k+1} \mid \mathcal{F}_k] \leq \|z^k - z^\star\|^2 + (A_{k+1} + \alpha_k \varepsilon_k)\|h^k - Hz^k\|^2 - \alpha_k\Big(1 - \tfrac{1}{\varepsilon_k(1-\gamma L_F)^2}\Big)\|Hz^k - H\bar{z}^k\|^2$$
$$+ \alpha_k(\alpha_k - \tfrac{2\rho}{\gamma} + \alpha_k B_{k+1})\|h^k - H\bar{z}^k\|^2 + \alpha_k^2(1 + B_{k+1})\gamma^2 \sigma_F^2$$
$$\leq \|z^k - z^\star\|^2 + (A_{k+1} + \alpha_k(\varepsilon_k + \tfrac{1}{b}(1 - \tfrac{1}{\varepsilon_k(1-\gamma L_F)^2})))\|h^k - Hz^k\|^2 \quad \text{(E.12)}$$
$$+ \alpha_k\Big(\alpha_k - \tfrac{2\rho}{\gamma} + \alpha_k B_{k+1} - \tfrac{1}{1+b}(1 - \tfrac{1}{\varepsilon_k(1-\gamma L_F)^2})\Big)\|h^k - H\bar{z}^k\|^2$$
$$+ \alpha_k^2(1 + B_{k+1})\gamma^2 \sigma_F^2,$$

where the last inequality follows from Young's inequality with positive $b$ and requiring $1 - \frac{1}{\varepsilon_k(1-\gamma L_F)^2} \geq 0$ as also stated in (E.3). By defining

$$X_k^1 := A_{k+1} + \alpha_k(\varepsilon_k + \tfrac{1}{b}(1 - \tfrac{1}{\varepsilon_k(1-\gamma L_F)^2}))$$
$$X_k^2 := \alpha_k\Big(\alpha_k - \tfrac{2\rho}{\gamma} + \alpha_k B_{k+1} - \tfrac{1}{1+b}(1 - \tfrac{1}{\varepsilon_k(1-\gamma L_F)^2})\Big) \quad \text{(E.13)}$$

and applying (E.6), we finally obtain

$$
\begin{aligned}
\mathbb{E}[\mathcal{U}_{k+1} \mid \mathcal{F}_k] - \mathcal{U}_k &\leq X_k^2 \|h^k - H\bar{z}^k\|^2 \\
&\quad + (X_k^1(1-\alpha_k)^2 - A_k)\|h^{k-1} - Hz^{k-1}\|^2 \\
&\quad + (2X_k^1(1-\alpha_k)^2\gamma^2 L_{\hat{F}}^2 - B_k)\|z^k - z^{k-1}\|^2 \\
&\quad + 2X_k^1\alpha_k^2\gamma^2\sigma_F^2 + \alpha_k^2(1+B_{k+1})\gamma^2\sigma_F^2,
\end{aligned}
\tag{E.14}
$$

We can pick $B_k = 2\gamma^2 L_{\hat{F}}^2 A_k$ in which case, to get a recursion, we only require the following.

$$
X_k^1(1-\alpha_k)^2 - A_k \leq 0 \quad \text{and} \quad X_k^2 < 0
\tag{E.15}
$$

Set $A_k = A$, $\varepsilon_k = \varepsilon$. For the first requirement of (E.15),

$$
\begin{aligned}
X_k^1(1-\alpha_k)^2 - A_k &= \alpha_k(1-\alpha_k)^2(\varepsilon + \tfrac{1}{b}(1 - \tfrac{1}{\varepsilon(1-\gamma L_F)^2})) + (1-\alpha_k)^2 A - A \\
&\leq \alpha_k(\varepsilon + \tfrac{1}{b}(1 - \tfrac{1}{\varepsilon(1-\gamma L_F)^2})) + (1-\alpha_k)^2 A - A \\
&\leq \alpha_k(\varepsilon + \tfrac{1}{b}(1 - \tfrac{1}{\varepsilon(1-\gamma L_F)^2})) + (1-\alpha_k)A - A \\
&= \alpha_k(\varepsilon + \tfrac{1}{b}(1 - \tfrac{1}{\varepsilon(1-\gamma L_F)^2})) - \alpha_k A
\end{aligned}
\tag{E.16}
$$

where the first inequality follows from $(1-\alpha_k)^2 \leq 1$ and the second inequality follows from $(1-\alpha_k)^2 \leq (1-\alpha_k)$. Thus, to satisfy the first inequality of (E.15) it suffice to pick

$$
A \geq \varepsilon + \tfrac{1}{b}(1 - \tfrac{1}{\varepsilon(1-\gamma L_F)^2}).
\tag{E.17}
$$

The noise term in (E.14) can be made independent of $k$ by using $\alpha_k \leq \alpha_0$ and (E.17) as follows

$$
\begin{aligned}
2X_k^1 + 1 + B_{k+1} &= 1 + 2A(1 + \gamma^2 L_{\hat{F}}^2) + 2\alpha_k(\varepsilon + \tfrac{1}{b}(1 - \tfrac{1}{\varepsilon(1-\gamma L_F)^2})) \\
&\leq 1 + 2A(1 + \gamma^2 L_{\hat{F}}^2) + 2\alpha_0 A = C.
\end{aligned}
\tag{E.18}
$$

Thus it follows from (E.14) and $\alpha_k \leq \alpha_0$ that

$$
\begin{aligned}
&\mathbb{E}[\mathcal{U}_{k+1} \mid \mathcal{F}_k] - \mathcal{U}_k \\
&\leq \alpha_k\big(\alpha_0 - \tfrac{2\rho}{\gamma} + 2\alpha_0\gamma^2 L_{\hat{F}}^2 A - \tfrac{1}{1+b}(1 - \tfrac{1}{\varepsilon_k(1-\gamma L_F)^2})\big)\|h^k - H\bar{z}^k\|^2 + \alpha_k^2 C\gamma^2\sigma_F^2.
\end{aligned}
\tag{E.19}
$$

The result is obtained by total expectation and summing the above inequality while noting that the initial iterate were set as $z^{-1} = z^0$. $\qquad\square$

***Proof of Theorem 7.1.*** The theorem is a specialization of Theorem E.2 with a particular a choice of $b$ and $\varepsilon$. The second requirement in (E.3) can be rewritten as,

$$
\varepsilon \geq \tfrac{1}{(1-\gamma L_F)^2},
\tag{E.20}
$$

which is satisfied by $\varepsilon = \tfrac{1}{\sqrt{\alpha_0}(1-\gamma L_F)^2}$. We substitute in the choice of $\varepsilon$, $b = \sqrt{\alpha_0}$ and denotes $\eta := A$.

The weighted sum in (E.4) can be converted into an expectation over a sampled iterate in the style of Ghadimi & Lan (2013),

$$
\mathbb{E}[\|h^{k_\star} - H\bar{z}^{k_\star}\|^2] = \sum_{k=0}^{K} \tfrac{\alpha_k}{\sum_{j=0}^{K}\alpha_j}\mathbb{E}[\|h^k - H\bar{z}^k\|^2]
$$

with $k_\star$ chosen from $\{0, 1, \ldots, K\}$ according to probability $\mathcal{P}[k_\star = k] = \tfrac{\alpha_k}{\sum_{j=0}^{K}\alpha_j}$.

Noticing that $h^{k_\star} - H\bar{z}^{k_\star} \in \gamma(F\bar{z}^{k_\star} + A\bar{z}^{k_\star}) = \gamma T\bar{z}^{k_\star}$ so

$$
\mathbb{E}[\|h^{k_\star} - H\bar{z}^{k_\star}\|^2] \geq \min_{u \in T\bar{z}^{k_\star}} \mathbb{E}[\|\gamma u\|^2] \geq \mathbb{E}[\min_{u \in T\bar{z}^{k_\star}} \|\gamma u\|^2] =: \mathbb{E}[\mathbf{dist}(0, \gamma T\bar{z}^{k_\star})^2]
$$

where the second inequality follows from concavity of the minimum. This completes the proof. $\quad\square$

## F  PROOF FOR NP-PDEG THROUGH A NONLINEAR ASYMMETRIC PRECONDITIONER

### F.1  PRELIMINARIES

Consider the decomposition $z = (z_1, \ldots, z_m)$, $u = (u_1, \ldots, u_m)$ with $z_i, u_i \in \mathbb{R}^{n_i}$ and define the short-hand notation $u_{\leq i} := (u_1, u_2, \ldots, u_i)$ and $u_{\geq i} := (u_i, \ldots, u_m)$ for the truncated vectors. Moreover sup-

pose that $A$ conforms to the decomposition $Az = (A_1, z_1, \ldots, A_m z_m)$ with $A_i : \mathbb{R}^{n_i} \rightrightarrows \mathbb{R}^{n_i}$ maximally monotone. Consistently with the decomposition define $\Gamma = \mathbf{blkdiag}(\Gamma_1, \ldots, \Gamma_m)$ where $\Gamma_i \in \mathbb{R}^{n_i \times n_i}$ are positive definite matrices and let

$$P_u(z) := \Gamma^{-1} z + Q_u(z), \quad \text{where } Q_u(z) = (0, q_1(z_1, u_{\geq 2}), q_2(z_1, z_2, u_{\geq 3}), \ldots, q_{m-1}(z_{\leq m-1}, u_m)) \quad \text{(F.1)}$$

When $P_u$ furnishes such an asymmetric structure the preconditioned resolvent has full domain, thus ensuring that the algorithm is well-defined.

In the following lemma we show that the iterates in (8.1) are well-defined for a particular choice of the preconditioner $P_u$ in (F.1). The proof is similar to that of (Latafat & Patrinos, 2017, Lem. 3.1) and is included for completeness.

**Lemma F.1.** *Let $z = (z_1, \ldots, z_m)$, $u = (u_1, \ldots, u_m)$ be given vectors, suppose that $A$ conforms to the decomposition $Az = (A_1, z_1, \ldots, A_m z_m)$ with $A_i : \mathbb{R}^{n_i} \rightrightarrows \mathbb{R}^{n_i}$ maximally monotone, and let $P_u$ be defined as in (F.1). Then, the preconditioned resolvent $(P_u + A)^{-1}$ is Lipschitz continuous and has full domain. Moreover, the update $\bar{z} = (P_u + A)^{-1} z$ reduces to the following update*

$$\bar{z}_i = \begin{cases} (\Gamma_1^{-1} + A_1)^{-1} z_1 & \text{if } i = 1 \\ (\Gamma_i^{-1} + A_i)^{-1} (z_i - q_{i-1}(\bar{z}_{\leq i-1}, u_{\geq i}) & \text{if } i = 2, \ldots, m \end{cases} \quad \text{(F.2)}$$

*Proof.* Owing to the asymmetric structure (F.1), the resolvent may equivalently be expressed as

$$\bar{z} = (\bar{z}_1, \ldots \bar{z}_m) = (P_u + A)^{-1} z \iff \Gamma_i^{-1} \bar{z}_i + A_i(\bar{z}_i) \in z_i - q_{i-1}(\bar{z}_{\leq i-1}, u_{\geq i}), \quad i = 1, \ldots, m,$$

where $q_0 \equiv 0$. The Gauss-Seidel-type update in (F.2) is of immediate verification after noting that $(\Gamma_i^{-1} + A_i)^{-1}$ is single-valued (in fact Lipschitz continuous) since the sum of $\Gamma_i > 0$ and $A_i$ is (maximally) strongly monotone. This also implies that $\Gamma_i^{-1} + A_i = \bar{A}_i + \beta I$ for some $\beta > 0$ and some maximally monotone operator $\bar{A}_i$. Thus $\mathbf{dom}\,((\Gamma_i^{-1} + A_i)^{-1}) = \mathbf{range}(\Gamma_i^{-1} + A_i) = \mathbf{range}(\frac{1}{\beta} \bar{A} + I) = \mathbb{R}^n$, where we used Minty's theorem in the last equality. $\qquad\square$

## F.2 DETERMINISTIC LEMMAS

To eventually prove Theorem F.5 we will compare the stochastic algorithm (8.4) with its deterministic counterpart (8.1), so we introduce

$$H_u(z) := P_u(z) - F(z) \quad \text{(F.3a)}$$

$$\bar{G}(z) := (P_z + A)^{-1}(H_z(z)) \quad \text{(F.3b)}$$

$$G(z) := z - \alpha_k \Gamma \big( H_z(z) - H_z(\bar{G}(z)) \big). \quad \text{(F.3c)}$$

We first derive results for the deterministic operator $G$ and then shows that $z^{k+1}$ from the stochastic scheme behaves similarly to $G(z^k)$ when $\alpha_k$ is small enough, even if $\Gamma$, which also appears inside the preconditioner $\hat{P}_u(\cdot, \xi)$, remains large.

Instead of making assumptions on $F$ directly, we instead consider the following important operator,

$$M_u(z) := F(z) - Q_u(z). \quad \text{(F.4)}$$

such that we can write (F.3b) as $H_u(z) = \Gamma^{-1} z - M_u(z)$. As a shorthand we write $M(z) = M_z(z)$.

**Assumption VII.** *The operator $M_u$ as defined in (F.4) is $L_M$-Lipschitz with $L_M \leq 1$ with respect to a positive definite matrix $\Gamma \in \mathbb{R}^{n \times n}$, i.e.*

$$\|M_u(z) - M_u(z')\|_{\Gamma} \leq L_M \|z - z'\|_{\Gamma^{-1}} \quad \forall z, z' \in \mathbb{R}^n. \quad \text{(F.5)}$$

**Remark 6.** This is satisfied by the choice of $Q_u$ in (8.7) and Assumptions IV*(ii)* and IV*(iii)*.

With $M_u$ defined, it is straightforward to establish that $H_u$ is $1/2$-cocoercive and strongly monotone.

**Lemma F.2.** *Suppose Assumption VII holds. Then,*

*(i) The mapping $H_u$ is $1/2$-cocoercive for all $u \in \mathbb{R}^n$, i.e.*

$$\langle H_u(z') - H_u(z), z' - z \rangle \geq \tfrac{1}{2} \|H_u(z') - H_u(z)\|_{\Gamma}^2 \quad \forall z, z' \in \mathbb{R}^n. \quad \text{(F.6)}$$

*(ii) Furthermore, $H_u$ is $(1 - L_M)$-monotone for all $u \in \mathbb{R}^n$, and in particular*

$$\|H_u(z') - H_u(z)\|_{\Gamma} \geq (1 - L_M)\|z' - z\|_{\Gamma^{-1}} \quad \forall z, z' \in \mathbb{R}^n. \quad \text{(F.7)}$$

*Proof.* By expanding using (F.4),

$$H_u(z) - H_u(z') = \Gamma^{-1}(z - z') - (M_u(z) - M_u(z')). \tag{F.8}$$

Using this we can show cocoercivity,

$$\langle H_u(z') - H_u(z), z' - z \rangle = \langle H_u(z') - H_u(z), H_u(z') - H_u(z) - (M_u(z) - M_u(z')) \rangle_\Gamma$$

$$\text{(F.8)} = \tfrac{1}{2}\|H_u(z') - H_u(z)\|_\Gamma^2 + \tfrac{1}{2}\|z' - z\|_{\Gamma^{-1}}^2 - \tfrac{1}{2}\|M_u(z) - M_u(z')\|_\Gamma^2$$

$$\text{Assumption VII} \geq \tfrac{1}{2}\|H_u(z') - H_u(z)\|_\Gamma^2 \tag{F.9}$$

That $H_u$ is strongly monotone follows from Cauchy-Schwarz and Assumption VII,

$$\langle H_u(z') - H_u(z), z' - z \rangle = \|z' - z\|_{\Gamma^{-1}}^2 - \langle M_u(z') - M_u(z), z' - z \rangle \geq (1 - L_M)\|z' - z\|_{\Gamma^{-1}}^2. \tag{F.10}$$

The last claim follows from Cauchy-Schwarz and dividing by $\|z' - z\|_{\Gamma^{-1}}$. $\qquad\square$

We will rely on the resolvent remaining nonexpansive when preconditioned with a variable stepsize matrix.

**Lemma F.3.** *Let $\Gamma \in \mathbb{R}^{n \times n}$ be positive definite and the operator $A : \mathbb{R}^n \rightrightarrows \mathbb{R}^n$ be maximally monotone. Then, $R = (\Gamma^{-1} + A)^{-1}$ is nonexpansive, i.e. $\|Rx - Ry\|_{\Gamma^{-1}} \leq \|x - y\|_\Gamma$ for all $x, y \in \mathbb{R}^n$.*

*Proof.* Let $v \in Rx$ and $u \in Ry$. By maximal monotonicity of $A$,

$$0 \leq \langle v - \Gamma^{-1}x - u + \Gamma^{-1}y, x - y \rangle = -\|x - y\|_{\Gamma^{-1}}^2 + \langle v - u, x - y \rangle.$$

Therefore, using the Cauchy–Schwarz inequality

$$\|x - y\|_{\Gamma^{-1}}^2 \leq \langle v - u, x - y \rangle \leq \|x - y\|_{\Gamma^{-1}}\|v - u\|_\Gamma \tag{F.11}$$

The proof is complete by rearranging. $\qquad\square$

### F.3 STOCHASTIC RESULTS

The stochastic assumptions on $\hat{F}$ in Theorem F.5 propagates to $\hat{M}$ and $\hat{Q}_u$ as captured by the following lemma.

**Lemma F.4.** *Suppose Assumptions II(ii) and IV(iv) for $\hat{F}(z, \xi) = (\nabla_x \hat{\varphi}(z, \xi), -\nabla_y \hat{\varphi}(z, \xi))$ as defined in (8.6). Let $\hat{M}$ and $M$ be as defined in (F.15) and $\hat{Q}_u$ and $Q_u$ as in (8.7) with $\theta \in [0, \infty)$. Then, the following holds for all $z, z' \in \mathbb{R}^n$*

*(i) $\mathbb{E}_\xi[\hat{M}(z, \xi)] = M(z)$ and $\mathbb{E}_\xi[\hat{Q}_{z'}(z, \xi)] = Q_{z'}(z)$*

*(ii) $\mathbb{E}_\xi[\|M(z) - \hat{M}(z, \xi)\|_\Gamma^2] \leq ((1 - \theta)^2 + \theta^2)\sigma_F^2$ and $\mathbb{E}_\xi[\|Q_{z'}(z) - \hat{Q}_{z'}(z, \xi)\|_\Gamma^2] \leq \theta^2 \sigma_F^2$.*

*Proof.* Unbiasedness follows immediately through Assumption II(ii). For the second claim we have for all $(x, y) = z \in \mathbb{R}^n$

$$\mathbb{E}_\xi[\|M(z) - \hat{M}(z, \xi)\|_\Gamma^2] = \mathbb{E}_\xi\left[\left\|\begin{pmatrix} \nabla_x \hat{\varphi}(z, \xi) - \nabla_x \varphi(z) \\ (1 - \theta)(\nabla_y \hat{\varphi}(z, \xi) - \nabla_y \varphi(z')) \end{pmatrix}\right\|_\Gamma^2\right]$$

$$= \mathbb{E}_\xi\left[\left\|\begin{pmatrix} (1 - \theta)(\nabla_x \hat{\varphi}(z, \xi) - \nabla_x \varphi(z)) + \theta(\nabla_x \hat{\varphi}(z, \xi) - \nabla_x \varphi(z)) \\ (1 - \theta)(\nabla_y \hat{\varphi}(z, \xi) - \nabla_y \varphi(z)) \end{pmatrix}\right\|_\Gamma^2\right]$$

$$\text{(Assumption II(ii))} \leq (1 - \theta)^2 \mathbb{E}_\xi\left[\left\|\begin{pmatrix} \nabla_x \hat{\varphi}(z, \xi) - \nabla_x \varphi(z) \\ \nabla_y \hat{\varphi}(z, \xi) - \nabla_y \varphi(z) \end{pmatrix}\right\|_\Gamma^2\right] + \theta^2 \mathbb{E}_\xi\left[\left\|\begin{pmatrix} \nabla_x \hat{\varphi}(z, \xi) - \nabla_x \varphi(z) \\ 0 \end{pmatrix}\right\|_\Gamma^2\right]$$

$$\text{(Assumption IV(iv))} \leq ((1 - \theta)^2 + \theta^2)\sigma_F^2. \tag{F.12}$$

The last claim follows directly through Assumption IV(iv). This completes the proof. $\qquad\square$

**Theorem F.5.** *Suppose that Assumption I(iii) to II(ii) and IV hold. Moreover, suppose that $\alpha_k \in (0, 1)$, $\theta \in [0, \infty)$ and for positive parameter $b$ and $\varepsilon$ the following holds,*

$$\mu := \frac{1}{1+b}\left(1 - \frac{1}{\varepsilon(1-L_M)^2}\right) + \frac{2\rho}{\gamma} - \alpha_0 - 2\alpha_0(\hat{c}_1 + 2\hat{c}_2(1 + \hat{c}_3))A > 0, \tag{F.13}$$

$$1 - 4\hat{c}_2\alpha_0 > 0 \quad and \quad 1 - \frac{1}{\varepsilon(1-L_M)^2} \geq 0$$

where $\bar{\gamma}$ denotes the smallest eigenvalue of $\Gamma$, $A \geq (1 + 4\hat{c}_2\alpha_0^2)(\varepsilon + \frac{1}{b}(1 - \frac{1}{\varepsilon(1-L_M)^2}))/(1 - 4\hat{c}_2\alpha_0)$ and

$$\hat{c}_1 := L_{\widehat{xz}}^2\|\Gamma D_{\widehat{xz}}\| + 2(1-\theta)^2 L_{\widehat{yz}}^2\|\Gamma D_{\widehat{yz}}\| + 2\theta^2 L_{\widehat{yy}}^2\|\Gamma_2 D_{\widehat{yy}}\|, \quad \hat{c}_2 := 2\theta^2 L_{\widehat{yx}}^2\|\Gamma_1 D_{\widehat{yx}}\|, \quad \hat{c}_3 := L_{\widehat{xz}}^2\|\Gamma D_{\widehat{xz}}\|,$$

$$L_M^2 := \max\{L_{xx}^2\|D_{xx}\Gamma_1\| + L_{yx}^2\|D_{yx}\Gamma_1\|, \|L_{xy}^2\|D_{xy}\Gamma_2\| + L_{yy}^2\|D_{yy}\Gamma_2\|\}.$$

Consider the sequence $(z^k)_{k\in\mathbb{N}}$ generated by Algorithm 3. Then, the following holds for all $z^\star \in \mathcal{S}^\star$

$$\sum_{k=0}^{K} \frac{\alpha_k}{\sum_{j=0}^{K}\alpha_j}\mathbb{E}[\|\Gamma^{-1}\hat{z}^k - S_{z^k}(\bar{z}^k; \bar{z}^k)\|_\Gamma^2] \leq \frac{\mathbb{E}[\|z^0 - z^\star\|_{\Gamma^{-1}}^2] + A\mathbb{E}[\|\Gamma^{-1}\hat{z}^{-1} - S_{z^{-1}}(z^{-1}; \bar{z}^{-1})\|_\Gamma^2] + C\sigma_F^2\sum_{j=0}^{K}\alpha_j^2}{\mu\sum_{j=0}^{K}\alpha_j}$$

(F.14)

where $C := 2(A + \alpha_0(\varepsilon + \frac{1}{b}(1 - \frac{1}{\varepsilon(1-L_M)^2})))(\Theta + 2\hat{c}_2) + 1 + 2(\hat{c}_1 + 2\hat{c}_2(1+\hat{c}_3))A$ and $\Theta = (1-\theta)^2 + 2\theta^2$.

***Proof of Theorem F.5.*** The proof relies on tracking the two following important operators instead of $F$ and $\hat{F}$

$$M(z) := F(z) - Q_z(z) \quad \text{and} \quad \hat{M}(z, \xi) := \hat{F}(z, \xi) - \hat{Q}_z(z, \xi). \tag{F.15}$$

We will denote $\hat{H}_k := \hat{P}_k(\bar{z}^k, \bar{\xi}_k) - \hat{F}(\bar{z}^k, \bar{\xi}_k)$, so that $z^{k+1} = z^k - \alpha_k\Gamma(h^k - \hat{H}_k)$. We will further need the following change of variables to later be able to apply weak MVI (see Appendix F.4):

$$
\begin{aligned}
s^k &= h^k - \hat{Q}_{z^k}(\bar{z}^k, \xi'_k) \\
\hat{S}_k &= \hat{H}_k - \hat{Q}_{z^k}(\bar{z}^k, \xi'_k) \\
S_u(\bar{z}) &= H_u(\bar{z}) - Q_u(\bar{z}) \\
S_u(z; \bar{z}) &= H_u(z) - Q_u(\bar{z})
\end{aligned}
\tag{F.16}
$$

where $Q_u(z)$ and $H_u$ are as defined in Section 8.

In contrast with the unconstrained smooth case we will rely on a slightly different potential function, namely,

$$\mathcal{U}_{k+1} := \|z^{k+1} - z^\star\|_{\Gamma^{-1}}^2 + A_{k+1}\|s^k - S_{z^k}(z^k; \bar{z}^k)\|_\Gamma^2 + B_{k+1}\|z^{k+1} - z^k\|_\Gamma^2$$

where $(A_k)_{k\in\mathbb{N}}$ and $(B_k)_{k\in\mathbb{N}}$ are positive scalar parameters to be identified.

We start by writing out one step of the update

$$\|z^{k+1} - z^\star\|_{\Gamma^{-1}}^2 = \|z^k - z^\star\|_{\Gamma^{-1}}^2 - 2\alpha_k\langle h^k - \hat{H}_k, z^k - z^\star\rangle + \alpha_k^2\|h^k - \hat{H}_k\|_\Gamma^2 \tag{F.17}$$

$$= \|z^k - z^\star\|_{\Gamma^{-1}}^2 - 2\alpha_k\langle s^k - \hat{S}_k, z^k - z^\star\rangle + \alpha_k^2\|s^k - \hat{S}_k\|_\Gamma^2 \tag{F.18}$$

In the algorithm, $s^k$ estimates $S_{z^k}(z^k; \bar{z}^k)$. Let us quantify how good this estimation is. We will make use of the careful choice of the bias-correction term to shift the noise index by 1 in the second equality.

$$
\begin{aligned}
s^k - S_{z^k}(z^k; \bar{z}^k) &= M(z^k) + Q_{z^k}(\bar{z}^k) - \hat{M}(z^k, \xi_k) - \hat{Q}_{z^k}(\bar{z}^k, \xi'_k) \\
&\quad + (1 - \alpha_k)(h^{k-1} - \Gamma^{-1}z^{k-1} + \hat{M}(z^{k-1}, \xi_k) - \hat{Q}_{z^{k-1}}(\bar{z}^{k-1}, \xi'_{k-1}) + \hat{Q}_{z^{k-1}}(\bar{z}^{k-1}, \xi'_k)) \\
&= M(z^k) + Q_{z^k}(\bar{z}^k) - \hat{M}(z^k, \xi_k) - \hat{Q}_{z^k}(\bar{z}^k, \xi'_k) \\
&\quad + (1 - \alpha_k)(s^{k-1} + \hat{Q}_{z^{k-1}}(\bar{z}^{k-1}, \bar{\xi}'_k) - \Gamma^{-1}z^{k-1} + \hat{M}(z^{k-1}, \xi_k)) \\
&= M(z^k) + Q_{z^k}(\bar{z}^k) - \hat{M}(z^k, \xi_k) - \hat{Q}_{z^k}(\bar{z}^k, \xi'_k) + (1 - \alpha_k)(s^{k-1} - S_{z^{k-1}}(z^{k-1}; \bar{z}^{k-1})) \\
&\quad + (1 - \alpha_k)(\hat{M}(z^{k-1}, \xi_k) - M(z^{k-1}) + \hat{Q}_{z^{k-1}}(\bar{z}^{k-1}, \bar{\xi}'_k) - Q_{z^{k-1}}(\bar{z}^{k-1}))
\end{aligned}
$$

Using the shorthand notation

$$
\begin{aligned}
\tilde{s}^k &:= s^k - S_{z^k}(z^k; \bar{z}^k), \\
\tilde{Q}_{z^k}(\bar{z}^k, \xi'_k) &:= Q_{z^k}(\bar{z}^k) - \hat{Q}_{z^k}(\bar{z}^k, \xi'_k), \\
\tilde{M}(z^k, \xi_k) &:= M(z^k) - \hat{M}(z^k, \xi_k),
\end{aligned}
$$

it follows that

$$
\begin{aligned}
\|\tilde{s}^k\|_\Gamma^2 &= (1 - \alpha_k)^2\|\tilde{s}^{k-1}\|_\Gamma^2 + \|\tilde{M}(z^k, \xi_k) + \tilde{Q}_{z^k}(\bar{z}^k, \xi'_k) - (1 - \alpha_k)(\tilde{M}(z^{k-1}, \xi_k) + \tilde{Q}_{z^{k-1}}(\bar{z}^{k-1}, \xi'_k))\|_\Gamma^2 \\
&\quad + 2(1 - \alpha_k)\langle\tilde{s}^{k-1}, \tilde{M}(z^k, \xi_k) + \tilde{Q}_{z^k}(\bar{z}^k, \xi'_k) - (1 - \alpha_k)(\tilde{M}(z^{k-1}, \xi_k) + \tilde{Q}_{z^{k-1}}(\bar{z}^{k-1}, \xi'_k))\rangle \quad \text{(F.19)}
\end{aligned}
$$

In the scalar product, the left term is known when $z^k$ is known. Moreover, since $\mathbb{E}[\cdot \mid \mathcal{F}_k] = \mathbb{E}[\mathbb{E}[\cdot \mid \mathcal{F}_k'] \mid \mathcal{F}_k]$, owing to $\mathcal{F}_k \subset \mathcal{F}_k'$, we have

$$\mathbb{E}\Big[\tilde{M}(z^k, \xi_k) + \tilde{Q}_{z^k}(\bar{z}^k, \xi_k') - (1-\alpha_k)(\tilde{M}(z^{k-1}, \xi_k) + \tilde{Q}_{z^{k-1}}(\bar{z}^{k-1}, \xi_k')) \mid \mathcal{F}_k\Big]$$
$$= \mathbb{E}\Big[\tilde{M}(z^k, \xi_k) - (1-\alpha_k)\tilde{M}(z^{k-1}, \xi_k) \mid \mathcal{F}_k\Big] = 0,$$

where we use Assumption II*(ii)* through Lemma F.4*(i)*.

Since the second moment is larger than the variance we have

$$\mathbb{E}\Big[\|\tilde{M}(z^k, \xi_k) + \tilde{Q}_{z^k}(\bar{z}^k, \xi_k') - \tilde{M}(z^{k-1}, \xi_k) - \tilde{Q}_{z^{k-1}}(\bar{z}^{k-1}, \xi_k')\|_\Gamma^2 \mid \mathcal{F}_k\Big] \leq$$
$$\mathbb{E}\Big[\|\hat{M}(z^k, \xi_k) - \hat{M}(z^{k-1}, \xi_k) + \hat{Q}_{z^k}(\bar{z}^k, \xi_k') - \hat{Q}_{z^{k-1}}(\bar{z}^{k-1}, \xi_k')\|_\Gamma^2 \mid \mathcal{F}_k\Big] \tag{F.20}$$

Using the Young inequality it follows from (F.19), (F.20) that

$$\mathbb{E}[\|\tilde{s}^k\|_\Gamma^2 \mid \mathcal{F}_k] \leq (1-\alpha_k)^2\|\tilde{s}^{k-1}\|_\Gamma^2 + 2\alpha_k^2\mathbb{E}[\|\tilde{M}(z^k, \xi_k) + \tilde{Q}_{z^k}(\bar{z}^k, \xi_k')\|_\Gamma^2 \mid \mathcal{F}_k]$$
$$+ 2(1-\alpha_k)^2\mathbb{E}[\|\tilde{M}(z^k, \xi_k) + \tilde{Q}_{z^k}(\bar{z}^k, \xi_k') - \tilde{M}(z^{k-1}, \xi_k) - \tilde{Q}_{z^{k-1}}(\bar{z}^{k-1}, \xi_k')\|_\Gamma^2 \mid \mathcal{F}_k]$$
$$\leq (1-\alpha_k)^2\|\tilde{s}^{k-1}\|_\Gamma^2 + 2\alpha_k^2\mathbb{E}[\|M(z^k) - \hat{M}(z^k, \xi_k) + Q_{z^k}(\bar{z}^k) - \hat{Q}_{z^k}(\bar{z}^k, \xi_k')\|_\Gamma^2 \mid \mathcal{F}_k]$$
$$+ \mathbb{E}[2(1-\alpha_k)^2\|\hat{M}(z^k, \xi_k) - \hat{M}(z^{k-1}, \xi_k) + \hat{Q}_{z^k}(\bar{z}^k, \xi_k') - \hat{Q}_{z^{k-1}}(\bar{z}^{k-1}, \xi_k')\|_\Gamma^2 \mid \mathcal{F}_k] \tag{F.21}$$

To bound the second last term of (F.21) we use unbiasedness due to Assumption II*(ii)* through Lemma F.4*(i)* and that $\mathbb{E}[\cdot \mid \mathcal{F}_k] = \mathbb{E}[\mathbb{E}[\cdot \mid \mathcal{F}_k'] \mid \mathcal{F}_k]$, owing to $\mathcal{F}_k \subset \mathcal{F}_k'$

$$\mathbb{E}[\|M(z^k) - \hat{M}(z^k, \xi_k) + Q_{z^k}(\bar{z}^k) - \hat{Q}_{z^k}(\bar{z}^k, \xi_k')\|_\Gamma^2 \mid \mathcal{F}_k]$$
$$= \mathbb{E}[\|M(z^k) - \hat{M}(z^k, \xi_k)\|_\Gamma^2 \mid \mathcal{F}_k] + \mathbb{E}[\mathbb{E}[\|Q_{z^k}(\bar{z}^k) - \hat{Q}_{z^k}(\bar{z}^k, \xi_k')\|_\Gamma^2 \mid \mathcal{F}_k'] \mid \mathcal{F}_k]$$
$$\leq \Theta\sigma_F^2 \tag{F.22}$$

with $\Theta = (1-\theta)^2 + 2\theta^2$. where the last inequality follows from Assumptions II*(ii)* and IV*(iv)* through Lemma F.4*(ii)*.

To bound the last term of (F.21) we use the particular choice of $Q_u$,

$$\hat{M}(z^k, \xi_k) - \hat{M}(z^{k-1}, \xi_k) + \hat{Q}_{z^k}(\bar{z}^k, \xi_k') - \hat{Q}_{z^{k-1}}(\bar{z}^{k-1}, \xi_k')$$
$$= \begin{pmatrix} \nabla_x\hat{\varphi}(z^k, \xi_k) - \nabla_x\hat{\varphi}(z^{k-1}, \xi_k) \\ (1-\theta)(\nabla_y\hat{\varphi}(z^{k-1}\xi_k) - \nabla_y\hat{\varphi}(z^k, \xi_k)) - \theta(\nabla_y\hat{\varphi}(\bar{x}^k, y^k, \xi_k') - \nabla_y\hat{\varphi}(\bar{x}^k, y^k, \xi_k')) \end{pmatrix}. \tag{F.23}$$

So Assumption IV*(v)* applies after application of Young's inequality and the tower rule, leading to the following bound

$$\mathbb{E}[\|\hat{M}(z^k, \xi_k) - \hat{M}(z^{k-1}, \xi_k) + \hat{Q}_{z^k}(\bar{z}^k, \xi_k') - \hat{Q}_{z^{k-1}}(\bar{z}^{k-1}, \xi_k')\|_\Gamma^2 \mid \mathcal{F}_k]$$
$$= \mathbb{E}[\|\nabla_x\hat{\varphi}(z^k, \xi_k) - \nabla_x\hat{\varphi}(z^{k-1}, \xi_k)\|_{\Gamma_1}^2 \mid \mathcal{F}_k]$$
$$+ \mathbb{E}[\|(1-\theta)(\nabla_y\hat{\varphi}(z^{k-1}\xi_k) - \nabla_y\hat{\varphi}(z^k, \xi_k)) - \theta\nabla_y\hat{\varphi}(\bar{x}^k, y^k, \xi_k') - \nabla_y\hat{\varphi}(\bar{x}^k, y^k, \xi_k')\|_{\Gamma_2}^2 \mid \mathcal{F}_k]$$
$$\leq \mathbb{E}[\|\nabla_x\hat{\varphi}(z^k, \xi_k) - \nabla_x\hat{\varphi}(z^{k-1}, \xi_k)\|_{\Gamma_1}^2 \mid \mathcal{F}_k]$$
$$+ 2(1-\theta)^2\mathbb{E}[\|(\nabla_y\hat{\varphi}(z^{k-1}\xi_k) - \nabla_y\hat{\varphi}(z^k, \xi_k))\|_{\Gamma_2}^2 \mid \mathcal{F}_k]$$
$$+ 2\theta^2\mathbb{E}\Big[\mathbb{E}[\|\nabla_y\hat{\varphi}(\bar{x}^k, y^k, \xi_k') - \nabla_y\hat{\varphi}(\bar{x}^k, y^k, \xi_k')\|_{\Gamma_2}^2 \mid \mathcal{F}_k'] \mid \mathcal{F}_k\Big]$$
$$\text{Assumption IV}(v) \leq L_{\widehat{xz}}^2\|z^k - z^{k-1}\|_{D_{\widehat{xz}}}^2 + 2(1-\theta)^2 L_{\widehat{yz}}^2\|z^k - z^{k-1}\|_{D_{\widehat{yz}}}^2$$
$$+ 2\theta^2 L_{\widehat{yy}}^2\|y^k - y^{k-1}\|_{D_{\widehat{yy}}}^2 + 2\theta^2 L_{\widehat{yx}}^2\|\bar{x}^k - \bar{x}^{k-1}\|_{D_{\widehat{yx}}}^2$$
$$\leq \hat{c}_1\|z^k - z^{k-1}\|_{\Gamma^{-1}}^2 + \hat{c}_2\|\bar{x}^k - \bar{x}^{k-1}\|_{\Gamma^{-1}}^2 \tag{F.24}$$

where $\hat{c}_1 := L_{\widehat{xz}}^2\|\Gamma D_{\widehat{xz}}\| + 2(1-\theta)^2 L_{\widehat{yz}}^2\|\Gamma D_{\widehat{yz}}\| + 2\theta^2 L_{\widehat{yy}}^2\|\Gamma_2 D_{\widehat{yy}}\|$ and $\hat{c}_2 := 2\theta^2 L_{\widehat{yx}}^2\|\Gamma_1 D_{\widehat{yx}}\|$.

Using (F.24) and (F.22) in (F.21) yields,

$$\mathbb{E}[\|\tilde{s}^k\|_\Gamma^2 \mid \mathcal{F}_k] \leq (1-\alpha_k)^2\|\tilde{s}^{k-1}\|_\Gamma^2 + 2\alpha_k^2\Theta\sigma_F^2 + 2(1-\alpha_k)^2\Big(\hat{c}_1\|z^k - z^{k-1}\|_{\Gamma^{-1}}^2 + \hat{c}_2\|\bar{x}^k - \bar{x}^{k-1}\|_{\Gamma_1^{-1}}^2\Big). \tag{F.25}$$

To majorize $\|\bar{x}^k - \bar{x}^{k-1}\|_{\Gamma_1^{-1}}$ in (F.25) let $s_x^k$ be the primal components of $s^k$ in what follows. Recall that $A$ decomposes as specified in Section 8, such that we can write $s_x^k \in \Gamma_1^{-1}\bar{x}^k + A_1(\bar{x}^k)$. By monotonicity

of $A_1$ we have through Lemma F.3 that

$$\|\bar{x}^k - \bar{x}^{k-1}\|_{\Gamma_1^{-1}} \le \|s_x^k - s_x^{k-1}\|_{\Gamma_1}. \tag{F.26}$$

We can go on as

$$\begin{aligned}
\|s_x^k - s_x^{k-1}\|_{\Gamma_1} &= \|\Gamma_1^{-1}x^k - \nabla_x\hat{\phi}(z^k, \xi_k) + (1-\alpha_k)(\Gamma_1^{-1}(x^{k-1} - x^{k-1}) + \hat{\nabla}_x\hat{\phi}(z^{k-1}, \xi_k)) - s_x^{k-1}\|_{\Gamma_1} \\
&\le (1-\alpha_k)\|x^k - x^{k-1}\|_{\Gamma_1^{-1}} + (1-\alpha_k)\|\nabla_x\hat{\phi}(z^k, \xi_k) - \nabla_x\hat{\phi}(z^{k-1}, \xi_k)\|_{\Gamma_1} \\
&\quad + \alpha_k\|\Gamma_1^{-1}x^k - \nabla_x\hat{\phi}(z^k, \xi_k) - s_x^{k-1}\|_{\Gamma_1} \\
\text{(Assumption IV}(v)) \quad &\le (1-\alpha_k)\|x^k - x^{k-1}\|_{\Gamma_1^{-1}} + (1-\alpha_k)L_{\widehat{xz}}\|z^k - z^{k-1}\|_{D_{\widehat{xz}}} \\
&\quad + \alpha_k\|\Gamma_1^{-1}x^k - \nabla_x\hat{\phi}(z^k, \xi_k) - s_x^{k-1}\|_{\Gamma_1} \\
&= (1-\alpha_k)\|x^k - x^{k-1}\|_{\Gamma_1^{-1}} + (1-\alpha_k)L_{\widehat{xz}}\|z^k - z^{k-1}\|_{D_{\widehat{xz}}} \\
&\quad + \alpha_k\|s_x^k - s_x^{k-1}\|_{\Gamma_1^{-1}} + \alpha_k(1-\alpha_k)\|\Gamma_1^{-1}x^{k-1} - \nabla_x\hat{\phi}(z^{k-1}, \xi_k) - s_x^{k-1}\|_{\Gamma_1},
\end{aligned}$$

where the last equality uses $\|a - b\|^2 = \|a\|^2 + \|b\|^2 - 2\langle a, b\rangle$ and unbiasedness from Assumption II$(ii)$ to conclude that the inner product is zero.

Hence, by subtracting $\alpha_k\|s_x^k - s_x^{k-1}\|_{\Gamma_1^{-1}}$ and diving by $1 - \alpha_k$, we get

$$\mathbb{E}[\|s_x^k - s_x^{k-1}\|_{\Gamma_1^{-1}}^2 \mid \mathcal{F}_k] \le 2(1+\hat{c}_3)\|x^k - x^{k-1}\|_{\Gamma_1^{-1}}^2 + 2\alpha_k^2\mathbb{E}[\|\Gamma^{-1}x^{k-1} - \nabla_x\hat{\phi}(z^{k-1}, \xi_k) - s_x^{k-1}\|_{\Gamma_1}^2 \mid \mathcal{F}_k]$$

$$\text{Assumptions II}(ii) \text{ and IV}(iv) \quad \le 2(1+\hat{c}_3)\|x^k - x^{k-1}\|_{\Gamma_1^{-1}}^2 + 2\alpha_k^2\mathbb{E}[\|\Gamma^{-1}x^{k-1} - \nabla_x\phi(z^{k-1}) - s_x^{k-1}\|_{\Gamma_1}^2 \mid \mathcal{F}_k] + 2\alpha_k^2\sigma_F^2$$

$$\le 2(1+\hat{c}_3)\|z^k - z^{k-1}\|_{\Gamma^{-1}}^2 + 2\alpha_k^2\mathbb{E}[\|S_{z^{k-1}}(z^{k-1}; \bar{z}^{k-1}) - s^{k-1}\|_{\Gamma}^2 \mid \mathcal{F}_k] + 2\alpha_k^2\sigma_F^2$$

where $\hat{c}_3 := L_{\widehat{xz}}^2\|\Gamma D_{\widehat{xz}}\|$ and the last inequality reintroduces the $y$-components.

We finally obtain

$$\mathbb{E}[\|\bar{x}^k - \bar{x}^{k-1}\|_{\Gamma^{-1}}^2 \mid \mathcal{F}_k] \le 2(1+\hat{c}_3)\|z^k - z^{k-1}\|_{\Gamma^{-1}}^2 + 2\alpha_k^2\mathbb{E}[\|s^{k-1} - S_{z^{k-1}}(z^{k-1}; \bar{z}^{k-1})\|_{\Gamma}^2 \mid \mathcal{F}_k] + 2\alpha_k^2\sigma_F. \tag{F.27}$$

Introducing (F.27) into (F.25) yields

$$\begin{aligned}
\mathbb{E}[\|s^k - S_{z^k}(z^k; \bar{z}^k)\|_{\Gamma}^2 \mid \mathcal{F}_k] &\le (1-\alpha_k)^2(1 + 4\hat{c}_2\alpha_k^2)\|s^{k-1} - S_{z^{k-1}}(z^{k-1}; \bar{z}^{k-1})\|_{\Gamma}^2 \\
&\quad + 2(1-\alpha_k)^2(\hat{c}_1 + 2\hat{c}_2(1+\hat{c}_3))\|z^k - z^{k-1}\|_{\Gamma^{-1}}^2 \\
&\quad + 2\alpha_k^2(\Theta + (1-\alpha_k)^2 2\hat{c}_2)\sigma_F^2.
\end{aligned} \tag{F.28}$$

We continue with the inner term in (F.18) under conditional expectation.

$$\begin{aligned}
-\mathbb{E}[\langle s^k - \hat{\bar{S}}_k, z^k - z^\star\rangle_{\Gamma} \mid \mathcal{F}_k] \\
&= -\langle s^k - S_{z^k}(\bar{z}^k), z^k - z^\star\rangle \\
&= -\langle s^k - S_{z^k}(\bar{z}^k), z^k - \bar{z}^k\rangle - \langle s^k - S_{z^k}(\bar{z}^k), \bar{z}^k - z^\star\rangle \\
&= -\langle s^k - S_{z^k}(z^k; \bar{z}^k), z^k - \bar{z}^k\rangle - \langle S_{z^k}(z^k; \bar{z}^k) - S_{z^k}(\bar{z}^k), z^k - \bar{z}^k\rangle - \langle s^k - S_{z^k}(\bar{z}^k), \bar{z}^k - z^\star\rangle \\
&= -\langle s^k - S_{z^k}(z^k; \bar{z}^k), z^k - \bar{z}^k\rangle - \langle H_{z^k}(z^k) - H_{z^k}(\bar{z}^k), z^k - \bar{z}^k\rangle - \langle s^k - S_{z^k}(\bar{z}^k), \bar{z}^k - z^\star\rangle
\end{aligned}$$

where the last equality uses that $S_{z^k}(z^k; \bar{z}^k) - S_{z^k}(\bar{z}^k) = H_{z^k}(z^k) - H_{z^k}(\bar{z}^k)$.

By definition of $\bar{z}^k$ in (8.4b), we have $s^k = h^k - \hat{Q}_{z^k}(\bar{z}^k, \xi'_k) \in \Gamma^{-1}\bar{z}^k + A(\bar{z}^k)$, so that $s^k - S_{z^k}(\bar{z}^k) \in F(\bar{z}^k) + A(\bar{z}^k)$. Hence, using the weak MVI from Assumption I$(iii)$,

$$\langle s^k - S_{z^k}(\bar{z}^k), \bar{z}^k - z^\star\rangle \ge \rho\|s^k - S_{z^k}(\bar{z}^k)\|^2. \tag{F.29}$$

Using also cocoercivity of $H_u$ from Lemma F.2$(i)$, this leads to the following inequality, true for any $\varepsilon_k > 0$:

$$\begin{aligned}
-\mathbb{E}[\langle s^k - \hat{\bar{S}}_k, z^k - z^\star\rangle \mid \mathcal{F}_k] &\le \frac{\varepsilon_k}{2}\|s^k - S_{z^k}(z^k; \bar{z}^k)\|_{\Gamma}^2 + \frac{1}{2\varepsilon_k}\|\bar{z}^k - z^k\|_{\Gamma^{-1}}^2 \\
&\quad - \frac{1}{2}\|H_{z^k}(z^k) - H_{z^k}(\bar{z}^k)\|_{\Gamma}^2 - \rho\|s^k - S_{z^k}(\bar{z}^k)\|^2.
\end{aligned}$$

To majorize the term $\|\bar{z}^k - z^k\|_{\Gamma^{-1}}^2$, we may use Lemma F.2$(ii)$ for which we need to determind $L_M$. For the particular choice of $Q_u$, we have through Assumption IV$(ii)$ that

$$\|M(z') - M(z)\|_{\Gamma}^2 \le L_M^2\|z' - z\|_{\Gamma^{-1}}^2 \tag{F.30}$$

with $L_M^2 := \max\{L_{xx}^2\|D_{xx}\Gamma_1\| + L_{yx}^2\|D_{yx}\Gamma_1\|, \|L_{xy}^2\|D_{xy}\Gamma_2\| + L_{yy}^2\|D_{yy}\Gamma_2\|\}$. By the stepsize choice [Assumption IV](iii), $L_M < 1$, which will be important promptly.

From [Lemma F.2](ii) it then follows that

$$\|H_{z^k}(z^k) - H_{z^k}(\bar{z}^k)\|_\Gamma^2 \geq (1 - L_M)^2 \|z^k - \bar{z}^k\|_{\Gamma^{-1}}^2 .$$

Hence, given $L_M < 1$,

$$-\mathbb{E}[\langle s^k - \hat{\bar{S}}_k, z^k - z^\star\rangle_\Gamma \mid \mathcal{F}_k]$$
$$\leq \frac{\varepsilon_k}{2}\|s^k - S_{z^k}(z^k; \bar{z}^k)\|_\Gamma^2 + \left(\frac{1}{2\varepsilon_k(1-L_M)^2} - \frac{1}{2}\right)\|H_{z^k}(z^k) - H_{z^k}(\bar{z}^k)\|_\Gamma^2 - \rho\|s^k - S_{z^k}(\bar{z}^k)\|^2$$
$$= \frac{\varepsilon_k}{2}\|s^k - S_{z^k}(z^k; \bar{z}^k)\|_\Gamma^2 + \left(\frac{1}{2\varepsilon_k(1-L_M)^2} - \frac{1}{2}\right)\|S_{z^k}(z^k; \bar{z}^k) - S_{z^k}(\bar{z}^k)\|_\Gamma^2 - \rho\|s^k - S_{z^k}(\bar{z}^k)\|^2. \quad \text{(F.31)}$$

The conditional expectation of the third term in (F.18) is bounded by

$$\alpha_k^2\mathbb{E}[\|s^k - \hat{\bar{S}}_k\|_\Gamma^2 \mid \mathcal{F}_k] = \alpha_k^2\|s^k - S_{z^k}(\bar{z}^k)\|_\Gamma^2 + \alpha_k^2\mathbb{E}[\|F(\bar{z}^k) - \hat{F}(\bar{z}^k, \bar{\xi}_k)\|_\Gamma^2 \mid \mathcal{F}_k]$$
$$\leq \alpha_k^2\|s^k - S_{z^k}(\bar{z}^k)\|_\Gamma^2 + \alpha_k^2\sigma_F^2 \quad \text{(F.32)}$$

where we have used [Assumption IV](iv).

Combined with the update rule, (F.32) can also be used to bound the conditional expectation of the difference of iterates

$$\mathbb{E}[\|z^{k+1} - z^k\|_{\Gamma^{-1}}^2 \mid \mathcal{F}_k] = \mathbb{E}[\alpha_k^2\|s^k - \hat{\bar{S}}_k\|_\Gamma^2 \mid \mathcal{F}_k] \leq \alpha_k^2\|s^k - S_{z^k}(\bar{z}^k)\|_\Gamma^2 + \alpha_k^2\sigma_F^2 \quad \text{(F.33)}$$

Using (F.18), (F.31), (F.32), (F.33) and that $-\rho\|s^k - S_{z^k}(\bar{z}^k)\|^2 \leq -\frac{\rho}{\bar{\gamma}}\|s^k - S_{z^k}(\bar{z}^k)\|_\Gamma^2$ with $\bar{\gamma}$ denoting the smallest eigenvalue of $\Gamma$ we have,

$$\mathbb{E}[\mathcal{U}_{k+1} \mid \mathcal{F}_k] \leq \|z^k - z^\star\|_{\Gamma^{-1}}^2 + (A_{k+1} + \alpha_k\varepsilon_k)\|s^k - S_{z^k}(z^k; \bar{z}^k)\|_\Gamma^2 - \alpha_k\left(1 - \frac{1}{\varepsilon_k(1-L_M)^2}\right)\|S_{z^k}(z^k; \bar{z}^k) - S_{z^k}(\bar{z}^k)\|_\Gamma^2$$
$$+ \alpha_k(\alpha_k - \frac{2\rho}{\bar{\gamma}} + \alpha_k B_{k+1})\|s^k - S_{z^k}(\bar{z}^k)\|_\Gamma^2 + \alpha_k^2(1 + B_{k+1})\sigma_F^2$$
$$\leq \|z^k - z^\star\|_{\Gamma^{-1}}^2 + (A_{k+1} + \alpha_k(\varepsilon_k + \frac{1}{b}(1 - \frac{1}{\varepsilon_k(1-L_M)^2})))\|s^k - S_{z^k}(z^k; \bar{z}^k)\|_\Gamma^2$$
$$+ \alpha_k\left(\alpha_k - \frac{2\rho}{\bar{\gamma}} + \alpha_k B_{k+1} - \frac{1}{1+b}(1 - \frac{1}{\varepsilon_k(1-L_M)^2})\right)\|H_{z^k}(z^k) - H_{z^k}(\bar{z}^k)\|_\Gamma^2$$
$$+ \alpha_k^2(1 + B_{k+1})\sigma_F^2, \quad \text{(F.34)}$$

where the last inequality follows from Young's inequality with positive $b$ as long as $1 - \frac{1}{\varepsilon_k(1-L_M)^2} \geq 0$.

By defining

$$X_k^1 := A_{k+1} + \alpha_k(\varepsilon_k + \frac{1}{b}(1 - \frac{1}{\varepsilon_k(1-L_M)^2}))$$
$$X_k^2 := \frac{2\rho}{\bar{\gamma}} - \alpha_k - \alpha_k B_{k+1} + \frac{1}{1+b}(1 - \frac{1}{\varepsilon_k(1-L_M)^2}) \quad \text{(F.35)}$$

and applying (F.28), we finally obtain

$$\mathbb{E}[\mathcal{U}_{k+1} \mid \mathcal{F}_k] - \mathcal{U}_k \leq -\alpha_k X_k^2\|s^k - S_{z^k}(\bar{z}^k)\|_\Gamma^2$$
$$+ (X_k^1(1 - \alpha_k)^2(1 + 4\hat{c}_2\alpha_k^2) - A_k)\|s^{k-1} - S_{z^{k-1}}(z^{k-1}; \bar{z}^{k-1})\|_\Gamma^2$$
$$+ (2X_k^1(1 - \alpha_k)^2(\hat{c}_1 + 2\hat{c}_2(1 + \hat{c}_3)) - B_k)\|z^k - z^{k-1}\|_{\Gamma^{-1}}^2$$
$$+ 2X_k^1\alpha_k^2(\Theta + (1 - \alpha_k)^2 2\hat{c}_2)\sigma_F^2 + \alpha_k^2(1 + B_{k+1})\sigma_F^2, \quad \text{(F.36)}$$

If $A_k \geq X_k^1(1 - \alpha_k)^2(1 + 4\hat{c}_2\alpha_k^2)$, then it suffice to pick $B_k$ as

$$2X_k^1(1 - \alpha_k)^2(\hat{c}_1 + 2\hat{c}_2(1 + \hat{c}_3)) = \frac{2(\hat{c}_1 + 2\hat{c}_2(1+\hat{c}_3))A_k}{1 + 4\hat{c}_2\alpha_k^2} \leq 2(\hat{c}_1 + 2\hat{c}_2(1 + \hat{c}_3))A_k =: B_k. \quad \text{(F.37)}$$

To get a recursion, we then only require the following conditions

$$X_k^1(1 - \alpha_k)^2(1 + 4\hat{c}_2\alpha_k^2) \leq A_k \quad \text{and} \quad X_k^2 > 0. \quad \text{(F.38)}$$

Set $A_k = A$, $\varepsilon_k = \varepsilon$. For the first inequality of (F.38), since $(1 - \alpha_k)^2 \leq (1 - \alpha_k)$, the terms involving $A$ are bounded as

$$(1 - \alpha_k)^2(1 + 4\hat{c}_2\alpha_k^2)A - A$$
$$\leq (1 - \alpha_k)(1 + 4\hat{c}_2\alpha_k^2)A - A$$
$$= -\alpha_k A + (1 - \alpha_k)(4\hat{c}_2\alpha_k^2)A$$
$$\leq -\alpha_k(1 - 4\hat{c}_2\alpha_0)A \quad \text{(F.39)}$$

where the last inequality follows from $(1 - \alpha_k) \leq 1$ and $\alpha_k \leq \alpha_0$. Thus to satisfy the first inequality of (F.38) it suffice to pick

$$A \geq \frac{(1 + 4\hat{c}_2\alpha_0^2)(\varepsilon + \frac{1}{b}(1 - \frac{1}{\varepsilon(1-L_M)^2}))}{1 - 4\hat{c}_2\alpha_0} \tag{F.40}$$

where $1 - 4\hat{c}_2\alpha_0 > 0$ is required.

The second equality of (F.38) is satisfied owing to (F.13).

The noise term in (F.36) can be made independent of $k$ by using $\alpha_k \leq \alpha_0$,

$$2X_k^1(1 + (1 - \alpha_k)^2 2\hat{c}_2) + 1 + B_{k+1}$$
$$= 2(A + \alpha_k(\varepsilon + \tfrac{1}{b}(1 - \tfrac{1}{\varepsilon(1-L_M)^2})))(\Theta + (1 - \alpha_k)^2 2\hat{c}_2) + 1 + 2(\hat{c}_1 + 2\hat{c}_2(1 + \hat{c}_3))A$$
$$\leq 2(A + \alpha_0(\varepsilon + \tfrac{1}{b}(1 - \tfrac{1}{\varepsilon(1-L_M)^2})))(\Theta + 2\hat{c}_2) + 1 + 2(\hat{c}_1 + 2\hat{c}_2(1 + \hat{c}_3))A =: C. \tag{F.41}$$

Thus, it follows from (F.36) that

$$\mathbb{E}[\mathcal{U}_{k+1} \mid \mathcal{F}_k] - \mathcal{U}_k$$
$$\leq \alpha_k\left(\alpha_0 - \tfrac{2\rho}{\gamma} + 2\alpha_0(\hat{c}_1 + 2\hat{c}_2(1 + \hat{c}_3))A - \tfrac{1}{1+b}(1 - \tfrac{1}{\varepsilon(1-L_M)^2})\right)\|s^k - S_{z^k}(\bar{z}^k)\|_\Gamma^2 \tag{F.42}$$
$$+ \alpha_k^2 C \sigma_F^2.$$

The result is obtained by total expectation and summing the above inequality while noting that the initial iterate were set as $z^{-1} = z^0$. $\qquad\square$

***Proof of Theorem 8.2.*** The theorem is a specialization of Theorem F.5 for a particular a choice of $b$ and $\varepsilon$. The third requirement of (F.13) can be rewritten as,

$$\varepsilon \geq \tfrac{1}{(1-L_M)^2}, \tag{F.43}$$

which is satisfied by $\varepsilon = \frac{1}{\sqrt{\alpha_0}(1-L_M)^2}$. We substitute in the choice of $\varepsilon$, $b = \sqrt{\alpha_0}$ and denotes $\eta := A$.

The weighted sum in (F.14) is equivalent to an expectation over a sampled iterate in the style of Ghadimi & Lan (2013),

$$\mathbb{E}[\|\Gamma^{-1}\hat{z}^{k_\star} - S_{z^{k_\star}}(\bar{z}^{k_\star}; \bar{z}^{k_\star})\|_\Gamma^2] = \sum_{k=0}^{K} \tfrac{\alpha_k}{\sum_{j=0}^{K}\alpha_j} \mathbb{E}[\|\Gamma^{-1}\hat{z}^k - S_{z^k}(\bar{z}^k; \bar{z}^k)\|_\Gamma^2].$$

with $k_\star$ chosen from $\{0, 1, \ldots, K\}$ according to probability $\mathcal{P}[k_\star = k] = \frac{\alpha_k}{\sum_{j=0}^{K}\alpha_j}$.

Noticing that $\Gamma^{-1}\hat{z}^{k_\star} - S_{z^{k_\star}}(\bar{z}^{k_\star}; \bar{z}^{k_\star}) \in F\bar{z}^{k_\star} + A\bar{z}^{k_\star} = T\bar{z}^{k_\star}$ so

$$\mathbb{E}[\|\Gamma^{-1}\hat{z}^{k_\star} - S_{z^{k_\star}}(\bar{z}^{k_\star}; \bar{z}^{k_\star})\|_\Gamma^2] \geq \min_{u \in T\bar{z}^{k_\star}} \mathbb{E}[\|u\|_\Gamma^2] \geq \mathbb{E}[\min_{u \in T\bar{z}^{k_\star}} \|u\|_\Gamma^2] =: \mathbb{E}[\mathbf{dist}_\Gamma(0, T\bar{z}^{k_\star})^2]$$

where the second inequality follows from concavity of the minimum. This completes the proof.

$\qquad\square$

## F.4 EXPLANATION OF BIAS-CORRECTION TERM

Consider the naive analysis which would track $h^k$. By the definition of $\bar{z}^k$ in (8.4b) and $H_k(\bar{z}^k)$ we would have $h^k - H_k(\bar{z}^k) + P_k(\bar{z}^k) - \hat{P}_k(\bar{z}^k, \bar{\xi}_k) \in F(\bar{z}^k) + A(\bar{z}^k)$. Hence, assuming zero mean and using the weak MVI from Assumption I*(iii)*,

$$\mathbb{E}[\langle h^k - H_k(\bar{z}^k), \bar{z}^k - z^\star \rangle \mid \mathcal{F}_k'] = \mathbb{E}[\langle h^k - H_k(\bar{z}^k) + P_k(\bar{z}^k) - \hat{P}_k(\bar{z}^k, \xi_k'), \bar{z}^k - z^\star \rangle \mid \mathcal{F}_k'] \tag{F.44}$$
$$\geq \mathbb{E}[\rho\|h^k - H_k(\bar{z}^k) + P_k(\bar{z}^k) - \hat{P}_k(\bar{z}^k, \xi_k')\|^2 \mid \mathcal{F}_k'].$$

To proceed we could apply Young's inequality, but this would produce a noise term, which would propagate to the descent inequality in (F.36) with a $\alpha_k$ factor in front. To show convergence we would instead need a smaller factor of $\alpha_k^2$.

To avoid this error term entirely we instead do a change of variables with $s^k := h^k - \hat{P}_{z^k}(\bar{z}^k, \xi_k')$ such that,

$$h^k \in \hat{P}_{z^k}(\bar{z}^k, \xi_k') + A\bar{z}^k \Leftrightarrow h^k - \hat{P}_{z^k}(\bar{z}^k, \xi_k') \in A\bar{z}^k$$
$$\Leftrightarrow s^k \in A\bar{z}^k. \tag{F.45}$$

This make application of Assumption I*(iii)* unproblematic, but affects the choice of the bias-correction term, since the analysis will now apply to $s^k$. If we instead of the careful choice of $h^k$ in (8.4a) had made the choice

$$h^k = \hat{P}_{z^k}(z^k, \xi_k) - \hat{F}(z^k, \xi_k) + (1 - \alpha_k)(h^{k-1} - \hat{P}_{z^{k-1}}(z^{k-1}, \xi_k) + \hat{F}(z^{k-1}, \xi_k)) \tag{F.46}$$

then

$$s^k = \hat{P}_{z^k}(z^k, \xi_k) - \hat{F}(z^k, \xi_k) - \hat{P}_{z^k}(\bar{z}^k, \xi_k') + (1 - \alpha_k)(s^{k-1} - \hat{P}_{z^{k-1}}(z^{k-1}, \xi_k) + \hat{F}(z^{k-1}, \xi_k) - \hat{P}_{z^{k-1}}(\bar{z}^{k-1}, \xi_{k-1}')).$$

Notice how the latter term is evaluated under $\xi_{k-1}'$ instead of $\xi_k'$. The choice in (8.4a) resolves this issue.

# G    NEGATIVE WEAK MINTY VARIATIONAL INEQUALITY

In this section we consider the problem of finding a zero of the single-valued operator $F$ (with the set-valued operator $A \equiv 0$). Observe that the weak MVI in Assumption I*(iii)*, $\langle Fz, z - z^\star \rangle \geq \rho\|Fz\|^2$, for all $z \in \mathbb{R}^n$, is not symmetric and one may instead consider that the assumption holds for $-F$. As we will see below this simple observation leads to nontrivial problem classes extending the reach of extragradient-type methods both in the deterministic and stochastic settings.

**Assumption VIII** (negative weak MVI). *There exists a nonempty set $\mathcal{S}^\star \subseteq \mathbf{zer}\, T$ such that for all $z^\star \in \mathcal{S}^\star$ and some $\bar{\rho} \in (-\infty, 1/2L)$*

$$\langle Fz, z - z^\star \rangle \leq \bar{\rho}\|Fz\|^2, \quad \text{for all } z \in \mathbb{R}^n. \tag{G.1}$$

Under this assumption the algorithm of Pethick et al. (2022) leads to the following modified iterates:

$$\bar{z}^k = z^k + \gamma_k F z^k, \tag{G.2}$$

$$z^{k+1} = z^k + \lambda_k \alpha_k (H_k \bar{z}^k - H_k z^k) = z^k + \lambda_k \alpha_k \gamma_k F \bar{z}^k, \quad \text{where} \quad H_k := \mathrm{id} + \gamma_k F \tag{G.3}$$

We next consider the lower bound example of (Pethick et al., 2022, Ex. 5) to show that despite the condition for weak MVI being violated for $b$ smaller than a certain threshold, the negative weak MVI in Assumption VIII holds for any negative $b$ and thus the extragradient method applied to $-F$ is guaranteed to converge.

**Example 1.** Consider (Pethick et al., 2022, Ex. 5)

$$\underset{x \in \mathbb{R}}{\textbf{minimize}}\, \underset{y \in \mathbb{R}}{\textbf{maximize}}\, f(x, y) := axy + \frac{b}{2}(x^2 - y^2), \tag{G.4}$$

where $b < 0$ and $a > 0$. The associated $F$ is a linear mapping. For a linear mapping M, Assumption VIII holds if

$$\tfrac{1}{2}(M + M^\top) - \bar{\rho}M^\top M \preceq 0, \qquad \bar{\rho} \in (-\infty, 1/2L)$$

While Assumption I*(iii)* holds if

$$\tfrac{1}{2}(M + M^\top) - \rho M^\top M \succeq 0, \qquad \rho \in (-1/2L, \infty).$$

For this example $L = \sqrt{a^2 + b^2}$ and

$$F(z) = \overbrace{(bx + ay, -ax + by)}^{=:Mz}.$$

Since $M$ is a bisymmetric linear mapping, $M^\top M = (a^2 + b^2)I$ which according to the above characterizations implies

$$\rho \in (-\tfrac{1}{2L}, \tfrac{b}{a^2+b^2}], \qquad \bar{\rho} \in [\tfrac{b}{a^2+b^2}, \tfrac{1}{2L}).$$

The range for $\rho$ is nonempty if $b > -\frac{a}{\sqrt{3}}$ while this is not an issue for $\bar{\rho}$ which allows any negative $b$.

We complete this section with a corollary to Theorem 6.3 when replacing weak MVI assumption with Assumption VIII.

**Corollary G.1.** *Suppose that Assumptions I(i) and I(ii), Assumptions II, III and VIII hold. Let $(z^k)_{k\in\mathbb{N}}$ denote the sequence generated by Algorithm 1 applied to $-F$. Then, the claims of Theorem 6.3 hold true.*

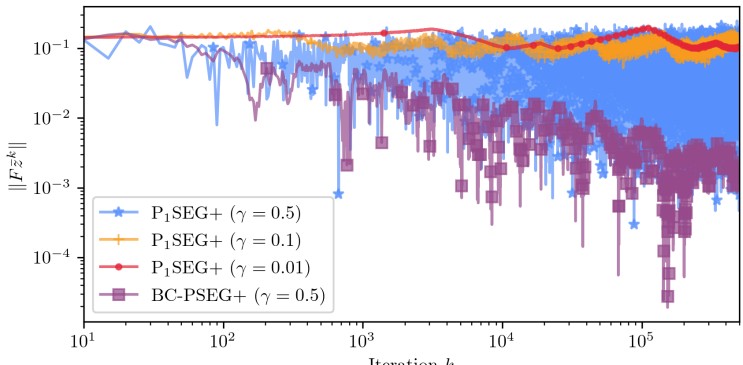

**Figure 3:** *The (projected)* (SEG+) *method needs to take $\gamma$ arbitrarily small to guarantee convergence to an arbitrarily small neighborhood. We show an instance satisfying the weak MVI where $\gamma$ cannot be taken arbitrarily small. The objective is $\psi(x, y) = \phi(x - 0.9, y - 0.9)$ under box constraints $\|(x, y)\|_\infty \leq 1$ with $\phi$ from Example 2 where $L = 1$ and $\rho = -1/10L$. The unique stationary point $(x^\star, y^\star) = (0.9, 0.9)$ lies in the interior, so even $\|Fz\|$ can be driven to zero. Taking $\gamma$ smaller does* not *make the neighborhood smaller as oppose to the monotone case in Figure 1.*

## H  EXPERIMENTS

### H.1  SYNTHETIC EXAMPLE

**Example 2** (Unconstrained quadratic game (Pethick et al., 2022, Ex. 5)). Consider,

$$\underset{x \in \mathbb{R}}{\textbf{minimize}} \; \underset{y \in \mathbb{R}}{\textbf{maximize}} \; \phi(x, y) := axy + \frac{b}{2}x^2 - \frac{b}{2}y^2, \tag{H.1}$$

where $a \in \mathbb{R}_+$ and $b \in \mathbb{R}$.

The problem constants in Example 2 can easily be computed as $\rho = \frac{b}{a^2+b^2}$ and $L = \sqrt{a^2 + b^2}$. We can rewrite Example 2 in terms of $L$ and $\rho$ by choosing $a = \sqrt{L^2 - L^4\rho^2}$ and $b = L^2\rho$.

**Example 3** (Constrained minimax (Pethick et al., 2022, Ex. 4)). Consider

$$\underset{|x| \leq 4/3}{\textbf{minimize}} \; \underset{|y| \leq 4/3}{\textbf{maximize}} \; \phi(x, y) := xy + \psi(x) - \psi(y), \tag{GlobalForsaken}$$

where $\psi(z) = \frac{2z^6}{21} - \frac{z^4}{3} + \frac{z^2}{3}$.

In both Example 2 and Example 3 the operator $F$ is defined as $Fz = (\nabla_x \phi(x, y), -\nabla_y \phi(x, y))$.

To simulate a stochastic setting in all examples, we consider additive Gaussian noise, i.e. $\hat{F}(z, \xi) = Fz + \xi$ where $\xi \sim \mathcal{N}(0, \sigma^2 I)$. We choose $\sigma = 0.1$ and initialize with $z^0 = 1$ if not specified otherwise. The default configuration is $\gamma = 1/2L_F$ with $\alpha_k = 1/18 \cdot (k/c+1)$, $c = 100$ and $\beta_k = \alpha_k$ for diminishing stepsize schemes and $\alpha = 1/18$ for fixed stepsize schemes. We make two exceptions: Figure 1 uses the slower decay $c = 1000$ when $\gamma = 0.1$ and Figure 3 uses $c = 5000$ for $\gamma = 0.01$ (and otherwise $c = 1000$) to ensure fast enough convergence. When the aggressive stepsize schedule is used then $\alpha_k = 1/18 \cdot \sqrt{k/100+1}$.

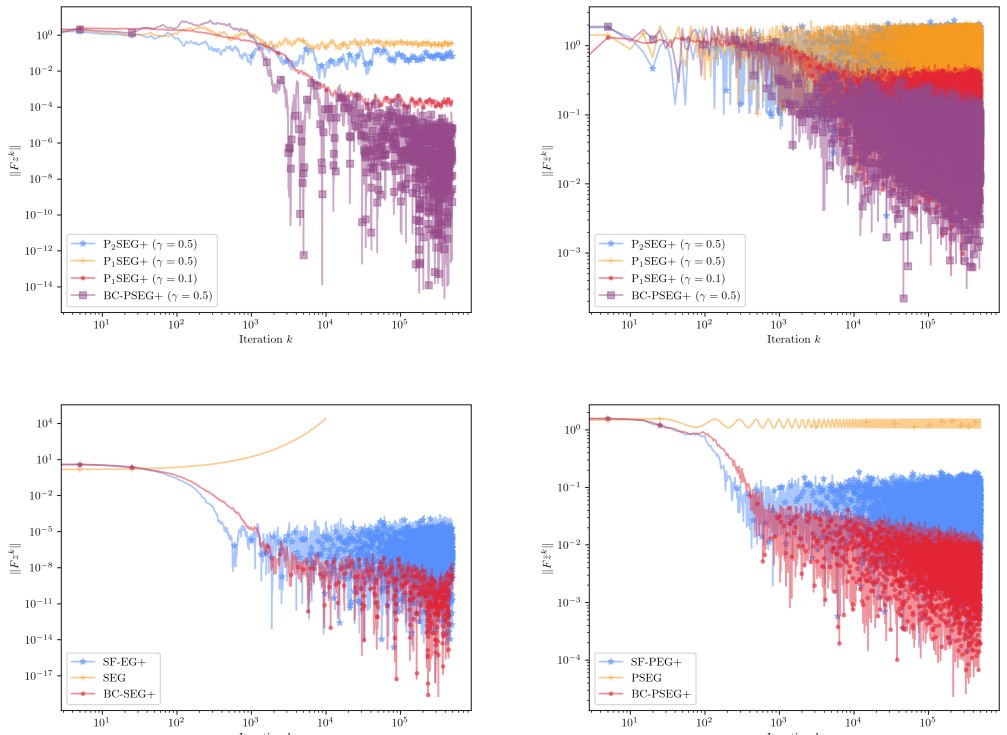

**Figure 4:** *Instead of taking $\alpha_k \propto 1/k$ (for which almost sure convergence is established through Theorem 6.3 and ??) we take $\alpha_k \propto 1/\sqrt{k}$ as permitted in Theorems 6.1 and 7.1. We consider the example provided in Figure 1 (top row) and the two examples from Figure 2 (bottom row). Under this more aggressive stepsize schedule the guarantee is only in expectation over the iterates which is also apparent from the relatively large volatility in comparison with Figures 1 and 2.*

## H.2 ADDITIONAL ALGORITHMIC DETAILS

For the constrained setting in Figure 1, we consider two extensions of (SEG+). One variant uses a single application of the resolvent as suggested by Pethick et al. (2022),

$$\bar{z}^k = (\mathrm{id} + \gamma A)^{-1}(z^k - \gamma \hat{F}(z^k, \xi_k)) \quad \text{with} \quad \xi_k \sim \mathcal{P}$$
$$z^{k+1} = z^k + \alpha_k\big((\bar{z}^k - z^k) - \gamma(\hat{F}(\bar{z}^k, \bar{\xi}_k) - \hat{F}(z^k, \xi_k))\big) \quad \text{with} \quad \bar{\xi}_k \sim \mathcal{P} \qquad (P_1\text{SEG+})$$

The other variant applies the resolvent twice as in stochastic Mirror-Prox (Juditsky et al., 2011),

$$\bar{z}^k = (\mathrm{id} + \gamma A)^{-1}(z^k - \gamma \hat{F}(z^k, \xi_k)) \quad \text{with} \quad \xi_k \sim \mathcal{P}$$
$$z^{k+1} = (\mathrm{id} + \alpha_k\gamma A)^{-1}(z^k - \alpha_k\gamma \hat{F}(\bar{z}^k, \bar{\xi}_k)) \quad \text{with} \quad \bar{\xi}_k \sim \mathcal{P} \qquad (P_2\text{SEG+})$$

When applying (SEG) to constrained settings we similarly use the following projected variants:

$$\bar{z}^k = (\mathrm{id} + \beta_k\gamma A)^{-1}(z^k - \beta_k\gamma \hat{F}(z^k, \xi_k)) \quad \text{with} \quad \xi_k \sim \mathcal{P}$$
$$z^{k+1} = (\mathrm{id} + \alpha_k\gamma A)^{-1}(z^k - \alpha_k\gamma \hat{F}(\bar{z}^k, \bar{\xi}_k)) \quad \text{with} \quad \bar{\xi}_k \sim \mathcal{P} \qquad (\text{PSEG})$$

and (EG+) (using stochastic feedback denoted SF)

$$\bar{z}^k = (\mathrm{id} + \gamma A)^{-1}(z^k - \gamma \hat{F}(z^k, \xi_k)) \quad \text{with} \quad \xi_k \sim \mathcal{P}$$
$$z^{k+1} = (\mathrm{id} + \alpha\gamma A)^{-1}(z^k - \alpha\gamma \hat{F}(\bar{z}^k, \bar{\xi}_k)) \quad \text{with} \quad \bar{\xi}_k \sim \mathcal{P} \qquad (\text{SF-PEG+})$$

which we in the unconstrained case ($A \equiv 0$) refer to as (SF-EG+) as defined below.

$$\bar{z}^k = z^k - \gamma \hat{F}(z^k, \xi_k) \quad \text{with} \quad \xi_k \sim \mathcal{P}$$
$$z^{k+1} = z^k - \alpha\gamma \hat{F}(\bar{z}^k, \bar{\xi}_k) \quad \text{with} \quad \bar{\xi}_k \sim \mathcal{P} \qquad (\text{SF-EG+})$$

# I    COMPARISON WITH VARIANCE REDUCTION

Consider the case where the expectation comes in the form a finite sum,

$$Fz = \frac{1}{N} \sum_{\xi=1}^{N} \hat{F}(z, \xi). \tag{I.1}$$

In the worst case the averaged Lipschitz constant $F_{\hat{F}}$ scales proportionally to the number of elements $N$ squared, i.e. $L_{\hat{F}} = \Omega(\sqrt{N}L_F)$. It is easy to construct such an example by taking one elements to have Lipschitz constant $NL$ while letting the remaining elements have Lipschitz constant $L$. Recalling the definition in Assumption III, $L_{\hat{F}}^2 = \frac{N^2 L^2}{N} + \frac{N-1}{N}L^2 \geq NL^2$ while the average becomes $L_F = \frac{NL}{N} + \frac{N-1}{N}L \leq 2L$ so $L_{\hat{F}} \geq \sqrt{N}/2L_F$. Thus, $L_{\hat{F}}$ can be $\sqrt{N}$ times larger than $L_F$, leading to a potentially strict requirement on the weak MVI parameter $\rho > -L_{\hat{F}}/2$ for variance reduction methods.

