# OpenReview forum: "Solving stochastic weak Minty variational inequalities without increasing batch size"
_ICLR.cc/2023/Conference — ICLR 2023 poster_

### Official Review · Reviewer_6PPj · 2022-10-25

**Confidence:** 4
**Correctness:** 3
**Technical Novelty And Significance:** 4
**Empirical Novelty And Significance:** 2
**Recommendation:** 8

**Clarity, Quality, Novelty And Reproducibility:**

**Clarity.** As I explained above, the clarity should be improved.

**Quality.** Although there are some small inaccuracies in the proofs and problems with clarity, the quality of the paper is high.

**Novelty.** The results are novel. I emphasize that the results are novel even for monotone case.

**Reproducibility.** The authors should provide the exact values of stepsizes that they used for each method.

**Strength And Weaknesses:**

## Strengths
1. **Strong theoretical results.** The open question that authors resolved (analysis in terms of the expected squared norm of the operator without large batchsizes) is highly non-trivial. In particular, even in the monotone case, all previous works that have convergence results in terms of the expected squared norm of the operator do rely on the usage of large $\mathcal{O}(1/\varepsilon)$ batchsizes (or even worse) to achieve $\min_{k = 0,\ldots,K}\mathbb{E}||F(z^k)||^2 \leq \varepsilon$. In contrast, the analysis of the method proposed in this work achieves the same complexity result *with $\mathcal{O}(1)$* batchsize. Moreover, it is done for weak MVIs with Lipschitz operator, which is a more general setup than monotone Lipschitz case. Even this result makes the paper strong enough to be accepted to the conference. However, the authors do not stop here and generalize their results (and algorithm) to the constrained case and show some interesting (though very hard to read) extensions.

## Weaknesses
1. **Clarity.** Some parts of the paper are very hard to understand. In particular, it is hard to follow some parts of the proofs (see my detailed comments below). Next, the paper does not provide the intuition behind the algorithm development: why does the bias-correction term was chosen this way? Moreover, Section 8 and Appendix E are very hard to follow due to the lack of the intuition and details. The authors introduce too many notations there, I cannot thoroughly check this part during the time given for reviewing. Therefore, I encourage the authors either to remove Section 8 from the paper or substantially improve the clarity there (why we consider such preconditioners, why we change the algorithms this way, what is the intuition behind the proof).

2. **Strong assumptions.** The key trick allowing the authors to get rid of the large batchsizes is that the authors assume bounded variance and Lipschitzness in mean at the same time (see derivation of formula (C.4)). Usually, these assumptions are not used simultaneously: either bounded variance is assumed without any assumptions like Assumption III or Assumption III is used without bounded variance assumption (instead one needs to assume just boundedness of the variance at the optimum). I think, this should be highlighted more in the paper and listed as a limitation of the derived results.

3. **Numerical experiments.** Based on what is written in the paper, the authors used similar stepsizes for all tested methods. However, this is not a fair comparison: one should tune stepsizes for each method separately. Next, the proposed method was not tested on real problems (minor). It would be interesting to see how BCSEG+ behaves in training GANs, for example.


## Main concerns/comments/questions (this is very important to address)

1. The authors write "When monotone-like conditions holds the story is different since it suffices
to take diminishing stepsizes." I believe, this is not accurate. All mentioned results in the monotone case either provide guarantees in terms of the gap function (and thus do not need large batchsizes) like Juditsky et al. (2011) and Mishchenko et al. (2020) or in terms of  the squared norm of the operator but with large batchiszes like in Gorbunov et al. (2022). Other works consider setups where one can achieve linear rates in deterministic case (and thus these cases are easier). Therefore, the question that the authors resolved was in fact open even in the monotone case.

2. Could the authors explain why the metric chosen in the constrained case is good? How does it relate to the standard squared residual metric like $||z^k - z^{k-1}||^2$?

3. Figure 1 does not imply that SEG+ does not converge in general: it implies only that it does not converge for given stepsize policy.

4. Figure 2: why EG+ does not converge to any predefined accuracy? Since EG+ is a deterministic method, it should converge with fixed stepsizes.

**Below I list the issues with the proofs to justify my claim about the clarity.**

5. In (B.7), the derivation is not accurate: one should have extra $-||v||^2$ in the second step.

6. In (C.1), $b$ is not defined.

7. Why the potential function $\mathcal{U}_{k+1}$ is chosen this way? How was it found?

8. The formula above (C.4), after the word "Therefore": in the first term of the inner product, one should have $F(z^{k-1})$ instead of $\hat F(z^{k-1})$.

9. Formula (C.6), third row from below" one should have $+ \frac{\gamma^2 \varepsilon_k}{2}||F(\bar{z}^k)||^2$ instead of $- \frac{\gamma^2 \varepsilon_k}{2}||F(\bar{z}^k)||^2$.

10. The end of the proof of Theorem C.2 is very hard to follow (after (C.11)). I have checked it, everything is correct. However, it required me to derive few things myself. The authors should provide all missing derivations with the detailed comments on each step.

11. In (C.16), one should have extra factor $2$ in front of the second $\gamma L_{\hat{F}}$ (it goes from (C.27), where in the second row one should have $\frac{2}{k+1+r}$ in the second multiplicative factor).

12. (C.22) should go before (C.21).

13. The part after (C.23) is very hard to check: the authors should refer to particular inequalities that they use in each step and also explain each step in details.

14. Page 18: "Plugging the value of $c_\ell$ and $\varepsilon_k$ from (C.21)" --> "Plugging the value of $c_\ell$ and $\varepsilon_k$ from Assumption VI and (C.20)"

15. The first inequality on page 20: the authors use $||a+b||^2 \geq ||a||^2 - ||b||^2$, which is not correct. The correct version: $||a+b||^2 \geq \frac{1}{2}||a||^2 - ||b||^2$. However, in this case, one should assume that $\gamma \leq \frac{1}{\sqrt{2}L}$ that reduces the maximal range for $\rho$. To avoid this one should estimate $||Hz - Hz'||$ instead of $||Hz - Hz'||^2$. In this case, one can use the triangle inequality instead of Young. As the result, the authors will get $(1 - \gamma L_F)^2$ factor instead of $1 - \gamma^2 L_F^2$.

16. I do not understand the derivation of (D.13). In particular, I do not see what left-hand side of (D.14) corresponds to. The authors should provide complete derivation with all the details.

17. How (D.15) was obtained? The authors should provide complete derivation with all the details.

## Minor comments

1. Assumption II(ii): the second term inside the norm should be $F(z)$.

2. "However, Pethick et al. (2022) also showed that the extrapolation stepsize plays a critical role for convergence under weak MVI." Could the authors provide a reference to the particular result from Pethick et al. (2022)?

3. Page 13, above (B.3): "The two latter term are" --> "The two latter terms are"

4. Page 15, the first sentence: "By step 1.2" --> "By step 1.4"

5. Sometimes the authors use $F(\bar{z}^k)$, sometimes they use $F\bar{z}^k$. The notation should be verified.

6. The end of page 16: "... is sufficient to satisfy (C.13)" -- > "... is sufficient to satisfy (C.13) and (C.14)"

7. Page 20, "The third term in (D.2) is bounded by": it is not the third term but its conditional expectation. One should fix it here and everywhere else.

8. In (D.9), $H_k(\bar{z}^k)$ is not defined. Is it $H(\bar{z}^k)$?

9. Above (D.18): one should have $\Gamma = I$.

**Summary Of The Paper:**

The paper addresses an important open question of deriving rates in terms of the expected squared norm of the operator for stochastic weak MVIs without large batchsizes. The result is novel even in the special case of monotone operators. The authors also propose the generalization to the constrained case and propose and analyze the variants with a nonlinear asymmetric preconditioner. The key algorithmic novelty is a modification of Stochastic Extragradient Method that uses bias correction term.

**Summary Of The Review:**

The main results are very good and outweigh multiple weaknesses that the paper has. I strongly encourage the authors to address the issues I mentioned. It will improve the paper a lot. If the authors address my concerns, I will increase my score.


========UPDATE==========

I thank the authors for the detailed response to my comments and for improving the paper. The problematic parts became much easier to follow (including Section 8) and almost all of my questions/concerns are resolved. In general, I am satisfied with the changes. Therefore, I am increasing my score from 6 to 8: as I explained in the original review, the paper fully deserves this.

---

> ### Author Response · Authors · 2022-11-12
> **Response to Reviewer 6PPj (Part I)**
>
> We thank the reviewer for a very thorough and positive review. We address all comments below.
>
> Main concerns:
>
> - **Contribution is important even in the monotone case** We thank the reviewer for bringing this to our attention. We have added more accurate context under related work (Section 2) and clarified this additional contribution in the introduction.
> - **Motivation for metric in constrained case** In the deterministic setting, (Pethick 2022) relies on $Hz^k-H\bar z^k \in \gamma (A\bar z^k+F\bar z^k)$. So convergence is effectively given for $\operatorname{dist}(0, A\bar z^k+F\bar z^k)$ (i.e. a zero of the operators). We fully agree with the reviewer that the previous metric could be considered opaque and have translated all the theorems in terms of $\operatorname{dist}(0, A\bar z^k+F\bar z^k)$. Please see theorem 7.1 and thm 8.2 and the end of their respective proofs for the conversion. (We have defined $\operatorname{dist}$ in the newly added "Preliminaries" section in appendix B together with basic definitions from operator theory.)
> - **Figure 1 is only for one stepsize choice of SEG+** We can essentially scale the current counterexample for any choice of $\gamma$ in SEG+. To see this, we choose $\phi(x,y)=(x-\varepsilon)(y-\varepsilon)$ such that the Lipschitz constant $L=1$ for any $\varepsilon$. Then we are given SEG+ with any arbitrary small (but fixed) stepsize $\gamma<1$. Only then do we pick the size of the box constraints and $\varepsilon$. If we pick them small enough then the construction will be identical up to a scaling and SEG+ will face the same issue. To make nonconvergence more apparent we have updated Figure 1 to include more iterations, where the separation from BCSEG+ becomes very obvious.
> - **Why does EG+ not converge in Figure 2?** EG+ also uses the stochastic oracles in our experiment so it is not deterministic.  We have now mentioned this explicitly in section 9 as well as explicitly defined the update rule in appendix H.2. We compare with EG+ (with stochastic feedback) since it also provably converge (while only to a neighborhood). We have updated Figure 2 with more iterations to make the failure to converge to an arbitrary small error more apparent. **We have also updated the naming in the experimental section** to make it clear that stochastic oracles are used and when projections are used.
>
> Additional comments:
>
> - **Clarity of proofs and section 8** We have made substantial modifications to address the concern with clarity. First of all we have explained the reasoning behind each step (including addressing the detailed feedback that the reviewer provided, which we respond to below). Section 8 has been updated to better explain the update. The proofs have been reorganized by extracting all the deterministic results into their own lemmas (see Lemma E.1 and Lemma F.2-F.3 specifically). This should make the logic of the proof much clearer.
> - **Why both Lipschitz and Lipschitz in mean?** We would like to clarify that mean square Lipschitz assumption implies Lipschitz continuity of $F$ through Jensen’s inequality, so we do not strictly need to assume Lipschitz of $F$ explicitly. We make both assumption to make the dependency more clear in the derived theorems.
> We have made it clear under contributions that we rely on this stronger assumption.
> - **Tune stepsizes** We are not interested in comparing the optimal rates (for which we would have to finetune the stepsize choices), but rather show convergence vs nonconvergence. We have increased the number of iterations in Figure 1 and Figure 2 substantially to make the separation between the method clearer.
> - **Reproducability** For exact stepsize choices please see appendix H.1.
>
>
> Proof typos:
>
> 1. Thanks for catching the typo, the mistake is in the equality, which should have read $\mathbb E_k\|\gamma B\zeta_k\|^2 = \gamma^2 \mathbb E_k\|F(\zeta_k)-F(0)\|^2$. This should make the following inequality correct.
> 2. $b$ is now defined.
> 3. Like many Lyapunov functions it arises from the analysis, but to provide some intuition: The $u^k=\bar z^k - (z^k - \gamma F(z^k))$ term measures how far the bias corrected step is from a deterministic extrapolation step. This is ultimately because the analysis relies on tracking the deterministic EG+ scheme. We have clarified this in the updated manuscript.
> 4. \hat typo has been fixed
> 5. Sign fixed
> 6.  We have provided details of the derivations
> 7.  The factor 2 is fixed. (we also noticed that it should be $L_F^2 L_{\hat F}^2$ instead of $L_F^4$)
> 8.  We have swapped the equations
> 9.  We have added details for the step
> 10. The Crefs has been updated
> 11. We have updated the constant $(1-\gamma^2L_F^2)$ to $(1-\gamma L_F)^2$. This result now follows through the deterministic lemmas (see Lemma E.1(ii) and Lemma F.2(ii)).
> 12. (and 17.) We have specified all steps and explicitly stated their dependencies.

---

> > ### Author Response · Authors · 2022-11-12
> > **Response to Reviewer 6PPj (Part II)**
> >
> > We would also like to address the minor comments:
> >
> > 1. Fixed
> > 2. The importance of the extrapolation stepsize appears in multiple places in (Pethick 2022). In Theorem 3.1 it appears through the stepsize requirement $\gamma_k > \max\{0, -2\rho\}$, suggesting that the stepsize needs to be large enough. In section 4 they exploit this relationship by adaptively picking the stepsize larger than $1/L$ using backtracking linesearch.
> > 3. Fixed
> > 4. Fixed
> > 5. We have added an overview of common notation for monotone operators in appendix B including $Fz=F(z)$.
> > 6. We still need to assume C.14 as it appears (in simplified form) in (6.3) of Theorem 6.3.
> > 7. Fixed
> > 8. Yes $H_k(\bar z^k)$ should have been $H\bar z^k$. It has now been fixed.
> > 9. Fixed

---

> > > ### Comment · Reviewer_6PPj · 2022-11-16
> > > **Reply to authors**
> > >
> > > I thank the authors for the detailed response to my comments and for improving the paper. The problematic parts became much easier to follow (including Section 8) and almost all of my questions/concerns are resolved. In general, I am satisfied with the changes. Therefore, I am increasing my score from 6 to 8: as I explained in the original review, the paper fully deserves this.
> > >
> > > However, I still have few minor comments/questions. I hope that the authors will have a chance to reply to them and modify the paper accordingly.
> > >
> > > 1. **On strong assumptions.** I believe, there is a misunderstanding. In my review, I mean that it is restrictive to assume both bounded variance (Assumption II (iii)) + Lipschitzness in mean (Assumption III). As I explained in my original review, I believe that this is the key trick allowing the authors to get rid of the large batchsizes (see the second point in the "Weaknesses" section). In contrast, other works either assume bounded variance without Lipschitzness in mean (Jusidtsky et al., 2011) or Lipschitzness in mean/almost surely without bounded variance (Mishchenko et al., 2020; Gorbunov et al., 2022). I believe that such a discussion about the differences between the setups will improve the paper and help the reader to clearly see the difference.
> > >
> > > 2. **Regarding the experiments.** It is still unclear why SEG+ does not converge for the bilinear problem. I guess the main reason is in the choice of the stepsizes. The authors write that they are interested whether the method converges or not, not in the tuning. However, currently the answer to this question is unclear from the experiments: SEG+ achieves some value of the operator norm and then it stops. This experiment is conducted with fixed stepsize as far as I get (at least for Figure 1). Such a behavior is completely natural, we know it from the standard stochastic optimization literature. *However, it does not mean that the method will not achieve better accuracy with the smaller stepsize, i.e., it does not mean that SEG+ is not converging.* To numerically verify the authors' claim, one needs to show that SEG+ does not improve the accuracy of the solution when the stepsizes are reduced. Only in this case one can conclude that SEG+ is not converging. That is why I asked the authors to tune the stepsizes: this is the way to check whether methods converge or not. I believe this is perfectly aligned with the authors' goal to check whether SEG+ converges without large batchsizes or not.
> > >
> > > I am also curious why the oscillations of BC-PSEG+ in Figure 1 are so significant?
> > >
> > > 3. The firs step in (E.2): I guess one should have $\gamma Fz - \gamma Fz'$ in the inner product.
> > >
> > > 4. The first inequality "0 \leq ..." in (E.16): I believe it should not be there, since the goal is to show the opposite.
> > >
> > > 5. **Regarding the claim about the result from (Pethick et al., 2022).** Theorem 3.1 from (Pethick et al., 2022) does not say that $\gamma$ has to be large to have convergence. It says only that if the extrapolation stepsize is large enough (satisfies a certain inequality), then the method converges. To the best of my knowledge, there is no result saying that if $\gamma$ does not satisfy this condition, then the method diverges.

---

> > > > ### Author Response · Authors · 2022-11-17
> > > > **Response to 6PPj**
> > > >
> > > > We would like to thank the reviewer for engaging so thoroughly with both our paper and the rebuttal. We hope the following addresses all remaining concerns:
> > > >
> > > > 1. **On strong assumptions** The fault is entirely on our side for engaging in the wrong question. To answer your question about bounded variance: (Mishchenko et al. 2020, Thm. 2) indeed provides result for relaxing bounded variance, but crucially requires the nonsmooth part $g$ to be strongly convex. When this condition is dropped in Theorem 3 (i.e. $g$ is only convex corresponding to monotone $A$ in our case) they still rely on bounded variance. The question of relaxing the variance is thus orthogonal to our work and is partially a matter of further restricting $A$ (and additionally assuming that the stochastic oracles are monotone). Concerning (Gorbunov et al., 2022) it is indeed interesting to see if the same-sample and monotone setting considered in (Gorbunov et al., 2022, Corollary F.2) can avoid the large batch size by introducing the bias-correction term. Similarly, in the independent-sample setting (Gorbunov et al., 2022, Corollary E.2), whether bias-correction works under the alternative variance control condition in Assumption 4.1. In light of this, we believe that it can be misleading to state that our result is heavily dependent on the interplay between bounded variance and Lipschitz in mean as the reviewer suggests. We still agree that it is important to state the Lipschitz in mean assumption up front, which we have kept stated under contributions.
> > > > 3. **Convergence to neighborhood for SEG+** What we intended to demonstrate with Figure 1 is that SEG+ only converges to a neighborhood dictated by the fixed $\gamma$ (independent of the horizon $K$). This is problematic if we cannot take $\gamma$ arbitrarily small as required for weak MVIs. We have now demonstrated this requirement in Figure 3 of appendix H where SEG+ cycles for small values of $\gamma$. We have additionally added a smaller choice of $\gamma$ for SEG+ in Figure 1 to demonstrate that convergence to a smaller neighborhood for the bilinear case is indeed possible. Section 5 have also been generally revised to make the argument clearer.
> > > > - **Oscillations of BC-PSEG+** The oscillation seems to be due to the aggressive stepsize $\alpha_k \propto 1/\sqrt{K}$ which Thm 6.1/Thm 7.1 permit. We have reproduced all the experiments with the stepsize choice $\alpha_k \propto 1/K$ instead (which gaurantees almost sure convergence). The results can be found in Figure 4 of Appendix H, where the oscillations becomes significantly less pronounced.
> > > > 3. Fixed
> > > > 4. Fixed
> > > > 5. **No divergence result for small $\gamma$** We have removed the misleading statement from Section 4 and rephrased in the introduction. Please see the newly added Figure 3 in appendix H for an experimental result where SEG+ with small $\gamma$ cycles.

---

> ### Comment · Reviewer_6PPj · 2022-11-16
> **Thank you for the detailed response, I will reply soon**
>
> I thank the authors for the detailed reply and taking into account my comments. Right now I am doing my pass through the authors' response and the updated manuscript. I will reply as soon as possible.

---

### Official Review · Reviewer_o1sj · 2022-10-27

**Confidence:** 3
**Correctness:** 3
**Technical Novelty And Significance:** 3
**Empirical Novelty And Significance:** 2
**Recommendation:** 6

**Clarity, Quality, Novelty And Reproducibility:**

Novelty: the algorithm is original.
Clarity: the paper is not hard to follow, but can be improved.

**Strength And Weaknesses:**

Strength:

(1) The design of the algorithm is interesting, as it only uses diminishing stepsize for in one step of EG update and introduces a novel correction term.

I have the following questions:

(1) The paper mentions the lwoer bound $\rho > \gamma/2$ from [Pethick et al., 2022] several times and it serves an intuition for the algorithm in page 4. However, based on my understanding, from the example given in  [Pethick et al., 2022], the stepsize $\gamma$ is fixed to be $1/L$. I am not sure whether that still holds for  other stepsize or time-changing stepsize.

(2) In Theorem 7, the convergence measurement $|| H\bar{z}^{k\star} - Hz^{k\star}||$ seems to only consider the operator F and ignore A by the definition of $H$. Why is it a good measurement here? Also in Algorithm 2, it returns $z^{k+1}$, but I do not know how it will guarantee that it will satisfy the constraint if operator A corresponds to a constraint.

(3) The experiment in Figure 1 is not representative. It is a bilinear game, so it can be easily solved by stochastic EG.

**Summary Of The Paper:**

The paper provides a stochastic algorithm for a class of problem characterized by weak minty variational inequality. The algorithm modifies stochastic extra-gradient by adding a bias-correction term in the exploration step.

**Summary Of The Review:**

I find the algorithm interesting, but a few questions remain.

---

> ### Author Response · Authors · 2022-11-12
> **Response for Reviewer o1sj**
>
> We thank the reviewer for the feedback and address all concerns below.
>
> 1. **Motivation for large stepsize and adaptive stepsize** The main intuition behind the large stepsize choice in not so much through the lower bound but rather the convergence result in (Pethick 2022, Thm. 3.1) which holds for positive $\gamma_k > -2\rho$ (see the stepsize condition in the theorem). Notice that this condition actually allows adaptive stepsize which they further exploit in section 4. The corresponding stepsize condition in the stochastic case appears in (6.1) of Thm 6.1 of our work, where it would indeed be possible to replace the $\gamma$ with $\gamma_k$. However, we have refrained from this, since the adaptive stepsize selection is nontrivial in the stochastic case (e.g. linesearch as in (Pethick 2022) relies on access to the deterministic $F$).
> 2. **Motivation for metric in constrained case** In the deterministic setting, (Pethick 2022) relies on $Hz^k-H\bar z^k \in \gamma (A\bar z^k+F\bar z^k)$. So convergence is effectively given for $\operatorname{dist}(0, A\bar z^k+F\bar z^k)$ (i.e. a zero of the operators). We fully agree with the reviewer that the previous metric could be considered opaque and have translated all the theorems in terms of $\operatorname{dist}(0, A\bar z^k+F\bar z^k)$. Please see theorem 7.1 and thm 8.2 and the end of their respective proofs for the conversion. (We have defined $\operatorname{dist}$ in the newly added "Preliminaries" section in appendix B together with basic definitions from operator theory.)
> 3. **Gaurantees for $z^{k+1}$?** In terms of the iterates, $\|z^{k+1}-z^\star\|$ is still asymptotically going to zero. We have provided formal proof of this in Thm. F.5 of the updated manuscript, which uses a very similar argument as for the smooth unconstrained case of Thm. 6.3. This result is intuitively possible because $\alpha_k$ is going to zero. However the reviewer is correct that the rate are provided in terms of the projected iterates $\bar z^k$ through $\operatorname{dist}(0, A\bar z^k+F\bar z^k)$.
> 4. **SEG can solve the constrained bilinear game**  Actually, even SEG fails in the unconstrained bilinear case in the last iterate sense (see (Hsieh et al., 2020, Fig. 1) which we referenced in the introduction). The reviewer is right though that the *average* would converge, but the point of the construction is to show a minimal failure case for *SEG+*, which, maybe surprisingly, can happen already in the monotone case. The counterexample shows that the result in (Hsieh 2020, Thm. 5(2)) (which also uses a fixed exploration stepsize) are thus quite specific, since the result does not extend beyond the unconstrained case even for bilinear games.

---

> > ### Comment · Reviewer_o1sj · 2022-12-07
> > **Minor suggestions**
> >
> > I am satisfied with most of the response and I have already updated my score.
> >
> > A few minor suggestions and questions:
> >
> > 1. I still suggest to modify the paragraphs in page 4, where it kinda suggests diminishing stepsize in SEG is not possible in this setting. However, based on my understanding, there is no theoretical evidence or lower bound.
> >
> > 2. With weak MVI condition, is it possible to have average iterate or last iterate convergence instead of random iterate convergence?

---

> > > ### Author Response · Authors · 2022-12-08
> > > **Response for Reviewer o1sj**
> > >
> > > We thank the reviewer for the follow up questions.
> > >
> > > 1. **Diminishing stepsize** Please see Figure 3 in Appendix H which illustrates an instance of weak MVI where $\gamma$ cannot be taken arbitrarily small. It is true though that there is no lower bound established in the literature (to our knowledge) which characterizes the relationship. We will update the camera ready version to make this clear.
> > > 2. **Last iterate** We show asymptotic convergence of the last iterate (in terms of $\|z^{k+1} - z^\star\|$) in Theorem 6.3 and Theorem F.6, but it indeed still remains open to show a rate for norm of the operator at the last iterate.

---

> > > > ### Comment · Reviewer_o1sj · 2022-12-08
> > > > **a minor question**
> > > >
> > > > Thanks for the response! Another minor question: is there logarithmic term in the complexity in Remark 6.2?

---

> > > > > ### Author Response · Authors · 2022-12-09
> > > > > **Response for Reviewer o1sj**
> > > > >
> > > > > There is no logarithmic term but notice that the existing rate in the literature is for the gap function. Our result appears to be the first result even in the monotone case showing $\frac{1}{\sqrt{T}}$ for the squared norm of the operator without increasing the batch size (see also the comment of Reviewer 6PPj).

---

### Official Review · Reviewer_eJWH · 2022-10-28

**Confidence:** 4
**Clarity, Quality, Novelty And Reproducibility:** Please see the above review for furth…
**Correctness:** 3
**Technical Novelty And Significance:** 4
**Empirical Novelty And Significance:** 2
**Recommendation:** 8

**Strength And Weaknesses:**

The paper is well-written and the idea is easy to follow. The authors did a great job on the separation of sections and on the presentation of the results. In particular, I find that it is very helpful for the reader that the authors included separate sections for the unconstrained and constrained setting.

However, I believe the paper has some issues in terms of notation and numerical evaluation. In addition, the paper missing some relevant recent works on other classes of structured non-monotone problems.

Let me provide some details below:

On presentation:
1. There is no part in the paper where the sets $zer T$, and $gph T$ are defined. Even if the definition is trivial is currently missing. In addition, what do we call a maximally monotone operator (assumption on operator A in the main problem)? this detail is required for a self-contained paper.

2. Inequality 3.2 used $\Gamma$ and $\Gamma^{-1}$ without a proper explanation of why the  $ \Gamma^{-1}$ is needed. The standard $L_f$-Lipschitz used identity matrices, so further details are needed. The same holds for Assumption III.

3. The paper mentions in several parts that: ``employing diminishing stepsizes is no longer possible in the weak MVI setting." but they do not properly explain why. Why is this true? is there an existing paper that proves that or is it a speculation of the authors? more details are needed.

4. Minor: The deterministic operator is defined as $Fz$ while the stochastic estimator is denoted with $F(z,\xi)$. It might have been better if one used F(z) as well for the deterministic variant.

5. After the definition of SEG, the paper mentions: ``Even with a two-timescale variant (when $\beta_k > \alpha_k$) it has only been possible to show convergence for MVI (Hsieh et al., 2020)."  what this means exactly? note that (Hsieh et al., 2020) has nothing to do with MVI.

On proofs:
1. What is b in equation C.1 in the appendix?
2. I find that the steps in the proofs require further explanation for the reader to be able to follow easily (probably by splitting steps into several parts). The parts where Young's inequality is used are not always very clear.
3. In C.8 the previous bound is used by now it has expectation conditional on $f_k$ even if the quantity is deterministic. This is not wrong but it is not standard.

On experiments: (This is probably an important issue of the current version of the paper)

The authors mentioned the following: "Except for BCSEG+, all methods fail to converge in these examples." and "In comparison (EG+) gets closer to a solution in both problems but fails to converge due to the non-diminishing stepsize, while BCSEG+ converges for both example."
In my viewpoint figure, 2 does not show any benefit of BCSEG+, compared to EG+. both methods converge to a very similar neighborhood $(10^{-2})$ of the solution. Probably the methods should be run for more iterations to obtain something useful.
Also, I suspect that the method here is SEG+ and not EG+, right?

Missing references on other structured non-monotone problems:

[1] Yang, J., Kiyavash, N., and He, N. (2020). Global convergence and variance reduction for a class of nonconvexnonconcave minimax problems. NeurIPS

[2] Song, C., Zhou, Z., Zhou, Y., Jiang, Y., and Ma, Y.
(2020). Optimistic dual extrapolation for coherent nonmonotone variational inequalities. NeurIPS

[3] Loizou, N., Berard, H., Gidel, G., Mitliagkas, I., and
Lacoste-Julien, S. (2021). Stochastic gradient descentascent and consensus optimization for smooth games:
Convergence analysis under expected co-coercivity.NeurIPS

[4] Loizou, N., Berard, H., Jolicoeur-Martineau, A., Vincent,
P., Lacoste-Julien, S., and Mitliagkas, I. (2020). Stochastic hamiltonian gradient methods for smooth games. ICML

[5] Kannan, A. and Shanbhag, U. V. (2019). Optimal
stochastic extragradient schemes for pseudomonotone
stochastic variational inequality problems and their variants. Computational Optimization and Applications,
74(3):779–820.

[6] Aleksandr Beznosikov, Eduard Gorbunov, Hugo Berard, and Nicolas
Loizou. Stochastic gradient descent-ascent: Unified theory and new
efficient methods. arXiv preprint arXiv:2202.07262, 2022


**Summary Of The Paper:**

The paper, proposes new variants of stochastic extra-gradient methods for solving inclusion problems that satisfy the minty variational inequality (MVI). The proposed algorithms BCSEG+ (Alg. 1 and Alg. 2) and NP-PDGG (Alg. 3) have been designed and analyzed for solving the inclusion problem under the unconstrained smooth case, constrained case, and min-max problem (8.5) respectively. The most important contribution of this work is that by using the new algorithms it is possible to provide analysis without requiring increasing batch sizes as the algorithm progress.

**Summary Of The Review:**

I give "6: marginally above the acceptance threshold" for this work.
The theoretical results are novel however there are several points of weakness (in terms of clarity) of the paper and limitations in terms of experiments.

---

> ### Author Response · Authors · 2022-11-12
> **Response to Reviewer eJWH**
>
> We thank the reviewer for the feedback and address all concerns below.
>
> First we would like to clarify a crucial difference between weak MVI and MVI, since we believe there might be a misunderstanding. In the unconstrained case the conditions relates as follows:
>
> - $\langle {Fz-Fz', z-z'} \rangle \geq 0$ (monotone)
> - $\langle {Fz, z-z^\star} \rangle \geq 0$ (MVI) This relaxes the monotone condition by only requiring the condition to hold w.r.t. to the solutions (also called star-monotonicity)
> - $\langle {Fz, z-z^\star} \rangle \geq \rho \|Fz\|^2$ (weak MVI) which relaxes MVI for possible negative $\rho$.
>
> Notice that weak MVI recovers MVI when $\rho=0$. Importantly, (Pethick 2022) shows that attracting limit cycles are possible when $\rho < 0$ which complicates the optimization landscape significantly for weak MVI. In this terminology what (Hsieh et al., 2020) treats is MVIs (so $\rho=0$), which we elaborate further on below.
>
> On presentation:
>
> 1. **Missing definitions** The updated manuscript now defines maximally monotone operators, the graph, the domain and other common notation from monotone operator in section B of the supplementary which we refer to in Section 3.
> 2. **$\Gamma$ is not properly explained under the problem formulation** The matrix $\Gamma$ is only needed for the primal-dual case covered in Section 8. We agree with the reviewer introduces unnecessary complexity in the beginning and the updated manuscript now introduces all assumptions without $\Gamma$. Definitions involving $\Gamma$ have been postponed to Section 8 where we have added further clarification for its role.
> 3. **Justification for why diminishing stepsize is not possible** This is largely motivated by the requirement $\rho > -\gamma_k/2$ in (Pethick 2022, Thm. 3.1) even in the deterministic case. Since the stochastic case relies on one step approximating one step of a deterministic scheme we face the same difficulty. If $\gamma_k \rightarrow 0$ then the condition would reduce to $\rho \geq 0$ (the is for example the case in (Hsieh et al. 2020, Thm. 1)).
> 4. **Fz vs F(z) notation** To avoid clutter we try to avoid parenthesis when possible and just write $Fz$. We now state in appendix B of the updated manuscript that we will use the notation interchangeably.
> 5. **What does (Hsieh et al., 2020) have to do with MVI?** They indeed never use the name MVI but their Assumption 3 is exactly what is known as the Minty variational inequality (MVI), which can be seen as a star variant of monotonicity (i.e. the condition is only required to hold with respect to the solutions). The weak MVI we study reduces to MVI when $\rho=0$. We have clarified this relationship in the revision under Remark 1 and Section 4.
>
> Proofs:
>
> 1. **What is b in equation C.1?** It is the parameter related to the Fenchel-Young inequality, so a positive real value. We have added the missing specification in the revision.
> 2. **Provide further clarification of steps** We have restructured the proofs significantly by decomposing the proof into a deterministic and a stochastic part which should help clarify greatly (see in particular Lemma E.1 and F.2-F.3). We have also expanded on many of steps throughout to make the reasoning explicit.
> 3. **Conditional expectation in C.8** We have clarified that we take conditional expectation and also made the use of the tower rule explicit.
>
> Additional comments:
>
> - **Is EG+ meant to be SEG+ in the experiments?** EG+ in Figure 2 is *not* SEG+ but rather SEG with *both* stepsizes taken constant (but with the second stepsize smaller). Thus, EG+ provably converges but only to a neighborhood. We have updated Figure 2 with more iterations to make the failure to converge to an arbitrary small error more apparent. **We have also updated the naming in the experimental section** to make it clear that stochastic oracles are used and when projections are used.
> - **Missing references** We have updated the manuscript with the references and rewritten a large portion of the related work (see Section 2).

---

> > ### Comment · Reviewer_eJWH · 2022-12-08
> > **Review Update**
> >
> > Thank you to the authors for providing further clarification on the raised points. I have read the other reviews and the rebuttal and browsed through the paper again.
> >
> > I have decided to increase my score from 6 to 8.
> >
> > I really like the updated version of the paper (especially the new version of Section 8 and of the numerical experiments).

---

### Official Review · Reviewer_sw4D · 2022-10-28

**Confidence:** 4
**Clarity, Quality, Novelty And Reproducibility:** The authors did a good job in all the…
**Correctness:** 4
**Technical Novelty And Significance:** 4
**Empirical Novelty And Significance:** 3
**Recommendation:** 8

**Strength And Weaknesses:**

Strength

This work answers affirmatively an open problem by proposing a *bias-corrected* stochastic extragradient (BCSEG+) algorithm that solves stochastic weak Minty variational inequalities without increasing the batch size. As the authors indicated, Pethick et al. (2022) "suﬃces in the special case of unconstrained quadratic games but can fail even in the monotone case ...". Also, earlier works such as Hsieh et al. (2020) adopt diminishing but larger exploration stepsize and smaller updating stepsize.


Weakness

There is not much from my perspective, as long as the proof is correct (which I took a high-leve look at but did not go into all details). Two small comments:

--MVI can be short for "monotone" variational inequality instead of "Minty" variational inequality. Adopting this shorthand as in some earlier work might cause unnecessary confusion. Therefore, I would suggest the authors avoid this shorthand as much as possible.

--The authors should do more literature reviews. Missing reference includes but not limited to "Bot et al., Minibatch Forward-Backward-Forward Methods for Solving Stochastic Variational Inequalities, 2021"

**Summary Of The Paper:**

For the first time, the authors introduces a family of stochastic extragradient-type algorithms that positively solves a class of nonconvex-nonconcave problems which can be cast as stochastic weak Minty variational inequality (MVI). In the monotone setting, extragradient methods adopt constant stepsizes and bounded batchsizes (both of which are critical in practical performances), and when extending to the weak MVI setting, only theories adopting expensive increasing batch sizes per iteration approaches are available.

**Summary Of The Review:**

This paper is theoretically strong in the sense that it made an important step in fixing SEG with constant extrapolation stepsize and diminishing updating stepsize. This makes an important step upon existing work (e.g. Hsieh et al., 2020; Diakonikolas et al., 2021; Pethick et al., 2022) even for the monotone case. This supports my high rating of this work. It would be even better if the authors have a chance to include richer empirical findings relative to this work.

---

> ### Author Response · Authors · 2022-11-12
> **Response to Reviewer sw4D**
>
> We thank the reviewer for the feedback and address all concerns below.
>
> - **MVI can be short for "monotone" variational inequality** Following several authors in the variational inequality community we take "M" to denote Minty. We have explicitly contrasted it with monotone variational inequalities by specifying that it is a star variant of monotonicity when $\rho=0$. This is now mentioned both under Section 2, Remark 1 and Section 4. We hope this provide sufficient clarification.
> - **Missing reference** We thank the reviewer for bringing the paper to our awareness. We have expanded the literature review significantly and rewritten a large portion (see Section 2).

---

### Author Response · Authors · 2022-12-07
**Deadline for discussion period is fast approaching**

Dear reviewers,

As the discussion period is coming to an end we would like to thank you for your valuable feedback. *If you have not done so already* we would greatly appreciate that you let us know if our changes have addressed your concerns. We remain available to answer any further questions.

---

### Decision · Program_Chairs · 2023-01-20

**Decision:**

Accept: poster

**Justification For Why Not Higher Score:**

This is a conservative score; ok to bump up in the calibration phase as needed. I usually reserve spotlight/oral for particularly original/innovative papers and I did not get such an impression from this paper.

**Justification For Why Not Lower Score:**

Contributions are solid and the paper should definitely be accepted.

**Metareview: Summary, Strengths And Weaknesses:**

The paper introduces a new step size schedule for a family of extragradient-type methods targeting a class of not necessarily monotone variational inequalities that satisfy a weak MVI condition, in a stochastic setting. The step size schedule, wherein one of the two step sizes per iteration is kept constant and the other one diminishes is shown to be sufficient to guarantee almost sure convergence in stochastic settings without increasing batch sizes, when a bias correction introduced in this work is applied. This answers the open question raised in prior work on whether increasing batch sizes are necessary for convergence of extragradient-type methods for variational inequalities/min-max optimization. In terms of nonasymptotic guarantees, the complexity of the proposed method is $1/\epsilon^4$ to construct a point with operator norm at most $\epsilon$. This guarantee is for the best iterate and matches what was already known in the literature for the same class on nonmonotone VIs, but without increasing batch sizes. In the case of monotone VIs, this bound is worse than what can be achieved with stochastic Halpern iteration (Cai et al, 2021), however, again, it does not require increasing batch sizes as this prior work did.

**Note From Pc:**

if the above contains the word "oral" or "spotlight" please see: "oral" presentation means -> notable-top-5% and "spotlight" means -> notable-top-25%. As stated in our emails, we are disassociating presentation type from AC recommendations

**Summary Of Ac-Reviewer Meeting:**

N/A